# Branch Scaling Manifests as Implicit Architectural Regularization for Improving Generalization in Overparameterized ResNets

**Zixiong Yu** [* 1 2]   **Guhan Chen** [* 2]   **Jianfa Lai** [2]   **Bohan Li** [2 3]   **Songtao Tian** [‡ 2]

## Abstract

Scaling factors in residual branches have emerged as a prevalent method for boosting neural network performance, especially in normalization-free architectures. While prior work has primarily examined scaling effects from an optimization perspective, this paper investigates their role in residual architectures through the lens of generalization theory. Specifically, we establish that wide residual networks (ResNets) with constant scaling factors become asymptotically unlearnable as depth increases. In contrast, when the scaling factor exhibits rapid depth-wise decay combined with early stopping, over-parameterized ResNets achieve minimax-optimal generalization rates. To establish this, we demonstrate that the generalization capability of wide ResNets can be approximated by kernel regression associated with the Neural Tangent Kernel (NTK). Our theoretical findings are validated through experiments on synthetic data and real-world classification tasks, including MNIST and CIFAR-100.

## 1. Introduction

Recently, deep neural networks have become indispensable in various real-world domains, including computer vision (Dosovitskiy et al., 2021), natural language processing (Brown et al., 2020), and multimodal learning (Radford et al., 2021). The empirical success of deep learning largely stems from architectural innovations. A crucial aspect of these innovations lies in incorporating technical components designed to stabilize and accelerate the learning dynamics.

Among these technical components, the introduction of residual blocks and skip connections has become a key mechanism for enabling deep architectures (He et al., 2016). However, the original residual networks (ResNets) still exhibit certain limitations, which become particularly pronounced in ultra-deep architectures (Szegedy et al., 2017). Due to their strong dependency on residual branches, these networks tend to amplify minor parameter perturbations, leading to significant fluctuations in model outputs and consequently resulting in training instability (Liu et al., 2020). Therefore, contemporary residual architectures are commonly integrated with normalization methods (Ioffe & Szegedy, 2015; Ba et al., 2016) to address the aforementioned issues encountered during signal propagation.

While normalization is effective, its structural complexity and computational overhead have motivated the search for simpler alternatives. One representative direction is to replace or weaken normalization through residual branch scaling. For example, De & Smith (2020) pointed out that certain normalization mechanisms in ResNets have an effect similar to downweighting the residual branch, since both improve signal propagation by adjusting the relative scale of additive paths. Based on this insight, they further proposed SkipInit (De & Smith, 2020). Following this line of thought, studies such as ReZero (Bachlechner et al., 2021) and Deep-Net (Wang et al., 2024) regard scaling factors as a more efficient or simplified design strategy. Meanwhile, works including Fixup (Zhang et al., 2019) and Zhu et al. (2025) adopt distinct approaches to replace or simplify normalization, while still reflecting the same underlying principle.

While the capability of residual branch scaling to enhance network performance has gained increasing recognition, the understanding of its underlying mechanisms remains fragmented. Existing research, including its original design intention, has predominantly focused on its advantages for optimization stability (Hayou et al., 2021a). Although its positive effects on generalization have been empirically validated (Brock et al., 2021a; Touvron et al., 2021), theoretical comprehension in this domain remains notably underdeveloped. This study shifts the perspective by directly investigating the impact of scaling from the dimension of generalization capability. We theoretically demonstrate that

---

[*]Equal Contribution. Co-first Author Email: Zixiong Yu (Work begun during Ph.D. at Tsinghua University; completed during postdoc at Huawei) <yuzx19@tsinghua.org.cn>; Guhan Chen <chen-gh23@mails.tsinghua.edu.cn>. [1]Huawei Large Model Data Technology Lab, Shenzhen [2]Tsinghua University, Beijing [3]Kyoto University, Kyoto. [‡]Correspondence to: Songtao Tian <tiansongtao.2020@tsinghua.org.cn>.

*Proceedings of the 43rd International Conference on Machine Learning*, Seoul, South Korea. PMLR 306, 2026. Copyright 2026 by the author(s).

reducing residual branch weights plays a critical role in preserving the learnability of over-parameterized ResNets, revealing a fundamental regularization mechanism previously overlooked in original architectural designs.

## 1.1. Main Content and Contributions

In this paper, we investigate how residual branch scaling factors affect the generalization capability of ResNets. To this end, we draw on insights from the theory of over-parameterized neural networks, particularly the Neural Tangent Kernel (NTK) framework, which has proven highly effective in analyzing neural network generalization. Our key contributions are summarized as follows:

1. *Uniform Convergence of ResNet to Kernel Regression.* We first demonstrate that the training dynamics of wide ResNets converge uniformly over the entire compact input domain to those of kernel regression utilizing the Residual Neural Tangent Kernel (RNTK). As a direct and practically significant corollary, the generalization capability of wide ResNets can be effectively approximated by RNTK-based kernel regression.

2. *Influence of Scaling Factor on ResNets.* Building upon the uniform convergence, our analysis further reveals that wide ResNets exhibit poor generalization performance as network depth increases when the scaling factor $\alpha$ remains constant. In striking contrast, when $\alpha$ decays sufficiently rapidly with depth, i.e., $\alpha = L^{-\gamma}$, $\gamma \in (1/2, 1]$, ResNets optimized via gradient descent with early stopping can achieve the minimax rate.

3. *Residual Branch Scaling Mechanisms in Generalization.* Building upon these theoretical findings and experimental validation, we provide deeper insights into the role of residual branch scaling: Beyond its well-documented optimization benefits during training, it can directly contribute to enhanced generalization capability, particularly in deep architectures.

Our work distinguishes itself from existing literature (see Section 1.3) in the following aspect: Although previous studies have investigated the properties of over-parameterized neural networks and preliminarily established connections between wide ResNets and RNTK, to the best of our knowledge, this work is the first to directly demonstrate their asymptotic equivalence in generalization error bounds. This discovery opens up a new research perspective: moving beyond the prevailing focus on training optimization, we provide a theoretical framework to analyze residual branch scaling directly through the lens of generalization theory.

## 1.2. Organization of the Paper

The rest of this paper is organized as follows. In Section 1.3, we review related works, while Section 1.4 introduces neces-

sary notations and settings. Section 2 provides a brief review of RNTK properties and shows the uniform convergence for wide ResNets. Section 3 presents the network performance under different scaling factors. Experiments are conducted in Section 4. Finally, Section 5 summarizes our findings and examines the limitations of the study.

## 1.3. Related works

**Residual Branch Scaling Technique** Soon after the introduction of ResNets (He et al., 2016), Szegedy et al. (2017) observed that large-scale residual networks may suffer from training instability and showed that scaling the residual branch can effectively alleviate this issue. Subsequent studies, including Hayou et al. (2021b) and Hayou et al. (2021a), provided theoretical support for the critical role of residual branch scaling in stabilizing training dynamics. Related ideas also appear in the literature on normalization-free residual networks. For example, De & Smith (2020) showed that batch normalization in deep ResNets effectively downscales the residual branch relative to the skip connection at initialization and proposed SkipInit based on this insight, while Zhang et al. (2019) introduced Fixup initialization to enable stable training without normalization. More recent architectures such as ReZero (Bachlechner et al., 2021), DeepNet (Wang et al., 2024), and Zhu et al. (2025) further adopt scaling-based designs or closely related mechanisms to improve stability and efficiency. While these studies highlight the importance of branch scaling for optimization and trainability, its direct contribution to model generalization from a theoretical perspective remains underexplored. This paper aims to address this specific gap.

**Signal Propagation in Deep Networks** A related line of work explains the effectiveness of deep architectures, including ResNets, through the lens of signal propagation and trainability. Poole et al. (2016) showed that deep random networks can exhibit exponentially growing expressivity through transient chaos, while Schoenholz et al. (2017) developed the Deep Information Propagation framework and identified the edge of chaos as a critical regime for training very deep architectures. Extending these ideas to residual networks, Yang & Schoenholz (2017) showed that skip connections fundamentally alter propagation dynamics, replacing the exponential signal and gradient behavior of standard feedforward networks with sub-exponential, often polynomial, dynamics, thereby improving the stability of deep ResNets. Closely related to residual scaling, De & Smith (2020) showed that batch normalization biases residual blocks toward the identity function by effectively downscaling the residual branch at initialization, thereby improving trainability and motivating SkipInit. Relatedly, Fischer et al. (2025) provided a field-theoretic analysis of signal propagation in ResNets and its dependence on resid-

ual branch scaling. More recently, Zhu et al. (2025) extended this perspective to Transformers, showing that propagation geometry can also predict their trainability. These works mainly explain the benefits of residual design from the perspective of propagation and optimization, rather than from that of generalization, which is the focus of our work.

**Generalization Ability of Neural Networks** The generalization behavior of neural networks has been studied from multiple theoretical perspectives. Classical approaches such as Rademacher complexity (Bartlett & Mendelson, 2002) often fail to explain the empirical success of over-parameterized models, while more recent studies emphasize implicit regularization (Neyshabur et al., 2015) and spectral bias (Rahaman et al., 2019). Particularly relevant to our work is NTK theory (Jacot et al., 2018), which shows that infinitely wide neural networks in regression can be approximated by kernel regression induced by the NTK, whose generalization performance can be characterized through the decay of kernel eigenvalues (Zhang et al., 2024). This perspective has been further developed for fully connected networks in asymptotic regimes (Arora et al., 2019; Lai et al., 2023a; Li et al., 2024), and subsequent works have attempted to extend these results to classification problems (Ji & Telgarsky, 2020; Yu et al., 2025). In contrast, the corresponding theory for ResNets remains less developed: Huang et al. (2020) mainly analyzed the connection between wide ResNets and RNTK at initialization, while Belfer et al. (2024) studied the spectral properties of RNTK for deep ResNets. Our work goes beyond these results by relating the training dynamics of wide ResNets to the kernel regression induced by the RNTK and establishing a correspondence between their generalization performances, thereby moving beyond prior work that is largely confined to initialization analysis or heuristic explanations of generalization.

## 1.4. Notations and Settings

**Fundamental Settings** Let $f_*$ be a continuous function defined on a compact set $\mathcal{X} \subseteq \mathbb{R}^d$. Let $\mu_{\mathcal{X}}$ be a uniform measure supported on $\mathcal{X}$. Suppose that we have observed $n$ independent and identically distributed (i.i.d.) samples $\mathcal{D}_n = \{(\boldsymbol{x}_i, y_i), i \in [n]\}$ drawn from the model:

$$y_i = f_*(\boldsymbol{x}_i) + \varepsilon_i, \quad i = 1, \dots, n,$$

where inputs $\boldsymbol{x}_i \overset{\text{i.i.d.}}{\sim} \mu_{\mathcal{X}}$, the noise terms $\varepsilon_i \overset{\text{i.i.d.}}{\sim} \mathcal{N}(0, \sigma^2)$ (the centered normal distribution with variance $\sigma^2$) for some fixed $\sigma > 0$, and $[n]$ denotes the index set $\{1, 2, ..., n\}$. We collect $n$ i.i.d. samples into matrix $\boldsymbol{X} := (\boldsymbol{x}_1, \dots, \boldsymbol{x}_n)^\top \in \mathbb{R}^{n \times d}$ and vector $\boldsymbol{y} := (y_1, \dots, y_n)^\top \in \mathbb{R}^n$.

Throughout this paper, we focus exclusively on the regression setting. Our goal is to construct a regression estimator $\hat{f}_n$ based on these $n$ samples to minimize the excess risk,

defined as the difference between the expected risk of the estimator and that of the true function: $\mathcal{L}(\hat{f}_n) - \mathcal{L}(f_*)$, where $\mathcal{L}(f) = \mathbb{E}_{(\boldsymbol{x}, y)}[(f(\boldsymbol{x}) - y)^2]$. A direct calculation yields the following expression for the excess risk:

$$\mathcal{E}(\hat{f}_n) = \mathcal{L}(\hat{f}_n) - \mathcal{L}(f_*) = \int_{\mathcal{X}} \left[ \hat{f}_n(\boldsymbol{x}) - f_*(\boldsymbol{x}) \right]^2 \mathrm{d}\mu_{\mathcal{X}}(\boldsymbol{x}).$$

Evidently, the excess risk serves as an equivalent metric for evaluating the generalization performance of $\hat{f}_n$.

**Interpolation Space** Denote $L^2(\mathcal{X})$ as the $L^2$ space on $\mathcal{X}$. Throughout the paper, we denote by $\mathcal{H}$ a separable RKHS on $\mathcal{X}$ with respect to a continuous kernel function $K$. We also assume that $\sup_{\boldsymbol{x} \in \mathcal{X}} K(\boldsymbol{x}, \boldsymbol{x}) \leq C$ for some constant $C$. The celebrated Mercer's theorem states that there exist non-negative eigenvalues $\lambda_1 \geq \lambda_2 \geq \cdots$ and eigenfunctions $e_1, e_2, \cdots \in L^2(\mathcal{X})$ such that $\langle e_i, e_j \rangle_{L^2(\mathcal{X})} = \delta_{ij}$ and

$$K(\boldsymbol{x}, \boldsymbol{x}') = \sum_{j=1}^{\infty} \lambda_j e_j(\boldsymbol{x}) e_j(\boldsymbol{x}'), \tag{1}$$

where the series converges in $L^2(\mathcal{X})$. With these eigenvalues and eigenfunctions, the interpolation space for $s \geq 0$ is defined as (Steinwart & Scovel, 2012; Fischer & Steinwart, 2020; Zhang et al., 2024):

$$[\mathcal{H}]^s := \left\{ \sum_{j=1}^{\infty} f_j e_j \in L^2(\mathcal{X}) \,\Big|\, \sum_{j=1}^{\infty} f_j^2 / \lambda_j^s < \infty \right\}$$

equipped with the norm $\| \sum_{j=1}^{\infty} f_j e_j \|_{[\mathcal{H}]^s}^2 := \sum_{j=1}^{\infty} f_j^2 / \lambda_j^s$. In particular, we have $[\mathcal{H}]^0 \subseteq L^2(\mathcal{X})$ and $[\mathcal{H}]^1 = \mathcal{H}$. For $s_1 > s_2 \geq 0$, we have the inclusion $[\mathcal{H}]^{s_1} \subset [\mathcal{H}]^{s_2}$.

**Other Notations** For a function $h : \mathcal{X} \to \mathbb{R}$, we denote $h(\boldsymbol{X}) = (h(\boldsymbol{x}_1), \dots, h(\boldsymbol{x}_n))^\top \in \mathbb{R}^{n \times 1}$. For a symmetric kernel $k : \mathcal{X} \times \mathcal{X} \to \mathbb{R}$, we write $k(\boldsymbol{x}, \boldsymbol{X}) = (k(\boldsymbol{x}, \boldsymbol{x}_1), \dots, k(\boldsymbol{x}, \boldsymbol{x}_n)) \in \mathbb{R}^{1 \times n}$ for the kernel evaluation vector and $k(\boldsymbol{X}, \boldsymbol{X}) = (k(\boldsymbol{x}_i, \boldsymbol{x}_j))_{i,j=1}^n \in \mathbb{R}^{n \times n}$ for the Gram matrix. For two sequences $a_n$ and $b_n$, we write $a_n = \mathcal{O}(b_n)$ or $b_n = \Omega(a_n)$ if there exists a constant $C > 0$ such that $a_n \leqslant C b_n$. We also denote $a_n = \Theta(b_n)$ or $a_n \asymp b_n$ if $a_n = \mathcal{O}(b_n)$ and $a_n = \Omega(b_n)$ both hold. We write $a_n = o(b_n)$ if $a_n / b_n \to 0$ as $n \to \infty$. We will use $\text{poly}(x, y, \dots)$ to represent a polynomial of $x, y, \dots$ whose coefficients are absolute positive constants.

## 2. Uniform Convergence of ResNet to Kernel Regression

In this section, we show that the training behavior of wide ResNets can be characterized by kernel regression, with the corresponding kernel given by the RNTK. We adopt the over-parameterized framework for both theoretical and practical reasons: it helps overcome technical obstacles in

analysis, and modern neural networks are typically over-parameterized. Moreover, in large-scale regimes, architectural techniques such as residual connections and residual branch scaling become increasingly important, further motivating the study of over-parameterization for understanding their roles and underlying mechanisms.

To facilitate theoretical analysis, we focus on ResNets with a branch scaling factor $\alpha$. As previously discussed, scaling factor $\alpha$ plays a pivotal role in modulating inter-layer signal propagation and ensuring training stability. More significantly, we will demonstrate that this regulatory mechanism extends far beyond training stability, exerting substantial influence on model generalization performance.

### 2.1. Review of ResNet and RNTK

**Network Architecture and Initialization** Residual connections have become a fundamental component of modern deep neural network architectures (He et al., 2016; Vaswani et al., 2017). To clearly isolate the effect of residual connections, we focus on fully connected networks with residual connections. Specifically, we adopt the multi-hidden-layer ResNet architecture studied in Huang et al. (2020); Belfer et al. (2024); Tirer et al. (2022), which has width $m$, depth $L$, and includes a bias term in the input layer, as follows[1]:

$$f(\boldsymbol{x}, \boldsymbol{\theta}) = \boldsymbol{v}^\top \boldsymbol{x}^{(L)};$$
$$\boldsymbol{x}^{(\ell)} = \boldsymbol{x}^{(\ell-1)} + \alpha \sqrt{\tfrac{1}{m}} \boldsymbol{V}^{(\ell)} \sigma \left( \sqrt{\tfrac{2}{m}} \boldsymbol{W}^{(\ell)} \boldsymbol{x}^{(\ell-1)} \right);$$
$$\boldsymbol{x}^{(0)} = \sqrt{\tfrac{1}{m}} (\boldsymbol{A}\boldsymbol{x} + \boldsymbol{b}),$$

where $\ell \in [L]$ with parameters $\boldsymbol{v} \in \mathbb{R}^m$, $\boldsymbol{V}^{(\ell)}, \boldsymbol{W}^{(\ell)} \in \mathbb{R}^{m \times m}$, $\boldsymbol{A} \in \mathbb{R}^{m \times d}$ and $\boldsymbol{b} \in \mathbb{R}^m$. In addition, $\sigma(x) := \max\{x, 0\}$ denotes the ReLU activation function. All parameters are initialized as i.i.d. random variables drawn from the standard normal distribution.

$$\text{i.e., } \boldsymbol{v}_i, \boldsymbol{V}_{i,j}^{(\ell)}, \boldsymbol{W}_{i,j}^{(\ell)}, \boldsymbol{A}_{i,k}, \boldsymbol{b}_i \overset{\text{i.i.d.}}{\sim} \mathcal{N}(0,1)$$

for $i, j \in [m]$, $k \in [d]$ and $\ell \in [L]$. As in Huang et al. (2020), we assume that $\boldsymbol{v}$, $\boldsymbol{A}$ and $\boldsymbol{b}$ are all fixed at their initialization, while $\boldsymbol{V}^{(\ell)}$ and $\boldsymbol{W}^{(\ell)}$ are trainable. Thus, $\boldsymbol{\theta} = \text{vec}(\{\boldsymbol{W}^{(\ell)}, \boldsymbol{V}^{(\ell)}\}_{\ell=1}^L)$ represents the trainable parameters.

The parameter $\alpha$, serving as the scaling factor for residual branches, has undergone significant evolution in deep network research. In the seminal ResNet study (He et al., 2016), this parameter was simply set as a constant $\alpha = 1$, under

---

[1]For technical tractability, the analysis in the Appendix uses a modified architecture and initialization, following standard practices in the literature. These modifications do not alter our main findings. A detailed discussion is provided near Equation (4) and in Appendix B.1. The standard architecture is presented here to help readers grasp the main ideas without technical distractions.

which configuration normalization layers were typically required for optimal performance. With advancing research, Zhang et al. (2019) innovatively adopted a power-law decay formulation when proposing the Fixup initialization method, thereby replacing traditional normalization operations. Building upon these research foundations, this paper employs the following unified expression: $\alpha = C \cdot L^{-\gamma}$ for $C > 0$ and $0 \leq \gamma \leq 1$. A comparative analysis of network performance between constant $\alpha$ and depth-decaying $\alpha$ configurations can effectively demonstrate the effect of residual branch scaling on network behavior.

**Training** Training methods vary substantially across tasks, ranging from classical supervised learning to the increasingly prominent reinforcement learning paradigm (Schulman et al., 2017; Zhang et al., 2025). For technical tractability, however, this paper focuses on the basic regression setting, where the squared loss is typically used:

$$\widehat{\mathcal{L}}(\boldsymbol{\theta}) = \frac{1}{2n} \sum_{i=1}^n (f(\boldsymbol{x}_i, \boldsymbol{\theta}) - y_i)^2.$$

Neural networks are typically trained by gradient descent or its variants to minimize the empirical loss. For simplicity, we consider gradient flow, i.e., the continuous-time limit of gradient descent as the learning rate tends to zero (Li et al., 2024). Although practical training procedures are more involved, this simplification allows us to isolate the intrinsic effect of residual branch scaling.

For the parameters $\boldsymbol{\theta}$ at time $t \geq 0$ (denoted by $\boldsymbol{\theta}_t$), the gradient flow is given by the following differential equation:

$$\dot{\boldsymbol{\theta}}_t = -\nabla_{\boldsymbol{\theta}} \widehat{\mathcal{L}}(\boldsymbol{\theta}) = -\tfrac{1}{n} \nabla_{\boldsymbol{\theta}} f(\boldsymbol{X}, \boldsymbol{\theta}_t)(f(\boldsymbol{X}, \boldsymbol{\theta}_t) - \boldsymbol{y}) \quad (2)$$

where $\nabla_{\boldsymbol{\theta}} f(\boldsymbol{X}, \boldsymbol{\theta}_t)$ is a $2Lm^2 \times n$ matrix. From the gradient flow equation of the parameters, we can directly derive the evolution equation for the ResNet regression function:

$$\dot{f}(\boldsymbol{x}, \boldsymbol{\theta}_t) = -\tfrac{1}{n} r_t^m(\boldsymbol{x}, \boldsymbol{X}) (f(\boldsymbol{X}, \boldsymbol{\theta}_t) - \boldsymbol{y}), \quad (3)$$

where $r_t^m(\boldsymbol{x}, \boldsymbol{x}') = \nabla_{\boldsymbol{\theta}} f(\boldsymbol{x}, \boldsymbol{\theta}_t)^\top \nabla_{\boldsymbol{\theta}} f(\boldsymbol{x}', \boldsymbol{\theta}_t)$, which is called Empirical Residual Neural Tangent Kerne (Empirical RNTK) in this paper.

**RNTK and Kernel Regression** The gradient flow equations (2) and (3) imply highly nonlinear dynamics that are generally intractable. However, when $r_t^m(\boldsymbol{x}, \boldsymbol{X})$ is time-invariant, Equation (3) reduces to the gradient flow dynamics of standard kernel regression. Although this condition does not generally hold for neural networks, it has been empirically observed or preliminarily characterized (Jacot et al., 2018; Huang et al., 2020; Tirer et al., 2022) that, as the width $m$ approaches infinity, the Empirical RNTK $r_t^m(\boldsymbol{x}, \boldsymbol{x}')$ concentrates to a time-invariant kernel $r(\boldsymbol{x}, \boldsymbol{x}')$, referred to as the Residual Neural Tangent Kernel (RNTK), i.e., $r_t^m(\boldsymbol{x}, \boldsymbol{x}') \overset{p}{\to} r(\boldsymbol{x}, \boldsymbol{x}')$ as $m \to \infty$.

Furthermore, in Section 2.2, we establish stronger uniform convergence results that hold throughout the entire training process under certain conditions. Therefore, we consider the RNTK-based kernel regressor $\hat{f}_t^{\mathrm{RNTK}}(\boldsymbol{x})$, governed by the following gradient flow equation:

$$\frac{\partial}{\partial t} \hat{f}_t^{\mathrm{RNTK}}(\boldsymbol{x}) = -\frac{1}{n} r(\boldsymbol{x}, \boldsymbol{X})(\hat{f}_t^{\mathrm{RNTK}}(\boldsymbol{X}) - \boldsymbol{y}). \quad (4)$$

Moreover, if both Equation (3) and (4) are initialized at zero, the ResNet regressor $\hat{f}_t^{\mathrm{ResNet}}(\boldsymbol{x}) := f(\boldsymbol{x}, \boldsymbol{\theta}_t)$ can be well approximated by $\hat{f}_t^{\mathrm{RNTK}}(\boldsymbol{x})$. Crucially, based on the uniform convergence established in our analysis, this approximation extends to the generalization error (see Corollary 2.2). In addition, Equation (4) admits the following closed-form solution ($\boldsymbol{I}$ denotes the identity matrix):

$$\hat{f}_t^{\mathrm{RNTK}}(\boldsymbol{x}) = r(\boldsymbol{x}, \boldsymbol{X}) r(\boldsymbol{X}, \boldsymbol{X})^{-1} \left[ \boldsymbol{I} - \mathrm{e}^{-\frac{1}{n} r(\boldsymbol{X}, \boldsymbol{X}) t} \right] \boldsymbol{y}.$$

To simplify the setting and maintain focus, we adopt in the following a commonly used mirrored network architecture together with its corresponding initialization method from the existing literature (Hu et al., 2020; Chizat et al., 2019; Lai et al., 2023a; Li et al., 2024); the sole purpose of this adjustment is to ensure that $\hat{f}_0^{\mathrm{ResNet}}$ is initialized as the zero function. Since this adjustment is purely technical, we do not elaborate on it in the main text and instead defer the details to Section B.1 in the Supplementary Material for ease of reading. For further discussion on the effect of zero initialization, we refer the reader to Chen et al. (2024).

**Explicit Expression of the RNTK** We now present the explicit expression of the RNTK, which also serves as its formal definition in this paper. First, we introduce the following two functions:

$$\kappa_0(u) = \frac{1}{\pi}(\pi - \arccos u);$$
$$\kappa_1(u) = \frac{1}{\pi}\left( u\left(\pi - \arccos u\right) + \sqrt{1 - u^2} \right)$$

and let $\boldsymbol{x}, \boldsymbol{x}' \in \mathcal{X}$. The NTK of an $L$-hidden-layer ResNet, denoted as $r(\boldsymbol{x}, \boldsymbol{x}')$, is given by Huang et al. (2020):

$$r(\boldsymbol{x}, \boldsymbol{x}') = \alpha^2 \left\| \binom{\boldsymbol{x}}{1} \right\| \left\| \binom{\boldsymbol{x}'}{1} \right\| \cdot r_0(\tilde{\boldsymbol{x}}, \tilde{\boldsymbol{x}}');$$
$$r_0(\tilde{\boldsymbol{x}}, \tilde{\boldsymbol{x}}') = \sum_{\ell=1}^{L} B_{\ell+1} \Big[ (1+\alpha^2)^{\ell-1} \kappa_1\left(\frac{K_{\ell-1}}{(1+\alpha^2)^{\ell-1}}\right) \quad (5)$$
$$+ K_{\ell-1} \cdot \kappa_0\left(\frac{K_{\ell-1}}{(1+\alpha^2)^{\ell-1}}\right) \Big]$$

where $\tilde{\boldsymbol{x}} = \binom{\boldsymbol{x}}{1}/\left\|\binom{\boldsymbol{x}}{1}\right\|$, $\tilde{\boldsymbol{x}}' = \binom{\boldsymbol{x}'}{1}/\left\|\binom{\boldsymbol{x}'}{1}\right\|$ and $K_0 = \tilde{\boldsymbol{x}}^\top \tilde{\boldsymbol{x}}'$, $B_{L+1} = 1$,

$$K_\ell = K_{\ell-1} + \alpha^2(1+\alpha^2)^{\ell-1}\kappa_1\left(\frac{K_{\ell-1}}{(1+\alpha^2)^{\ell-1}}\right);$$
$$B_\ell = B_{\ell+1}\left[1 + \alpha^2 \kappa_0\left(\frac{K_{\ell-1}}{(1+\alpha^2)^{\ell-1}}\right)\right]$$

for $\ell \in [L]$. In the above equations, $\|\cdot\|$ denotes the Euclidean norm, $K_\ell$ and $B_\ell$ are abbreviations for $K_\ell(\tilde{\boldsymbol{x}}, \tilde{\boldsymbol{x}}')$ and $B_\ell(\tilde{\boldsymbol{x}}, \tilde{\boldsymbol{x}}')$, respectively.

## 2.2. Empirical RNTK Uniformly Converges to RNTK

Previous studies have demonstrated that wide neural networks can be approximated by kernel regressors (Jacot et al., 2018). However, the approximations provided by most prior works are incomplete, as they fail to characterize the generalization capability, which is precisely a crucial component.

One of the main technical contributions of this paper is to address the aforementioned issues, and a simplified version of the relevant results is stated in the following theorem (here we simplify the description of the requirements on $m$, see Theorem B.1 for the complete statement):

**Theorem 2.1.** *For any given training data* $\{(\boldsymbol{x}_i, y_i), i \in [n]\}$ *and any* $\delta \in (0, 1)$,

$$\sup_{t \geq 0} \sup_{\boldsymbol{x}, \boldsymbol{x}' \in \mathcal{X}} |r_t^m(\boldsymbol{x}, \boldsymbol{x}') - r(\boldsymbol{x}, \boldsymbol{x}')| \leq \mathcal{O}\left(m^{-\frac{1}{12}}\sqrt{\log m}\right)$$

*holds with probability at least* $1 - \delta$ *for sufficiently large* $m$.

This result may not be surprising intuitively, but it has long lacked rigorous justification. For instance, the closely related work of Huang et al. (2020) only established pointwise convergence at initialization, whereas our analysis goes beyond the static kernel properties at initialization and provides uniform control over the entire training dynamics.

Moreover, based on the uniform convergence of the empirical RNTK, we obtain the following result (we also simplify the statement of this corollary, see Corollary B.2 for the complete statement), which quantifies the proximity between the generalization error of ResNets $\mathcal{E}(\hat{f}_t^{\mathrm{ResNet}})$ and those of RNTK-regressors $\mathcal{E}(\hat{f}_t^{\mathrm{RNTK}})$, demonstrating that their excess risks are asymptotically equivalent.

**Corollary 2.2.** *For any given training data* $\{(\boldsymbol{x}_i, y_i), i \in [n]\}$, *any* $\epsilon > 0$ *and any* $\delta \in (0, 1)$,

$$\sup_{t \geq 0} \left| \mathcal{E}(\hat{f}_t^{\mathrm{ResNet}}) - \mathcal{E}(\hat{f}_t^{\mathrm{RNTK}}) \right| \leq \epsilon$$

*holds with probability at least* $1 - \delta$ *for sufficiently large* $m$.

We highlight the critical importance of establishing uniform convergence in this context. Achieving such convergence is not only technically more demanding, but also provides a necessary theoretical guarantee for approximating the generalization error. Since the excess risk is defined as an expectation (integral) over the entire input distribution, prior analyses are generally insufficient to bound the difference between the integrals. Uniform convergence over the entire domain ensures that the proximity of the training dynamics translates validly to that of the generalization errors.

Building on Corollary 2.2, we next investigate the impact of residual branch scaling on generalization performance, which forms the focus of the next section. For results on the generalization of finite-depth ResNets, we refer to the related earlier work (Lai et al., 2023b).

# 3. Effects of Residual Branch Scaling on Generalization

In this section, we investigate the impact of scaling factors on the generalization capability of over-parameterized ResNets. We first prove that a constant $\alpha$ leads to asymptotic unlearnability, as the generalization error is theoretically lower-bounded by a positive constant independent of the sample size. In striking contrast, we demonstrate that when $\alpha$ decays rapidly with depth, ResNets trained with appropriate early stopping can achieve the minimax-optimal generalization rate. Finally, we discuss how these findings characterize residual branch scaling as a critical form of implicit architectural regularization.

## 3.1. Constant Scaling Factors Lead to Poor Generalization in Deep Architectures

Building on Corollary 2.2, which establishes that the generalization error of ResNets can be approximated by that of RNTK regression, we now turn our focus to analyzing the generalization properties of the RNTK regression. Motivated by the prevalence of ultra-deep architectures (Wang et al., 2024) and the need to highlight distinct asymptotic behaviors, this section specifically examines the regime of extremely large depths.

To explicitly characterize the role of depth, we add the superscript $(L)$ to the kernel notation. For analytical convenience, we restrict the inputs to the unit sphere, i.e., $\mathcal{X} = \mathbb{S}^{d-1}$. We observe that for $\boldsymbol{x} \in \mathbb{S}^{d-1}$, the diagonal entries of the kernel satisfy $r^{(L)}(\boldsymbol{x}, \boldsymbol{x}) = 4L\alpha^2(1+\alpha^2)^{L-1}$, which diverges as $L \to \infty$. To ensure a well-defined limit, we consider the normalized RNTK (denoted as $\bar{r}^{(L)}$, and hereafter referred to simply as the RNTK), given that input-independent scaling factors do not affect the prediction or generalization performance of kernel regression:

$$\bar{r}^{(L)}(\boldsymbol{x}, \boldsymbol{x}') := r^{(L)}(\boldsymbol{x}, \boldsymbol{x}') / \left[ 4L\alpha^2(1+\alpha^2)^{L-1} \right]. \quad (6)$$

We first derive the limiting behavior of $\bar{r}^{(L)}(\boldsymbol{x}, \boldsymbol{x}')$ as $L$ approaches infinity when $\alpha$ is a fixed positive constant.

**Theorem 3.1.** *Let $\alpha$ be a fixed positive constant. For any given $\boldsymbol{x}, \boldsymbol{x}' \in \mathbb{S}^{d-1}$, the normalized RNTK satisfies:*

$$\bar{r}^{(L)}(\boldsymbol{x}, \boldsymbol{x}') = \begin{cases} \frac{1}{4} + \mathcal{O}\left( \frac{\text{polylog } L}{L} \right), & \text{if } \boldsymbol{x} \neq \boldsymbol{x}'; \\ 1, & \text{if } \boldsymbol{x} = \boldsymbol{x}'. \end{cases}$$

*As a result, for any $t \geq 0$, we have $\mathcal{E}(f_t^{\bar{r}^\infty}) = \Theta(1)$, where $f_t^{\bar{r}^\infty}$ is the kernel regression predictor associated with the limiting kernel $\bar{r}^\infty = \lim_{L \to \infty} \bar{r}^{(L)}$.*

Theorem 3.1 reveals that with a constant scaling factor, the RNTK degenerates into a "spike" kernel (a constant background value plus a Dirac delta at the diagonal) as depth

increases. This implies that the network loses its discriminative power, effectively treating all distinct inputs as equally correlated. Consequently, the model fails to capture the underlying geometric structure of the data distribution, resulting in trivial generalization performance that does not improve with sample size.

## 3.2. Rapidly-Decaying Scaling Factors Achieve the Minimax Optimal Rate

Having established the limitations of constant scaling, we now demonstrate that for optimal learnability, $\alpha$ should decay rapidly with increasing depth. To rigorously characterize the generalization performance, we first specify the function class containing the target regression function $f_*$. We introduce the following standard source condition:

**Assumption 3.2** (Source Condition). The regression target function satisfies $f_* \in [\mathcal{H}]^s$ with $\|f_*\|_{[\mathcal{H}]^s} \leq R$ for some positive constant $R$, where $s > 0$ denotes the smoothness parameter and $\mathcal{H}$ is the RKHS associated with the one-hidden-layer RNTK $r^{(1)}$.

The above condition is standard in the kernel regression literature (Caponnetto & De Vito, 2007; Yao et al., 2007; Raskutti et al., 2014; Blanchard & Mücke, 2018; Lin et al., 2020; Zhang et al., 2024). It is a relatively mild assumption, since $s$ can be arbitrarily small, and in the limit $s \to 0$, the space $[\mathcal{H}]^s$ approaches $L^2(\mathbb{S}^{d-1})$. Based on this setup, we can derive the following conclusion:

**Theorem 3.3.** *Let $\alpha = L^{-\gamma}$ for $\gamma \in (1/2, 1]$. Suppose Assumption 3.2 holds. For any given $\delta \in (0, 1)$, if the ResNet is trained via gradient flow with early stopping at time $t_* \propto n^{d/[(s+1)d-1]}$, then for sufficiently large $m$ and $L$, there exists a constant $C$ independent of $\delta$ and $n$, such that*

$$\mathcal{E}(\hat{f}_{t_*}^{\text{ResNet}}) \leq Cn^{-\frac{sd}{(s+1)d-1}} \log^2(6/\delta)$$

*holds with probability at least $1 - \delta$ for sufficiently large $n$.*

In sharp contrast to the asymptotic unlearnability established in Section 3.1, ResNets with rapidly decaying scaling factors exhibit strong performance. Notably, under Assumption 3.2, the derived generalization error bound achieves the minimax-optimal rate. This indicates that properly scaled residual branches allow the model to preserve its capacity to capture complex geometric structures as depth increases.

Crucially, the decay rate must be sufficiently fast. If the decay is too slow, the kernel may still degenerate, albeit at a slower rate. We illustrate this with the case where $\gamma = 1/4$:

**Theorem 3.4.** *Let $\alpha = L^{-1/4}$. For any given $\boldsymbol{x}, \boldsymbol{x}' \in \mathbb{S}^{d-1}$, the normalized RNTK satisfies: $\bar{r}^{(L)}(\boldsymbol{x}, \boldsymbol{x}') = 1$ for $\boldsymbol{x} = \boldsymbol{x}'$ and $\bar{r}^{(L)}(\boldsymbol{x}, \boldsymbol{x}') = 1/4 + \mathcal{O}(1/\text{polylog } L)$ for $\boldsymbol{x} \neq \boldsymbol{x}'$.*

Theoretically, the early stopping mechanism in Theorem 3.3 is indispensable to mitigate overfitting to noise (Li et al.,

2023). In practical, explicit regularization methods, such as weight decay ($L_2$ regularization) or cross-validation, serve a similar purpose and can achieve comparable effects.

### 3.3. Further Discussion

We have presented the core theoretical findings of this study: as network depth increases, which is a prevailing trend in modern architectures, proper scaling of residual branches plays a decisive role in maintaining generalization performance. This is because in extremely deep networks, the absence of such scaling leads to asymptotic unlearnability. Consequently, this scaling mechanism functions as an implicit architectural regularizer. Although originally designed to stabilize the optimization process, it implicitly governs generalization, playing a critical role in practical networks that has remained largely unrecognized.

Crucially, successful training does not necessarily translate into improved model performance. Although wide networks can interpolate the training data (Hornik et al., 1989), such fitting ability does not guarantee better generalization: it may reflect memorization of individual training samples or reliance on spurious but predictive correlations in the training distribution (Zhou et al., 2026a;b). These observations motivate a careful distinction between the roles of residual-branch scaling in optimization stability and generalization.

It is also worth noting that in practical architectures, residual branches are typically coupled with normalization layers. Since extensive literature suggests that normalization induces an implicit scaling effect (Brock et al., 2021a), our theoretical framework offers valuable insights into how normalization influences generalization capability. Furthermore, while our analytical results are derived within a regression setting, the fundamental phenomena observed, particularly the criteria for kernel degeneration, are structural in nature and thus likely to hold in broader contexts.

## 4. Experiments

In this section, we present comprehensive numerical experiments to corroborate the theoretical findings of this paper. First, we visualize the asymptotic behavior of the RNTK when $\alpha$ is constant or decays slowly. We observe that the kernel value for distinct inputs converges to $1/4$ as the depth $L$ increases, providing direct empirical validation for Theorem 3.1 and Theorem 3.4. Second, we demonstrate that a sufficiently rapid decay of $\alpha$ is critical for the generalization performance of both RNTK-based kernel regression and finite-width Convolutional ResNets (ConvResNets) on synthetic and real-world datasets. This aligns with our theoretical analysis in Section 3.

While the theory relies on standard simplifying assumptions (e.g., fully connected architectures), we have extended the

experimental evaluation to broader scenarios (e.g., ConvResNets). These experiments demonstrate that the conclusions of this paper remain robust beyond the strict theoretical settings discussed in the main text. In addition, we compare the proposed scaling strategy with standard normalization techniques to preliminarily explore the connections between residual branch scaling and normalization mechanisms.

### 4.1. Fixed Kernel

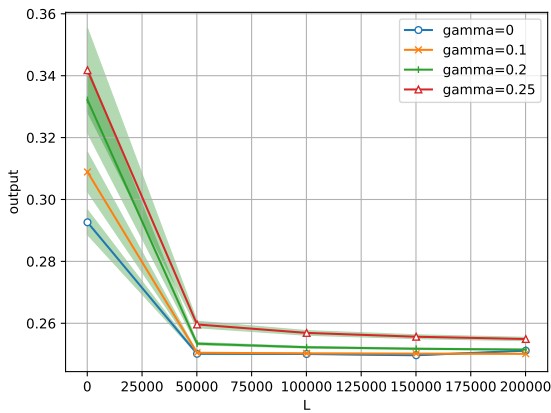

*Figure 1.* Average RNTK values for random input pairs $\boldsymbol{x}, \boldsymbol{x}' \sim \text{Uniform}(\mathbb{S}^2)$ as a function of depth $L$.

This subsection empirically verifies the asymptotic behavior of the RNTK in the large depth limit, as established in Theorem 3.1 and Theorem 3.4. To achieve this, we compute the average normalized RNTK value $\bar{r}^{(L)}(\boldsymbol{x}, \boldsymbol{x}')$ over 100 randomly sampled pairs of distinct inputs drawn from $\text{Uniform}(\mathbb{S}^2)$ (the uniform distribution over unit sphere $\mathbb{S}^2$), for increasing values of $L$. The results are illustrated in Figure 1, where $\gamma$ takes values in $\{0, 0.1, 0.2, 0.25\}$, $L$ is selected from $\{100, 50000, 100000, 150000, 200000\}$. The shaded regions denote the standard error over trials. We observe that as $L$ increases, the kernel value for distinct inputs progressively converges to $1/4$. Notably, at $L = 200,000$, the value closely approximates this theoretical limit, providing strong empirical support for our asymptotic analysis.

### 4.2. Improve Generalization Ability by Rapidly-Decaying Residual Branch Scaling

In this subsection, we empirically demonstrate that the residual branch scaling strategy proposed in this paper significantly enhances the generalization capability of ResNets. To this end, we conduct a systematic investigation using two distinct experimental frameworks: (1) kernel regression via gradient descent using the RNTK, and (2) finite-width ConvResNets. Specifically, in Section 4.2.1, we explore the scaling parameter $\alpha = L^{-\gamma}$ across a range of values for $\gamma \in \{0, 1/4, 1/2, 3/4, 1\}$. Our empirical analysis encompasses both synthetic and real-world datasets.

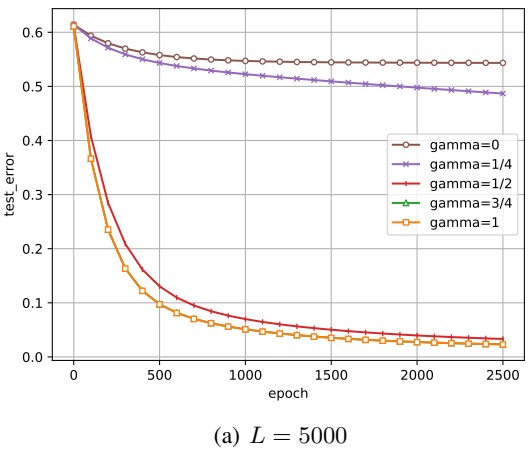 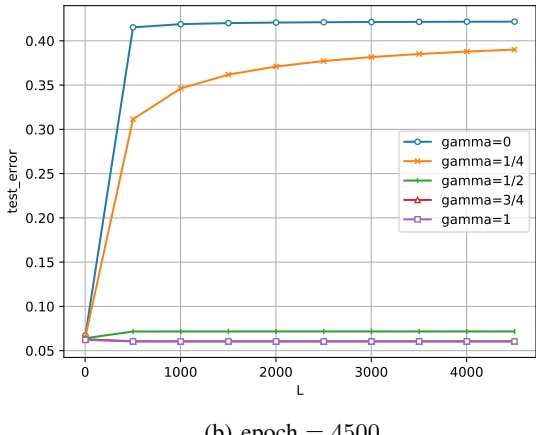

(a) $L = 5000$                    (b) epoch = 4500

*Figure 2.* Test error on synthetic data drawn from $\mathrm{Uniform}(\mathbb{S}^2)$ with different values of $\gamma$. Note that the curves for $\gamma = 3/4$ and $\gamma = 1$ almost coincide.

The results consistently show that networks with sufficiently large $\gamma$ values (particularly $\gamma > 1/2$) achieve significantly lower test errors compared to their counterparts with slow or constant decay ($\gamma = 0, 1/4$). Particularly noteworthy are the ConvResNet experiments in Section 4.2.2, which demonstrate that appropriate residual scaling can deliver comparable effects to batch normalization. By acting as an implicit regularizer, it effectively mitigates overfitting. These findings further underscore the critical role of depth-dependent scaling mechanisms in preserving the learnability of deep architectures.

### 4.2.1. RNTK-BASED KERNEL METHODS

**Synthetic Data**   We begin by analyzing a synthetic regression task where the data $(\boldsymbol{X}, Y)$ are generated by:

$$Y = \langle \boldsymbol{X}, \boldsymbol{\beta} \rangle + 0.1 \cdot \epsilon; \quad \boldsymbol{X} \sim \mathrm{Uniform}(\mathbb{S}^2),$$

where $\boldsymbol{\beta} = (1, 1, 1)^\top$, $\epsilon \sim \mathcal{N}(0, 1)$. We generate a total of 200 samples, which are randomly partitioned into a training set of 160 samples and a test set of 40 samples. We calculate the test error of RNTK-based kernel regression trained via gradient descent with various training epochs, $\gamma$ and $L$. The learning rate is fixed at $10^{-4}$.

The results are presented in Figure 2. In the left panel, we fix the depth at $L = 5000$ and plot the test error evolution over training epochs for varying $\gamma$. We observe that the test error for small decay values ($\gamma < 1/2$) is significantly higher than that for larger $\gamma$. In the right panel, we fix the training duration to 4500 epochs and examine the test error as a function of depth $L$ for different values of $\gamma$. Beyond the general trends consistent with the left panel, we observe a critical divergence in asymptotic behavior: for rapid decay rates ($\gamma = 3/4, 1$), the test error decreases as $L$ increases; conversely, for slow decay rates ($\gamma = 0, 1/4, 1/2$), the error

deteriorates as depth increases, particularly in the large-depth regime. This observation aligns perfectly with our theoretical predictions in Section 3.

**Real-World Data (MNIST)**   We extend our empirical validation to a real-world classification task using the MNIST dataset. We randomly sample a subset of 20000 images for training and 10000 images for testing. The results are shown in Figure 3. We fix the depth at $L = 50$ and evaluate the test error of RNTK-based kernel logistic regression with varying $\gamma$. We observe that, consistently across training epochs, the test error for rapid decay rates ($\gamma \geq 1/2$) is significantly lower than that for slow decay rates ($\gamma < 1/2$). This observation aligns with our theoretical results in Section 3.

### 4.2.2. CONVRESNETS ON REAL-WORLD DATASETS

We conduct experiments on CIFAR-100 (Krizhevsky et al., 2009) using the standard ResNet-34 model as the representative ConvResNet. This architecture represents a moderately deep network with standard width, deliberately selected to relax the strict theoretical assumptions of infinite width and depth. This validates the applicability of our theory to practical architectures (Tiny-ImageNet and Transformer experiments are provided in Appendix A.1). The network is optimized using Adam with an initial learning rate of $3 \times 10^{-4}$ and an exponential decay factor of $0.95$ per epoch.

To systematically evaluate the effectiveness of different strategies, we compare three distinct approaches: (i) a vanilla baseline without any normalization or scaling, (ii) Residual Branch Scaling with $\gamma = 1$, and (iii) standard Batch Normalization. The comparative results regarding training and test accuracy across these strategies (abbreviated as *base*, *scale*, and *norm*, respectively) are presented in Figure 4. Our experimental findings reveal two key insights:

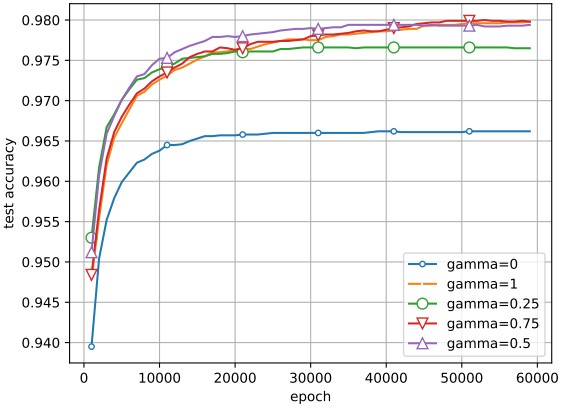

*Figure 3.* Test accuracy of the RNTK-regressor on the MNIST dataset for different values of residual scaling exponent $\gamma$.

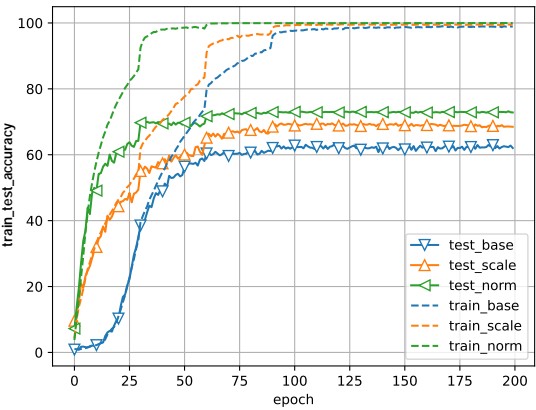

*Figure 4.* Training and test accuracy of ResNet-34 on the CIFAR-100 dataset with different strategies.

(1) The scaling approach indeed improves the final performance of the model (as does the normalization scheme). It is important to note that the slightly inferior performance of the scaling method relative to normalization in this context is expected. To facilitate theoretical analysis, this paper intentionally simplifies certain settings. Notably, numerous studies proposing normalization-replacement schemes based on similar residual branch scaling techniques (Zhang et al., 2019; Brock et al., 2021b) have employed more sophisticated and intricate configurations to achieve performance on par with normalization techniques. Our goal is to provide a theoretical explanation for these techniques, rather than to propose a new state-of-the-art method.

(2) The performance improvement stems from the scaling method's ability to preserve the model's generalization adaptability to data distributions, rather than merely optimizing the training process: Although both the scaling approach and normalization significantly improve the training speed (this feature confirms previous conclusions regarding the role of these techniques in optimization), all methods achieve nearly 100% training accuracy after sufficient training. On this basis, the scaling approach and normalization still enhance the accuracy on the test set, indicating that the performance improvement of scaling approach and normalization lies not only in making training easier but also in enhancing the generalization capability.

In addition, even when training accuracy is close to 100%, the training loss may continue to decrease. We therefore supplement our analysis with a matched-training-loss evaluation to further support our conclusions. Due to space limitations, we present these results in Appendix A.2. This experiment also includes additional baselines, including Fixup (Zhang et al., 2019), ReZero (Bachlechner et al., 2021), and DeepNorm (Wang et al., 2024), to demonstrate the broader applicability of our argument.

These observations are consistent with our theoretical findings in Section 3, providing strong empirical evidence that residual branch scaling acts as an effective regularization technique. Additionally, experiments related to normalization appear to indicate a comparable operational mechanism.

## 5. Discussion

**Conclusion** This paper investigates how residual branch scaling influences generalization capabilities using NTK theory. By establishing the uniform convergence properties of wide ResNets, we demonstrate that over-parameterized ResNets achieve superior generalization when the residual branch scaling factor decays rapidly with depth, whereas the constant scaling factors lead to significant performance degradation. These theoretical findings provide crucial insights into the fundamental mechanisms underlying residual branch scaling techniques. Furthermore, our experimental results are highly consistent with the theoretical analysis, empirically validating the core conclusions of this paper.

**Limitations** Our theoretical analysis relies on standard simplifying assumptions prevalent in the NTK literature, such as the infinite-width limit, and focuses on fully connected networks with residual branches. These settings may not fully capture the complexity of architectures used in practice. While our results are derived within a regression framework, the spectral properties analyzed serve as a robust proxy for characterizing classification performance. However, extending these rigorous generalization guarantees to classification tasks and more complex scaling mechanisms remains an important direction for future research. Moreover, our experiments remain primarily limited to relatively small-scale models, datasets, and image classification tasks, and have not yet covered more modern, larger-scale, or more diverse application scenarios.

## Acknowledgements

All authors of this paper are either current students in or have benefited from academic interactions with the Department of Statistics and Data Science at Tsinghua University. We would like to express our sincere gratitude to the department, our advisors, and fellow students for their guidance, training, and valuable exchanges. We also thank members of the Huawei Large Model Data Technology Lab for their kind suggestions and support. Furthermore, we are grateful to the anonymous reviewers and the area chair for their insightful and constructive comments, which have greatly improved the quality of this work.

## Impact Statement

This paper presents work whose goal is to advance the field of Machine Learning. There are many potential societal consequences of our work, none which we feel must be specifically highlighted here.

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

# A. Additional Experiments

Modern deep learning models have achieved remarkable empirical success across a wide range of real-world applications, from complex reasoning (Jiang et al., 2025; Hu et al., 2026) and tool use (Schick et al., 2023; Xu et al., 2026; Wu et al., 2026) to human motion (Yu et al., 2024; Li et al., 2025; 2026; Jia et al., 2026), embodied intelligence (Driess et al., 2023; Zeng et al., 2025; 2026), and other real-world domains. Under the current large-scale training paradigm, models are often trained through multi-stage and increasingly complex pipelines on large, high-quality, and heterogeneous datasets (Zhou et al., 2023; Rao et al., 2025a;b; Yu et al., 2026), and are evaluated using downstream tasks or standard benchmarks (Cai et al., 2026; Jiang & Ferraro, 2026). For fundamental architectural mechanisms such as residual pathway design, their practical effects should ideally be examined under a broad range of conditions, including different application domains, data modalities, training scales, training procedures, and evaluation protocols.

However, given the theoretical focus of this paper, together with space and computational constraints, a comprehensive evaluation of residual branch scaling in modern large-scale deep learning settings is beyond the scope of this work. Nevertheless, we include additional experiments as a supplement to the empirical results in Section 4.2.2, thereby providing further empirical support for our theoretical findings. These supplementary experiments are still limited to image classification tasks and relatively moderate-scale architectures. Within this scope, we further examine the behavior of residual branch scaling on additional real-world datasets and across different network architectures.

## A.1. Additional Experiments on Different Datasets and Architectures

Due to space limitations, the experiments regarding real-world data and network architectures in the main text were restricted to ConvResNet on CIFAR-100, where the results were consistent with theoretical expectations (see Section 4.2.2). In this section, we further validate the practical applicability of our theory through supplementary experiments across a broader range of datasets and architectures.

Following the conventions in Section 4.2.2, the labels *base*, *scale*, and *norm* in subsequent figures denote three configurations: (i) the vanilla baseline without normalization or scaling, (ii) Residual Branch Scaling with $\gamma = 1$, and (iii) standard normalization (Batch Normalization for ConvResNets; Layer Normalization for Transformers).

**ConvResNets on Tiny-ImageNet Dataset** To validate the scalability of our conclusions across varying data scales, we conducted supplementary experiments using ResNet-34 on the Tiny-ImageNet dataset (Le & Yang, 2015), with results illustrated in Figure 5. The core observations are consistent with the findings in Section 4.2.2: residual scaling (and normalization) not only facilitates the training process but also significantly enhances generalization capability. Notably, these performance gains persist even when the model approaches saturation (with training accuracy near 100%), indicating that improved training dynamics are not the sole source of benefit. This further corroborates our core conclusion. Given the higher complexity of Tiny-ImageNet compared to CIFAR, the training schedule was extended to ensure full convergence.

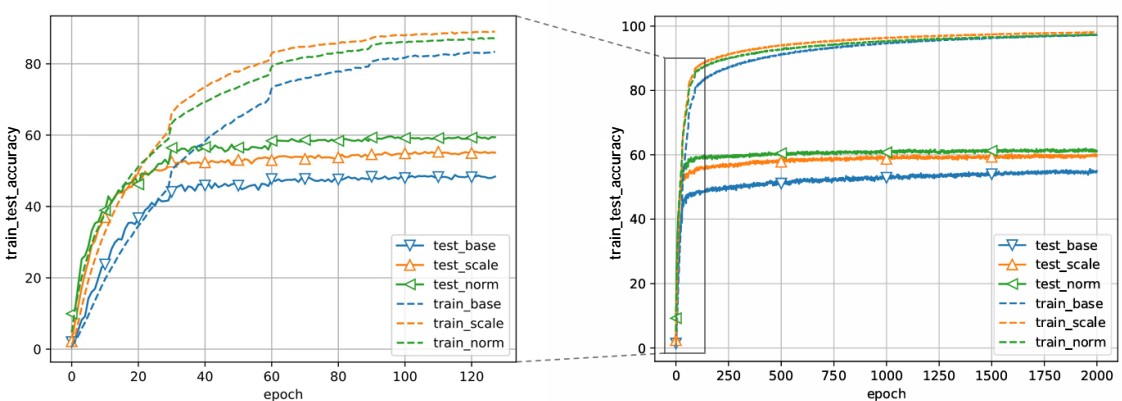

*Figure 5.* Training and test accuracy of ResNet-34 on Tiny-ImageNet under different strategies. The right panel displays the complete 2000-epoch training trajectory, demonstrating that residual branch scaling and normalization improve generalization performance even when training accuracy saturates near 100%. The left panel provides a zoomed-in view of the first 130 epochs, highlighting the differences in early convergence speeds.

**Transformers on CIFAR-100 Dataset**    Our theoretical framework was primarily established on Fully Connected ResNets, and prior experiments have validated the effectiveness of these conclusions within ConvResNets. However, given the growing dominance of Transformer architectures (Vaswani et al., 2017), it is crucial to investigate the performance of our proposed mechanism in such settings. To this end, we selected the Vision Transformer (ViT, Dosovitskiy et al., 2021) as a representative model and conducted supplementary experiments on the CIFAR-100 dataset.

As shown in Figure 6, despite the significant disparity in architectural design, the experimental results exhibit trends consistent with prior experiments shown in Figure 4 and Figure 5 (detailed descriptions are hence omitted for brevity), strongly corroborating our core theoretical predictions regarding the residual scaling mechanism. This suggests that our proposed method captures intrinsic nature, and its applicability broadly extends to diverse architectural backbones.

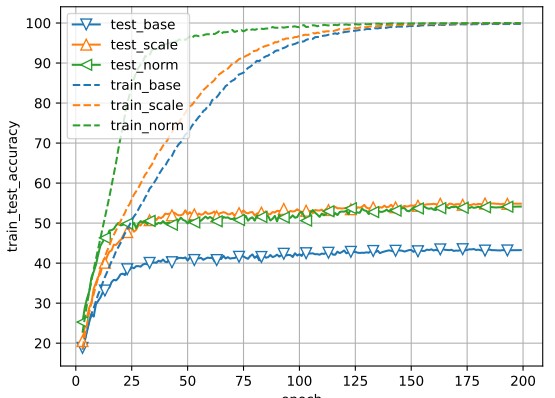

*Figure 6.* Training and test accuracy of Vision Transformers on the CIFAR-100 dataset with different strategies.

*Table 1.* **Matched Training Loss Accuracy on MNIST.** Matched Accuracy (Matched Acc.) reports the test accuracy when each method reaches a comparable training loss ($\approx 0.05$), reflecting the generalization ability of different methods, while Best Training Loss corresponds to the minimum training loss achieved within the same number of epochs, reflecting their optimization performance.

| **Method** | **Matched Acc. (%)** **(Generalization)** | **Best Train Loss ($\downarrow$)** **(Optimization)** |
|---|---|---|
| **Base** | **96.66** | **0.0315** |
| Fixup | 97.84 | 0.0085 |
| ReZero | 97.49 | 0.0046 |
| DeepNorm | 97.30 | 0.0135 |
| Proposed | 97.29 | 0.0047 |

## A.2. Matched Training Loss Evaluation

**Experimental Setting**    Although we have compared test accuracy under similar training accuracy in the main experiments, the corresponding training losses may still differ across methods even when their training accuracies are close. To address this issue and provide a more controlled comparison of generalization performance, we supplement the evaluation with a matched-training-loss protocol. Specifically, we record the test accuracy of each method when it reaches approximately the same optimization level, i.e., training loss $\approx 0.05$. At the same time, to assess the optimization performance of different methods, we also report the best training loss achieved within the same number of training epochs.

In addition, to better demonstrate that the mechanism analyzed in this paper broadly applies to different residual-branch scaling methods, we compare our simplified scaling rule with representative baselines, including Fixup (Zhang et al., 2019), ReZero (Bachlechner et al., 2021), and DeepNorm (Wang et al., 2024). It is worth noting that DeepNorm was originally developed for deep Transformers in a Post-Layer-Normalization-style setting and includes additional design choices beyond a simple scaling factor, such as a dedicated initialization scheme. In our experiments, we adopt only the DeepNorm-style residual scaling rule in order to isolate the effect of scaling itself.

**Experimental Results**    These experiments are conducted on the MNIST dataset, and the results are summarized in Table 1. As previously discussed, we observe that, when training losses are comparable across methods, various residual-branch scaling methods exhibit significantly higher test accuracy than the unscaled base model. This suggests that the improvement is not merely due to reaching a higher training accuracy, but is also reflected under a comparable optimization level. Furthermore, the inferior minimum training loss of the base model underscores the beneficial impact of branch scaling on the optimization process, providing additional empirical support for our main claims.

We emphasize that this paper does not aim to propose a new residual-scaling module or to compete with existing carefully engineered methods. Instead, our goal is to provide a partial mechanistic explanation for the generalization benefits observed in residual-branch scaling methods. Therefore, even if the simplified scaling rule studied in this paper, $\alpha = L^{-1}$, does not outperform strong baselines specifically designed for practical training, the similar behavior observed across these methods still provides meaningful evidence for the mechanism analyzed in this paper and is sufficient to support our main conclusion.

# B. Proof of Theorem 2.1

## B.1. Restatement of Settings and the Proposition

Theorem 2.1 discusses the uniform convergence of Empirical RNTK to RNTK. However, as noted in Section 2.1, we assumed that the initial output is zero. This assumption is not satisfied by the network structure and initialization method described in Section 2.1. For brevity, we omitted these details in the main text and address them in the appendix. Following related work (Li et al., 2024), we make minor adjustments to the network structure and initialization method. The reason we adopt zero initialization is to simplify the technical proof. Chen et al. (2024) has shown that, for FCNs, this simplification does not fundamentally affect the uniform convergence, and the same holds for ResNets.

Although these minor adjustments do not substantially affect the content we aim to present, we must restate some settings and notations here. Some of these differ from those agreed upon in the main text. These notations and conventions are limited to Section B, which provides a self-contained proof of Theorem 2.1 (of course, due to the changes in notation and settings, the proposition will also be restated as Theorem B.1). Thus, this does not affect the readability of other sections. In the absence of additional statements, the conventions in Section 1.4 still hold.

**Network Architecture and Initialization** We define a fully connected ResNet with $L$ hidden layers and width $m$ as follows (see Figure 7 for the structural diagram):

$$f^m(\boldsymbol{x}; \boldsymbol{\theta}) = \tfrac{\sqrt{2}}{2} \left[ f^{(1),m}\left(\boldsymbol{x}; \boldsymbol{\theta}^{(1)}\right) - f^{(2),m}\left(\boldsymbol{x}; \boldsymbol{\theta}^{(2)}\right) \right];$$

$$f^{(p),m}\left(\boldsymbol{x}; \boldsymbol{\theta}^{(p)}\right) = \boldsymbol{v}^{(p)\top} \boldsymbol{\alpha}^{(p,L)};$$

$$\boldsymbol{\alpha}^{(p,l)} = \boldsymbol{\alpha}^{(p,l-1)} + a\sqrt{\tfrac{1}{m}}\boldsymbol{V}^{(p,l)}\sigma\left(\sqrt{\tfrac{2}{m}}\boldsymbol{W}^{(p,l)}\boldsymbol{\alpha}^{(p,l-1)}\right); \tag{7}$$

$$\boldsymbol{\alpha}^{(p,0)} = \sqrt{\tfrac{1}{m}}\boldsymbol{A}^{(p)}\boldsymbol{x}, \qquad \boldsymbol{x} \in \mathcal{D} \subseteq \mathbb{R}^{d+1},$$

where $p = 1, 2$ and $l \in [L]$. The network parameters are given by $\boldsymbol{v}^{(p)} \in \mathbb{R}^m$, $\boldsymbol{V}^{(p,l)}, \boldsymbol{W}^{(p,l)} \in \mathbb{R}^{m \times m}$ and $\boldsymbol{A}^{(p)} \in \mathbb{R}^{m \times (d+1)}$. Additionally, the activation function is defined as $\sigma(x) := \max\{x, 0\}$, which corresponds to the ReLU function. The scaling factor $a$ (note that this notation differs from the main text) in the residual branch is a hyperparameter. It is set as $C \cdot L^\gamma$ for $\gamma \in [0, 1]$ and $C > 0$.

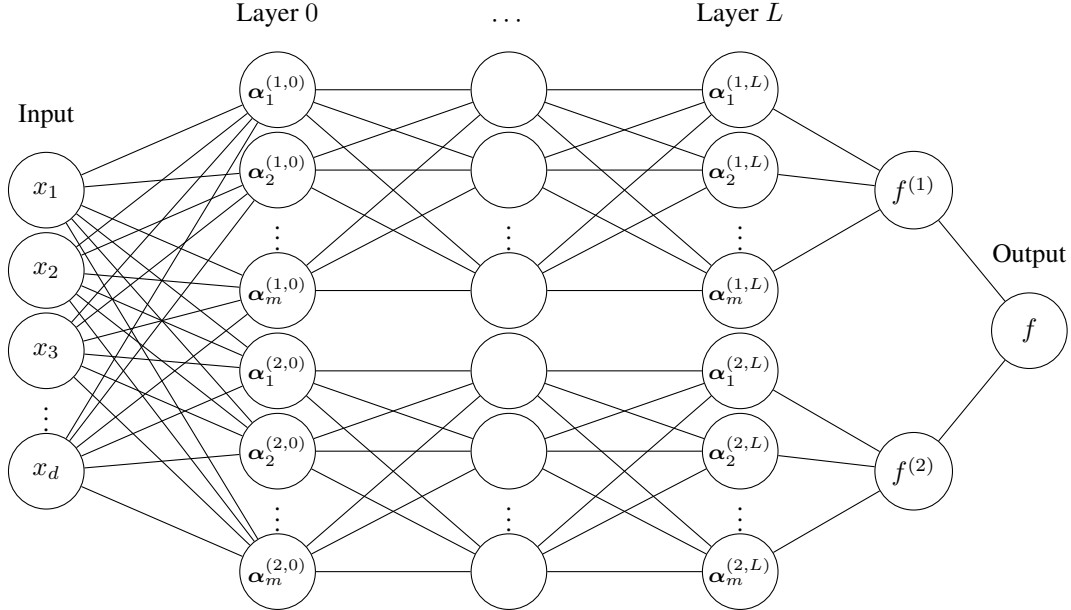

*Figure 7.* Special initialization

Notably, the network structure given in Equation (7) differs from that presented in the main text in two key aspects: besides

incorporating mirrored initialization, it also omits the bias term in the input layer. However, this omission is not essential. If we augment the input by appending a final component of 1, i.e., replacing $\boldsymbol{x}$ with $(\hat{\boldsymbol{x}}^\top, 1)^\top$ for $\hat{\boldsymbol{x}} \in \mathcal{X}$, and define $\boldsymbol{A}^{(p)} = [\boldsymbol{A}_0^{(p)}, \boldsymbol{b}]$, then we obtain

$$\boldsymbol{A}^{(p)}\boldsymbol{x} = \left(\boldsymbol{A}_0^{(p)}, \boldsymbol{b}\right)\begin{pmatrix}\hat{\boldsymbol{x}} \\ 1\end{pmatrix} = \boldsymbol{A}_0^{(p)}\hat{\boldsymbol{x}} + \boldsymbol{b},$$

which effectively restores the case where the input layer includes a bias term. This also justifies our choice of setting the input dimension to $d + 1$. Therefore, we only need to set $\mathcal{D} = \mathcal{X} \times \{1\}$. As assumed in the main text, we continue to assume that $\mathcal{X}$ is a compact set, ensuring that $\mathcal{D}$ is bounded below by 1 and above by some constant $C_{\mathcal{D}} > 0$.

To align with the mirrored architecture, we adopt the following mirrored initialization:

$$\text{for } i, j \in [m], \ k \in [d+1], \ l \in [L]: \quad \boldsymbol{A}_{i,k}^{(1)}, \ \boldsymbol{W}_{i,j}^{(1,l)}, \ \boldsymbol{V}_{i,j}^{(1,l)}, \ \boldsymbol{v}_i^{(1)} \overset{\text{i.i.d.}}{\sim} \mathcal{N}(0,1);$$

$$\boldsymbol{A}^{(1)} = \boldsymbol{A}^{(2)}, \quad \boldsymbol{W}^{(1,l)} = \boldsymbol{W}^{(2,l)}, \quad \boldsymbol{V}^{(1,l)} = \boldsymbol{V}^{(2,l)}, \quad \boldsymbol{v}^{(1)} = \boldsymbol{v}^{(2)}.$$

Thus, at initialization, we have $f^m(\boldsymbol{x}; \boldsymbol{\theta}) = 0$ for any $\boldsymbol{x} \in \mathcal{D}$.

**Training**   Let us consider the empirical square loss

$$\hat{\mathcal{L}}_n(f^m) = \frac{1}{2n}\sum_{i=1}^n \left(y_i - f^m(\boldsymbol{x}_i; \boldsymbol{\theta})\right)^2.$$

Denoting $\boldsymbol{\theta}_t$ as the parameter at time $t \geq 0$. Other time-varying quantities will also be indexed by $t$ when necessary. The network is trained by the gradient flow

$$\dot{\boldsymbol{\theta}}_t = -\nabla_{\boldsymbol{\theta}}\hat{\mathcal{L}}_n(f_t^m) = -\frac{1}{n}\nabla_{\boldsymbol{\theta}}f_t^m(\boldsymbol{X})(f_t^m(\boldsymbol{X}) - \boldsymbol{y})$$

where we emphasize that $\nabla_{\boldsymbol{\theta}}f_t^m(\boldsymbol{X})$ is a $2Lm^2 \times n$ matrix. We denote the resulting neural network function by $f_t(\boldsymbol{x}) = f(\boldsymbol{x}; \boldsymbol{\theta}_t)$.

**Empirical RNTK and RNTK**   With the above architecture, we have

$$\begin{aligned}
r_t^m(\boldsymbol{x}, \boldsymbol{x}') &:= \langle \nabla_{\boldsymbol{\theta}}f_t(\boldsymbol{x}; \boldsymbol{\theta}_t), \nabla_{\boldsymbol{\theta}}f_t(\boldsymbol{x}'; \boldsymbol{\theta}_t)\rangle \\
&= \frac{1}{2}\sum_{p=1}^2 \left\langle \nabla_{\boldsymbol{\theta}^{(p)}}\left(f_t^{(1),m}(\boldsymbol{x}) - f_t^{(2),m}(\boldsymbol{x})\right), \nabla_{\boldsymbol{\theta}^{(p)}}\left(f_t^{(1),m}(\boldsymbol{x}) - f_t^{(2),m}(\boldsymbol{x})\right)\right\rangle \\
&= \frac{1}{2}\sum_{p=1}^2 \left\langle \nabla_{\boldsymbol{\theta}^{(p)}}f_t^{(p),m}(\boldsymbol{x}), \nabla_{\boldsymbol{\theta}^{(p)}}f_t^{(p),m}(\boldsymbol{x}')\right\rangle = \frac{1}{2}\left(r_t^{(1),m}(\boldsymbol{x}, \boldsymbol{x}') + r_t^{(2),m}(\boldsymbol{x}, \boldsymbol{x}')\right),
\end{aligned}$$

where $r_t^{(p),m}(\boldsymbol{x}, \boldsymbol{x}')$ is the Empirical RNTK of $f_t^{(p),m}$, which is the vanilla neural network. Consequently, we have

$$r_0^{(1),m}(\boldsymbol{x}, \boldsymbol{x}') = r_0^{(2),m}(\boldsymbol{x}, \boldsymbol{x}') = r_0^m(\boldsymbol{x}, \boldsymbol{x}').$$

Thus, we only need to show the uniform convergence of one of the Empirical RNTKs $\{r_0^{(p),m}\}_{p=1,2}$ since another Empirical RNTK has the same uniform convergence.

Since the Empirical RNTK of the mirrored network is the average of the Empirical RNTKs of its two components, and the mirrored parts share the same RNTK, the mirrored architecture does not affect the expression of the RNTK. However, since our network structure adopts a bias-free input layer, the expression of the RNTK will be slightly modified (also denoted as $r$), as detailed below:

$$r(\boldsymbol{x}, \boldsymbol{x}') = a^2\|\boldsymbol{x}\|\|\boldsymbol{x}'\|\sum_{l=1}^L B_{l+1}(\tilde{\boldsymbol{x}}, \tilde{\boldsymbol{x}'})\left[(1+a^2)^{l-1}\kappa_1\left(\frac{K_{l-1}(\tilde{\boldsymbol{x}}, \tilde{\boldsymbol{x}'})}{(1+a^2)^{l-1}}\right) + K_{l-1}(\tilde{\boldsymbol{x}}, \tilde{\boldsymbol{x}'}) \cdot \kappa_0\left(\frac{K_{l-1}(\tilde{\boldsymbol{x}}, \tilde{\boldsymbol{x}'})}{(1+a^2)^{l-1}}\right)\right],$$

where $\tilde{\boldsymbol{x}} = \boldsymbol{x}/\|\boldsymbol{x}\|$, $\tilde{\boldsymbol{x}}' = \boldsymbol{x}'/\|\boldsymbol{x}'\|$, $K_0(\tilde{\boldsymbol{x}}, \tilde{\boldsymbol{x}}') = \tilde{\boldsymbol{x}}^\top \tilde{\boldsymbol{x}}'$, $B_{L+1}(\tilde{\boldsymbol{x}}, \tilde{\boldsymbol{x}}') = 1$ and

$$\kappa_0(u) = \frac{1}{\pi}(\pi - \arccos u), \qquad \kappa_1(u) = \frac{1}{\pi}\left(u(\pi - \arccos u) + \sqrt{1 - u^2}\right),$$

$$K_l(\tilde{\boldsymbol{x}}, \tilde{\boldsymbol{x}}') = K_{l-1}(\tilde{\boldsymbol{x}}, \tilde{\boldsymbol{x}}') + a^2(1 + a^2)^{l-1}\kappa_1\left(\frac{K_{l-1}(\tilde{\boldsymbol{x}}, \tilde{\boldsymbol{x}}')}{(1 + a^2)^{l-1}}\right),$$

$$B_l(\tilde{\boldsymbol{x}}, \tilde{\boldsymbol{x}}') = B_{l+1}(\tilde{\boldsymbol{x}}, \tilde{\boldsymbol{x}}')\left[1 + a^2\kappa_0\left(\frac{K_{l-1}(\tilde{\boldsymbol{x}}, \tilde{\boldsymbol{x}}')}{(1 + a^2)^{l-1}}\right)\right].$$

It is not difficult to see that when the input is replaced by $(\boldsymbol{x}^\top, 1)^\top$, the expression of the RNTK becomes consistent with Equation (5).

**The complete statement of Theorem 2.1 and Corollary 2.2**   Let us denote the minimal eigenvalue of the kernel matrix $r$ as $\lambda_0 = \lambda_{\min}(r(\boldsymbol{X}, \boldsymbol{X}))$. Lemma B.5 will show that $r$ is positive definite and thus $\lambda_0 > 0$ almost surely.

**Theorem B.1** (**The complete statement of Theorem 2.1**). *There exists a polynomial* $\text{poly}(\cdot) : \mathbb{R}^4 \to \mathbb{R}$, *such that for any given training data* $\{(\boldsymbol{x}_i, y_i), i \in [n]\}$ *and any* $\delta \in (0, 1)$, *when the width* $m \geq \text{poly}(n, \lambda_0^{-1}, \|\boldsymbol{y}\|_2, \log(1/\delta))$, *we have*

$$\sup_{t \geq 0} \sup_{\boldsymbol{x}, \boldsymbol{x}' \in \mathcal{D}} |r_t^m(\boldsymbol{x}, \boldsymbol{x}') - r(\boldsymbol{x}, \boldsymbol{x}')| \leq O\left(m^{-\frac{1}{12}}\sqrt{\log m}\right),$$

*with probability at least* $1 - \delta$.

**Corollary B.2** (**The complete statement of Corollary 2.2**). *There exists a polynomial* $\text{poly}(\cdot) : \mathbb{R}^5 \to \mathbb{R}$, *such that for any given training data* $\{(\boldsymbol{x}_i, y_i), i \in [n]\}$, *any* $\epsilon > 0$ *and any* $\delta \in (0, 1)$, *when the width* $m \geq \text{poly}(n, \lambda_0^{-1}, \|\boldsymbol{y}\|_2, \log(1/\delta), 1/\epsilon)$, *we have*

$$\sup_{t \geq 0}\left|\mathcal{E}(\hat{f}_t^{\text{ResNet}}) - \mathcal{E}(\hat{f}_t^{\text{RNTK}})\right| \leq \epsilon$$

holds with probability at least $1 - \delta$ with respect to the random initialization.

*Remark* B.3.   Applying the proof strategy in Proposition 3.2 and Theorem 3.1 of Lai et al. (2023a), we can utilize Theorem B.1 to complete the proof of Corollary B.2.

**Further notations**   For a vector $\boldsymbol{v} = (v_1, v_2, \cdots, v_m) \in \mathbb{R}^m$, we use $\|\boldsymbol{v}\|_2$ (or simply $\|\boldsymbol{v}\|$) to represent the Euclidean norm. Additionally, if we have a univariate function $f : \mathbb{R} \to \mathbb{R}$, we define $f(\boldsymbol{v}) = (f(v_1), f(v_2), \cdots, f(v_m)) \in \mathbb{R}^m$. We denote by $\|\boldsymbol{M}\|_2$ and $\|\boldsymbol{M}\|_F$ the spectral and Frobenius norm of a matrix $\boldsymbol{M}$ respectively. Also, we use $\|\cdot\|_0$ to represent the number of non-zero elements of a vector or matrix. For matrices $\boldsymbol{A} \in \mathbb{R}^{n_1 \times n_2}$ and $\boldsymbol{B} \in \mathbb{R}^{n_2 \times n_1}$, we define $\langle \boldsymbol{A}, \boldsymbol{B} \rangle = \text{Tr}(\boldsymbol{A}\boldsymbol{B}^\top)$. We remind that $\langle \boldsymbol{M}, \boldsymbol{M} \rangle = \|\boldsymbol{M}\|_F^2$ in this way.

To simplify the notation, except for $f^{(p),m}$ and $r^{(p),m}$, we sometimes omit the index $p$ on the parameters $\boldsymbol{W}^{(l)}, \boldsymbol{V}^{(l)}, \boldsymbol{A}, \boldsymbol{v}$ and their derived notations. If there is no additional statement, the conclusions hold for $p = 1, 2$.

For $l \in \{0, 1, \cdots, L\}$, denote $\boldsymbol{\delta}^{(l)}(\boldsymbol{x}) = \nabla_{\boldsymbol{\alpha}^{(l)}} f^{(p),m}(\boldsymbol{x}) = \nabla_{\boldsymbol{\alpha}^{(l)}} \boldsymbol{\alpha}^{(L)}(\boldsymbol{x})\boldsymbol{v}$. It is easy to check that

$$\boldsymbol{\delta}^{(l)}(\boldsymbol{x}) = \begin{cases} \nabla_{\boldsymbol{\alpha}^{(l)}} \boldsymbol{\alpha}^{(l+1)}(\boldsymbol{x})\boldsymbol{\delta}^{(l+1)}(\boldsymbol{x}), & l = 0, 1, \cdots, L-1; \\ \boldsymbol{v}, & l = L, \end{cases}$$

where

$$\nabla_{\boldsymbol{\alpha}^{(l-1)}} \boldsymbol{\alpha}^{(l)}(\boldsymbol{x}) = \left(\boldsymbol{I}_m + \frac{\sqrt{2}a}{m}\boldsymbol{V}^{(l)}\boldsymbol{D}^{(l)}(\boldsymbol{x})\boldsymbol{W}^{(l)}\right)^\top \qquad \text{for } l \in [L]$$

and

$$\boldsymbol{D}^{(l)}(\boldsymbol{x}) = \text{diag}\left(\sigma'\left(\sqrt{\frac{2}{m}}\boldsymbol{W}^{(l)}\boldsymbol{\alpha}^{(l-1)}(\boldsymbol{x})\right)\right) \qquad \text{for } l \in [L].$$

The gradient of $\boldsymbol{W}^{(l)}$ and $\boldsymbol{V}^{(l)}$ can be presented as follows:

$$\nabla_{\boldsymbol{W}^{(l)}} f^{(p),m}(\boldsymbol{x}) = \frac{\sqrt{2}a}{m} \boldsymbol{D}^{(l)}(\boldsymbol{x}) \boldsymbol{V}^{(l),T} \boldsymbol{\delta}^{(l)} \boldsymbol{\alpha}^{(l-1),T} = a\boldsymbol{\gamma}^{(l)}(\boldsymbol{x})\boldsymbol{\alpha}^{(l-1),T}(\boldsymbol{x});$$

$$\nabla_{\boldsymbol{V}^{(l)}} f^{(p),m}(\boldsymbol{x}) = \frac{\sqrt{2}a}{m} \boldsymbol{\delta}^{(l)}(\boldsymbol{x}) \left[ \sigma\left( \boldsymbol{W}^{(l)} \boldsymbol{\alpha}^{(l-1)}(\boldsymbol{x}) \right) \right]^{\top} = a\boldsymbol{\delta}^{(l)}(\boldsymbol{x})\boldsymbol{\eta}^{(l),T}(\boldsymbol{x}),$$

(8)

where

$$\boldsymbol{\gamma}^{(l)}(\boldsymbol{x}) = \frac{\sqrt{2}}{m} \boldsymbol{D}^{(l)}(\boldsymbol{x}) \boldsymbol{V}^{(l),T} \boldsymbol{\delta}^{(l)}(\boldsymbol{x}); \qquad \boldsymbol{\eta}^{(l)}(\boldsymbol{x}) = \frac{\sqrt{2}}{m} \boldsymbol{D}^{(l)}(\boldsymbol{x}) \boldsymbol{W}^{(l)} \boldsymbol{\alpha}^{(l-1)}(\boldsymbol{x}).$$

And the Empirical RNTK can be formulated as

$$r_t^{(p),m}(\boldsymbol{x}, \boldsymbol{x}') = \sum_{l=1}^{L} \Big( \left\langle \nabla_{\boldsymbol{W}^{(l)}} f_t^{(p),m}(\boldsymbol{x}), \nabla_{\boldsymbol{W}^{(l)}} f_t^{(p),m}(\boldsymbol{x}') \right\rangle$$

$$+ \left\langle \nabla_{\boldsymbol{V}^{(l)}} f_t^{(p),m}(\boldsymbol{x}), \nabla_{\boldsymbol{V}^{(l)}} f_t^{(p),m}(\boldsymbol{x}') \right\rangle \Big).$$

(9)

To shorten the notations, we denote

$$\boldsymbol{\delta}_{t,\boldsymbol{x}}^{(l)} = \boldsymbol{\delta}_t^{(l)}(\boldsymbol{x}), \quad \boldsymbol{\alpha}_{t,\boldsymbol{x}}^{(l)} = \boldsymbol{\alpha}_t^{(l)}(\boldsymbol{x}), \quad \boldsymbol{D}_{t,\boldsymbol{x}}^{(l)} = \boldsymbol{D}_t^{(l)}(\boldsymbol{x}), \quad \boldsymbol{\gamma}_{t,\boldsymbol{x}}^{(l)} = \boldsymbol{\gamma}_t^{(l)}(\boldsymbol{x}), \quad \boldsymbol{\eta}_{t,\boldsymbol{x}}^{(l)} = \boldsymbol{\eta}_t^{(l)}(\boldsymbol{x}).$$

$$\Delta\boldsymbol{\delta}_{\boldsymbol{xz}}^{(l)} := \boldsymbol{\delta}_{t,\boldsymbol{x}}^{(l)} - \boldsymbol{\delta}_{0,\boldsymbol{z}}^{(l)}, \quad \Delta\boldsymbol{\alpha}_{\boldsymbol{xz}}^{(l)} = \boldsymbol{\alpha}_{t,\boldsymbol{x}}^{(l)} - \boldsymbol{\alpha}_{0,\boldsymbol{z}}^{(l)}, \quad \Delta\boldsymbol{\gamma}_{\boldsymbol{xz}}^{(l)} := \boldsymbol{\gamma}_{t,\boldsymbol{x}}^{(l)} - \boldsymbol{\gamma}_{0,\boldsymbol{z}}^{(l)}, \quad \Delta\boldsymbol{\eta}_{\boldsymbol{xz}}^{(l)} := \boldsymbol{\eta}_{t,\boldsymbol{x}}^{(l)} - \boldsymbol{\eta}_{0,\boldsymbol{z}}^{(l)}$$

and

$$\boldsymbol{D}_{\boldsymbol{xz}}^{(l)'} = \boldsymbol{D}_{t,\boldsymbol{z}}^{(l)} - \boldsymbol{D}_{0,\boldsymbol{z}}^{(l)}, \quad \boldsymbol{g}_{l,\boldsymbol{xz}}' = \sqrt{\frac{2}{m}} \boldsymbol{W}_t^{(l)} \boldsymbol{\alpha}_{t,\boldsymbol{x}}^{(l-1)} - \sqrt{\frac{2}{m}} \boldsymbol{W}_0^{(l)} \boldsymbol{\alpha}_{0,\boldsymbol{z}}^{(l-1)},$$

$$\Delta\boldsymbol{V}^{(l)} = \boldsymbol{V}_t^{(l)} - \boldsymbol{V}_0^{(l)}; \quad \Delta\boldsymbol{W}^{(l)} = \boldsymbol{W}_t^{(l)} - \boldsymbol{W}_0^{(l)}.$$

## B.2. Positive Definiteness of RNTK

As noted in Caponnetto & De Vito (2007); Lin et al. (2020), studying the spectral properties of kernels is essential in classical kernel regression. Therefore, in this subsection, we review key spectral properties of the RNTK.

To ensure the uniform convergence of the neural network kernel to NTK in kernel regression (see Section 2.2), the positive definiteness of the kernel function is crucial. We first explicitly recall the following definition of positive definiteness to avoid potential confusion.

**Definition B.4.** A kernel function $K$ is positive definite (semi-definite) over domain $\mathcal{A}$ if for any positive integer $n$ and any $n$ different points $\boldsymbol{x}_1, \ldots, \boldsymbol{x}_n \in \mathcal{A}$, the smallest eigenvalue $\lambda_{\min}$ of the matrix $K(\boldsymbol{X}, \boldsymbol{X}) = (K(\boldsymbol{x}_i, \boldsymbol{x}_j))_{1 \le i,j \le n}$ is positive (non-negative).

The positive definiteness of FCNTK defined on the unit sphere was first proved by Jacot et al. (2018). Recently, Lai et al. (2023a) proved the positive definiteness of NTK for one-hidden-layer biased FCNs on $\mathbb{R}$, and Li et al. (2024) generalized it to multiple-hidden-layer FCNTK on $\mathbb{R}^d$.

As for RNTK, Belfer et al. (2024) showed that for inputs distributed uniformly on the hypersphere $\mathbb{S}^d$, the $k^{th}$ multiple eigenvalue of multiple-hidden-layer RNTK $r(\boldsymbol{x}, \boldsymbol{x}')$ is $\mu_k \asymp k^{-(d+1)}$, which implies $r(\boldsymbol{x}, \boldsymbol{x}')$ is positive definite on $\mathbb{S}^d$ when $L \ge 2$. For the input on $\mathcal{D} = \mathcal{X} \times \{1\} \subset \mathbb{R}^{d+1}$, the following lemma establishes the positive definiteness of $r(\boldsymbol{x}, \boldsymbol{x}')$.

**Lemma B.5.** $r(\boldsymbol{x}, \boldsymbol{x}')$ is positive definite on $\mathcal{D} = \mathcal{X} \times \{1\}$.

*Proof.* For $L = 1$, the RNTK $r$ is equal to the NTK of FCNs (Belfer et al., 2024), At this point, the lemma is already covered by Proposition 2.1 of Lai et al. (2023a). Thus, in the following, we consider only the case where $L \ge 2$.

For any positive integer $n$ and any $n$ different points $X_n = [\boldsymbol{x}_1, \ldots, \boldsymbol{x}_n]$, we have

$$r(X_n, X_n) = \text{diag}(\|\boldsymbol{x}_1\|, ..., \|\boldsymbol{x}_n\|) r(\tilde{X}_n, \tilde{X}_n) \text{diag}(\|\boldsymbol{x}_1\|, ..., \|\boldsymbol{x}_n\|),$$

where $\tilde{X}_n = (\boldsymbol{x}_1/\|\boldsymbol{x}_1\|_2, \ldots, \boldsymbol{x}_n/\|\boldsymbol{x}_n\|_2)$. Since Belfer et al. (2024) showed that $r(\tilde{X}_n, \tilde{X}_n)$ is positive definite. Thus, $r(X_n, X_n)$ is positive definite. $\square$

### B.3. Initialization

**Theorem B.6** (Theorem 4 of Huang et al. (2020))**.** *There exist some positive absolute constants $C_1 > 0$ and $C_2 \geq 1$, such that if $\varepsilon \in (0, 1/2)$, $\delta \in (0, 1)$ and $m \geq C_1 \varepsilon^{-4} \log(C_2/\delta)$, then for any fixed $z, z' \in \mathbb{S}^{d-1}$, with probability at least $1 - \delta$, we have*

$$\left| r_0^{(p),m}(z, z') - r(z, z') \right| \leq \varepsilon.$$

According to this result, we can get the following corollary.

**Corollary B.7.** There exist some positive absolute constants $C_1 > 0$ and $C_2 \geq 1$, such that if $\delta \in (0, 1)$ and $m \geq C_1 \left( \log(C_2/\delta) \right)^5$, then for any fixed $z, z' \in \mathcal{D}$, with probability at least $1 - \delta$, we have

$$\left| r_0^{(p),m}(z, z') - r(z, z') \right| = O\left( m^{-1/5} \right).$$

*Proof.* For any fixed $z, z' \in \mathcal{D}$, we have $z / \|z\|, z' / \|z'\| \in \mathbb{S}^d$. According to Theorem B.6 and let $\varepsilon = m^{-1/5}$, under the conditions we have previously established, we can obtain that

$$\left| r_0^{(p),m}\left( \frac{z}{\|z\|}, \frac{z'}{\|z'\|} \right) - r\left( \frac{z}{\|z\|}, \frac{z'}{\|z'\|} \right) \right| = O\left( m^{-1/5} \right).$$

By combining this result with the following relationships:

$$r_0^{(p),m}(z, z') = \|z\| \|z'\| r_0^{(p),m}\left( \frac{z}{\|z\|}, \frac{z'}{\|z'\|} \right); \qquad r(z, z') = \|z\| \|z'\| r\left( \frac{z}{\|z\|}, \frac{z'}{\|z'\|} \right)$$

and $\|z\|, \|z'\| \leq C_{\mathcal{D}}$, we can get the conclusion. $\qquad \square$

**Lemma B.8** (Corollary 5.35 in Vershynin (2011))**.** *Let $M$ be an $a \times b$ matrix whose entries are independent standard normal random variables. Then for every $t \geq 0$, with probability at least $1 - 2\exp\left( -t^2/2 \right)$, we have*

$$\|M\|_2 \leq \sqrt{a} + \sqrt{b} + t.$$

According to this lemma, we can directly get

**Corollary B.9** (Random matrix)**.** At initialization, there exists a positive absolute constant $C$, such that if $m \geq C$, then with probability at least $1 - \exp(-\Omega(m))$, the spectral norm of each matrix satisfies

$$\|A\|_2 = O\left( \sqrt{m} \right), \quad \left\| W_0^{(l)} \right\|_2 = O\left( \sqrt{m} \right), \quad \left\| V_0^{(l)} \right\|_2 = O\left( \sqrt{m} \right) \quad \text{and} \quad \|v\|_2 = O\left( \sqrt{m} \right) \quad \text{for } l \in [L].$$

**Lemma B.10.** *There exists a positive absolute constant $C$, such that if $m \geq C$, then with probability at least $1 - \exp(-\Omega(m))$ over the randomness of $A$, $W_0^{(l)}$, $V_0^{(l)}$ and $v$, for any $x \in \mathcal{D}$, we have*

$$\left\| \alpha_{0,x}^{(l)} \right\|_2 = O(1) \quad \text{and} \quad \left\| \delta_{0,x}^{(l)} \right\|_2 = O\left( \sqrt{m} \right) \qquad \text{for } l \in \{0, 1, \cdots, L\}.$$

*Proof.* We prove by induction that $\left\| \alpha_{0,x}^{(l)} \right\|_2 = O(1)$.

Base case: since $\|x\|_2 \leq C_{\mathcal{D}}$, with high probability over the random initialization of $A$, we have $\left\| \alpha_{0,x}^{(0)} \right\|_2 = \|Ax/\sqrt{m}\|_2 \leq C_{\mathcal{D}} \|A\|_2 / \sqrt{m} = O(1)$. Assume that $\left\| \alpha_{0,x}^{(l-1)} \right\|_2 = O(1)$, we can get

$$\left\| \alpha_{0,x}^{(l)} \right\|_2 = \left\| \alpha_{0,x}^{(l-1)} + a\frac{1}{\sqrt{m}} V_0^{(l)} \sigma\left( \sqrt{\frac{2}{m}} W_0^{(l)} \alpha_{0,x}^{(l-1)} \right) \right\|_2$$

$$\leq \left( 1 + \frac{\sqrt{2}a}{m} \left\| V_0^{(l)} \right\|_2 \left\| W_0^{(l)} \right\|_2 \right) \left\| \alpha_{0,x}^{(l-1)} \right\|_2 = O(1)$$

with high probability over the random initialization of $\boldsymbol{W}_0^{(l)}$ and $\boldsymbol{V}_0^{(l)}$. Therefore, for all $l \in \{0, 1, \cdots, L\}$, we have $\left\|\boldsymbol{\alpha}_{0,\boldsymbol{x}}^{(l)}\right\|_2 = O(1)$.

Next, we prove by induction that $\left\|\boldsymbol{\delta}_{0,\boldsymbol{x}}^{(l)}\right\|_2 = O(\sqrt{m})$. Recall that

$$\boldsymbol{\delta}_{0,\boldsymbol{x}}^{(l)} = \left( \boldsymbol{I}_m + \frac{\sqrt{2}a}{m} \boldsymbol{V}^{(l+1)} \boldsymbol{D}^{(l+1)} \boldsymbol{W}^{(l+1)} \right)^{\top} \boldsymbol{\delta}_{0,\boldsymbol{x}}^{(l+1)}.$$

Base case: $\left\|\boldsymbol{\delta}_{0,\boldsymbol{x}}^{(L)}\right\|_2 = \|\boldsymbol{v}\|_2 = O(\sqrt{m})$. Assume that $\left\|\boldsymbol{\delta}_{0,\boldsymbol{x}}^{(l+1)}\right\|_2 = O(\sqrt{m})$, we can get

$$\left\|\boldsymbol{\delta}_{0,\boldsymbol{x}}^{(l)}\right\|_2 = \left\| \left( \boldsymbol{I}_m + \frac{\sqrt{2}a}{m} \boldsymbol{W}_0^{(l+1)} \boldsymbol{D}_{0,\boldsymbol{x}}^{(l+1)} \boldsymbol{V}_0^{(l+1)} \right)^{\top} \boldsymbol{\delta}_{0,\boldsymbol{x}}^{(l+1)} \right\|_2$$

$$= \left( 1 + \frac{\sqrt{2}a}{m} \left\| \boldsymbol{W}_0^{(l+1)} \right\|_2 \left\| \boldsymbol{V}_0^{(l+1)} \right\|_2 \right) \left\| \boldsymbol{\delta}_{0,\boldsymbol{x}}^{(l+1)} \right\|_2 = O(\sqrt{m})$$

with probability at least $1 - \exp(-\Omega(m))$. Therefore, for all $l \in \{0, 1, \cdots, L\}$, we have $\left\|\boldsymbol{\delta}_{0,\boldsymbol{x}}^{(l)}\right\|_2 = O(\sqrt{m})$.

$\square$

**Lemma B.11.** *There exists a positive absolute constant $C$, such that for any fixed $\boldsymbol{z} \in \mathcal{D}$, with probability at least $1 - \exp\left(-\Omega(m^{5/6})\right)$ over the randomness of $\boldsymbol{A}$, $\boldsymbol{W}_0^{(l)}$ and $\boldsymbol{V}_0^{(l)}$, we have*

$$\left\|\boldsymbol{\alpha}_{0,\boldsymbol{z}}^{(l)}\right\|_2 = \Omega(1) \qquad \text{for } l \in \{0, 1, \cdots, L\}$$

*when $m$ is greater than the positive constant $C$.*

*Proof of Lemma B.11.* From the proof of Theorem 3 in Huang et al. (2020), we can know that as long as $m \geq \Omega((1 + a^2)^{12\ell}(1 + 1/4\pi)^{12L}/\epsilon^{12})$, with probability at least $1 - \exp\left(-\Omega(m^{5/6})\right)$, we have

$$\left| \|\boldsymbol{\alpha}_{0,\boldsymbol{z}}^{(l)}\|^2 - K_l(\boldsymbol{z}, \boldsymbol{z}) \right| \leq \frac{\epsilon(1 + a^2)^l}{(1 + 1/4\pi)^{L-l}}$$

for any sufficiently small $\epsilon > 0$. By triangle inequality, one has

$$\|\boldsymbol{\alpha}_{0,\boldsymbol{z}}^{(l)}\|^2 \geq (1 + a^2)^l - \frac{\epsilon(1 + a^2)^l}{(1 + 1/4\pi)^{L-l}} \geq \Omega(1).$$

$\square$

**Lemma B.12.** *There exists a positive absolute constant $C$, such that with probability at least $1 - \exp(-\Omega(m))$, for any $\boldsymbol{x} \in \mathcal{D}$, we have*

$$\left\| \nabla_{\boldsymbol{W}^{(l)}} f_0^{(p),m}(\boldsymbol{x}) \right\|_F = O(1), \quad \left\| \nabla_{\boldsymbol{V}^{(l)}} f_0^{(p),m}(\boldsymbol{x}) \right\|_F = O(1) \quad l \in \{1, \cdots, L\}$$

*when $m$ is greater than the positive constant $C$.*

*Proof of Lemma B.12.* First of all, because

$$\left\| \boldsymbol{a}\boldsymbol{b}^{\top} \right\|_F^2 = \text{Tr}\left( \boldsymbol{a}\boldsymbol{b}^{\top}\boldsymbol{b}\boldsymbol{a}^{\top} \right) = \text{Tr}\left( \boldsymbol{a}^{\top}\boldsymbol{a}\boldsymbol{b}^{\top}\boldsymbol{b} \right) = \|\boldsymbol{a}\|_2^2 \|\boldsymbol{b}\|_2^2$$

holds for two vectors $\boldsymbol{a}$ and $\boldsymbol{b}$, we can easily get

$$\left\| \nabla_{\boldsymbol{W}^{(l)}} f_0^{(p),m}(\boldsymbol{x}) \right\|_F = \frac{\sqrt{2}a}{m} \left\| \left( \boldsymbol{D}_{0,\boldsymbol{x}}^{(l)} \boldsymbol{V}_0^{(l),T} \boldsymbol{\delta}_{0,\boldsymbol{x}}^{(l)} \right) \boldsymbol{\alpha}_{0,\boldsymbol{x}}^{(l-1),T} \right\|_F$$

$$= \frac{\sqrt{2}a}{m} \left\| \boldsymbol{D}_{0,\boldsymbol{x}}^{(l)} \boldsymbol{V}_0^{(l),T} \boldsymbol{\delta}_{0,\boldsymbol{x}}^{(l)} \right\|_2 \left\| \boldsymbol{\alpha}_{0,\boldsymbol{x}}^{(l-1),T} \right\|_2 \leq \frac{\sqrt{2}a}{m} \left\| \boldsymbol{V}_0^{(l)} \right\|_2 \left\| \boldsymbol{\delta}_{0,\boldsymbol{x}}^{(l)} \right\|_2 \left\| \boldsymbol{\alpha}_{0,\boldsymbol{x}}^{(l-1)} \right\|_2$$

and

$$\left\| \nabla_{\boldsymbol{V}^{(l)}} f_0^{(p),m}(\boldsymbol{x}) \right\|_F = \frac{\sqrt{2}a}{m} \left\| \boldsymbol{\delta}_0^{(l)} \sigma^\top \left( \boldsymbol{W}_0^{(l)} \boldsymbol{\alpha}_0^{(l-1)} \right) \right\|_F$$

$$= \frac{\sqrt{2}a}{m} \left\| \boldsymbol{\delta}_0^{(l)} \right\|_2 \left\| \sigma \left( \boldsymbol{W}_0^{(l)} \boldsymbol{\alpha}_0^{(l-1)} \right) \right\|_2 \leq \frac{\sqrt{2}a}{m} \left\| \boldsymbol{\delta}_0^{(l)} \right\|_2 \left\| \boldsymbol{W}_0^{(l)} \right\|_2 \left\| \boldsymbol{\alpha}_0^{(l-1)} \right\|_2.$$

By Corollary B.9, we know that with probability at least $1 - \exp(-\Omega(m))$, $\left\| \boldsymbol{W}_0^{(l)} \right\|_2 = O(\sqrt{m})$ and $\left\| \boldsymbol{V}_0^{(l)} \right\|_2 = O(\sqrt{m})$ hold when $m$ is greater than some positive constant. Also, by Lemma B.10, we have shown that $\left\| \boldsymbol{\alpha}_{0,\boldsymbol{x}}^{(l-1)} \right\|_2 = O(1)$ and $\left\| \boldsymbol{\delta}_{0,\boldsymbol{x}}^{(l)} \right\|_2 = O(\sqrt{m})$ will hold under the similar conditions. Thus, we have

$$\left\| \nabla_{\boldsymbol{W}^{(l)}} f_0^{(p),m}(\boldsymbol{x}) \right\|_F = O(1), \quad \left\| \nabla_{\boldsymbol{V}^{(l)}} f_0^{(p),m}(\boldsymbol{x}) \right\|_F = O(1) \quad \text{for } l \in \{1, \cdots, L\}.$$

$\square$

## B.4. During training

**Lemma B.13** (Corollary 8.4 in Allen-Zhu et al. (2019)). *Suppose $\delta \in [0, O(1)]$ and $\boldsymbol{W}_0 \in \mathbb{R}^{m \times m}$ is a random matrix with entries drawn i.i.d from $\mathcal{N}(0, 1)$. With probability at least $1 - \exp\left(-\Omega(m\delta^{2/3})\right)$, the following holds. Fix any vector $\boldsymbol{h} \in \mathbb{R}^m$ with $\|\boldsymbol{h}\|_2 = \Theta(1)$ and for all $\boldsymbol{g}' \in \mathbb{R}^m$ with $\|\boldsymbol{g}'\|_2 \leq \delta$.*

*Let $\boldsymbol{D}'$ be the diagonal matrix where*

$$(\boldsymbol{D}')_{k,k} = \mathbf{1} \left\{ \left( \sqrt{\tfrac{2}{m}} \boldsymbol{W}_0 \boldsymbol{h} + \boldsymbol{g}' \right)_k > 0 \right\} - \mathbf{1} \left\{ \left( \sqrt{\tfrac{2}{m}} \boldsymbol{W}_0 \boldsymbol{h} \right)_k > 0 \right\}.$$

*Then, letting $\boldsymbol{u} = \boldsymbol{D}' \left( \sqrt{2/m} \boldsymbol{W}_0 \boldsymbol{h} + \boldsymbol{g}' \right)$, we have*

$$\|\boldsymbol{u}\|_0 \leq \|\boldsymbol{D}'\|_0 = O(m\delta^{2/3}), \quad \|\boldsymbol{u}\|_2 = O(\delta).$$

**Lemma B.14.** *Suppose each entry of matrix $\boldsymbol{W} \in \mathbb{R}^{a \times b}$ follows $W_{ij} \overset{\text{i.i.d.}}{\sim} \mathcal{N}(0, 1)$. Let $c = \max(a, b)$. If $s \geq 0$, then with probability at least $1 - \exp(-s \log c)$, the following holds:*

$$\forall \boldsymbol{u} \in \mathbb{R}^a, \boldsymbol{v} \in \mathbb{R}^b, s.t. \|\boldsymbol{u}\|_0, \|\boldsymbol{v}\|_0 \leq s, \quad \text{we have} \quad |\boldsymbol{u}^\top \boldsymbol{W} \boldsymbol{v}| \leq 9\sqrt{s \log c} \|\boldsymbol{u}\|_2 \|\boldsymbol{v}\|_2.$$

*Proof.* First of all, it is easy to see that when $s < 1$ or $c = 1$, the proposition is trivial. So we only need to consider the result under condition that $s \geq 1$ and $c \geq 2$.

Note that we aims to prove the inequality holds uniformly for all $\boldsymbol{u}, \boldsymbol{v}$ such that $\|\boldsymbol{u}\|_0, \|\boldsymbol{v}\|_0 \leq s$ at a high probability, we consider the non-zero entris of $\boldsymbol{u}, \boldsymbol{v}$ at first.

Let $A \subseteq [a]$ such that $|A| = \min\{a, \lfloor s \rfloor\}$, and let $U_A = \{\boldsymbol{u} \in \mathbb{R}^{|A|} : \forall i \notin A, u_i = 0\}$ be a set that contains vectors of which non-zero entries are only located in $A$. In the same way, let $B \subseteq [b]$ such that $|B| = \min\{b, \lfloor s \rfloor\}$, and let $V_B = \{\boldsymbol{v} \in \mathbb{R}^{|B|} : \forall j \notin B, v_j = 0\}$. Then we have

$$\boldsymbol{u}^\top \boldsymbol{W} \boldsymbol{v} = \sum_{i=1}^a \sum_{j=1}^b u_i W_{ij} v_j = \sum_{i \in A, j \in B} u_i W_{ij} v_j = \boldsymbol{u}_A^\top \boldsymbol{W}_{AB} \boldsymbol{v}_B,$$

in which $\boldsymbol{u}_A = (\boldsymbol{u}_i)_{i \in A}^\top, \boldsymbol{v}_B = (\boldsymbol{v}_j)_{j \in B}^\top, \boldsymbol{W}_{AB} = (\boldsymbol{W}_{ij})_{i \in A, j \in B}$. According to the definition of spectral norm, we know that

$$\left| \boldsymbol{u}^\top \boldsymbol{W} \boldsymbol{v} \right| = \left| \boldsymbol{u}_A^\top \boldsymbol{W}_{AB} \boldsymbol{v}_B \right| \le \|\boldsymbol{u}_A\|_2 \|\boldsymbol{W}_{AB}\|_2 \|\boldsymbol{v}_B\|_2.$$

Now we consider the spectral norm of $\boldsymbol{W}_{AB} \in \mathbb{R}^{|A| \times |B|}$. By Lemma B.8, we know when $t \ge \sqrt{\lfloor s \rfloor}$, with probability at least $1 - 2\exp(-t^2/2)$, we have $\|\boldsymbol{W}_{AB}\|_2 \le 3t$. Then we have

$$\forall \boldsymbol{u} \in U_A, \forall \boldsymbol{v} \in V_B, \quad |\boldsymbol{u}^\top \boldsymbol{W} \boldsymbol{v}| \le \|\boldsymbol{u}\|_2 \|\boldsymbol{W}_{AB}\|_2 \|\boldsymbol{v}\|_2 \le 3t \|\boldsymbol{u}\|_2 \|\boldsymbol{v}\|_2.$$

Now we consider all possible $A$ and $B$, or to say all possible location of non-zero entries. We know there are $\binom{a}{|A|}$ kinds of $A$ and $\binom{b}{|B|}$ kinds of $B$ in total. Therefore, with probability at least $1 - 2\binom{a}{|A|}\binom{b}{|B|}\exp(-t^2/2)$, the following proposition holds:

$$\forall \boldsymbol{u} \in \mathbb{R}^a, \forall \boldsymbol{v} \in \mathbb{R}^b, s.t. \|\boldsymbol{u}\|_0, \|\boldsymbol{v}\|_0 \le s, \quad \text{we have } |\boldsymbol{u}^\top \boldsymbol{W} \boldsymbol{v}| \le 3t \|\boldsymbol{u}\|_2 \|\boldsymbol{v}\|_2.$$

With the trivial inequality $\binom{n}{k} \le n^k$, we have a control for the probability above:

$$1 - 2\binom{a}{|A|}\binom{b}{|B|}\exp(-t^2/2) \ge 1 - 2a^{|A|}b^{|B|}\exp(-t^2/2) \ge 1 - 2a^{\lfloor s \rfloor}b^{\lfloor s \rfloor}\exp(-t^2/2)$$

$$\ge 1 - 2c^{2s}\exp(-t^2/2) = 1 - \exp(-(t^2/2 - 2s\log c - \log 2)).$$

Finally, let $t = \sqrt{8s\log c} \ge \sqrt{s}$, and then we get the expected result. $\qquad\square$

**Lemma B.15.** *Let* $\tau = O(\sqrt{m}/(\log m)^3)$ *and* $T \subseteq [0, \infty)$. *Suppose that* $\left\|\boldsymbol{W}_t^{(l)} - \boldsymbol{W}_0^{(l)}\right\|_F \le \tau$ *and* $\left\|\boldsymbol{V}_t^{(l)} - \boldsymbol{V}_0^{(l)}\right\|_F \le \tau$ *hold for all* $t \in T$ *and* $l \in [L]$. *Then there exists a positive absolute constant* $C$, *such that for any fixed* $\boldsymbol{z} \in \mathcal{D}$, *with probability at least* $1 - \exp(-\Omega(m^{2/3}\tau^{2/3}))$, *for all* $t \in T$ *and* $l \in [L]$, *we have*

- *i)* $\left\|\boldsymbol{g}'_{l,\boldsymbol{z}\boldsymbol{z}}\right\|_2 = O(\tau/\sqrt{m})$;

- *ii)* $\left\|\boldsymbol{D}_{\boldsymbol{z}\boldsymbol{z}}^{(l)\prime}\right\|_0 = O(m^{2/3}\tau^{2/3})$ *and* $\left\|\boldsymbol{D}_{\boldsymbol{z}\boldsymbol{z}}^{(l)\prime}\boldsymbol{W}_t^{(l)}\boldsymbol{\alpha}_{t,\boldsymbol{z}}^{(l-1)}\right\|_2 = O(\tau)$;

- *iii)* $\left\|\Delta\boldsymbol{\alpha}_{\boldsymbol{z}\boldsymbol{z}}^{(l)}\right\|_2 = O(\tau/\sqrt{m})$.

*when* $m$ *is greater than the positive constant* $C$.

*Proof of Lemma B.15.* We have shown that $\left\|\boldsymbol{W}_0^{(l)}\right\|_2 = O(\sqrt{m})$ and $\left\|\boldsymbol{V}_0^{(l)}\right\|_2 = O(\sqrt{m})$ hold with probability at least $1 - \exp(-\Omega(m))$. Combine with $\left\|\Delta\boldsymbol{W}^{(l)}\right\|_F \le \tau$ and $\left\|\Delta\boldsymbol{V}^{(l)}\right\|_F \le \tau$, we can get

$$\left\|\boldsymbol{W}_t^{(l)}\right\|_2 = O(\sqrt{m}) \qquad \text{and} \qquad \left\|\boldsymbol{V}_t^{(l)}\right\|_2 = O(\sqrt{m}).$$

Since $\boldsymbol{A}$ does not change during the training, we can easily check that $\left\|\Delta\boldsymbol{\alpha}_{\boldsymbol{z},\boldsymbol{z}}^{(0)}\right\|_2 = \|\boldsymbol{0}\|_2 = 0 = O(\tau/\sqrt{m})$, which means that $iii)$ holds for $l = 0$. Then it only needs to be proven that

$$iii) \text{ holds for } l = k \quad \implies \quad \text{Lemma holds for } l = k+1.$$

Now we assume that $iii)$ holds for $l = k \in \{0, 1, \cdots, L-1\}$, then with probability at least $1 - \exp(-\Omega(m))$, we have

$$\left\|\boldsymbol{\alpha}_{t,\boldsymbol{z}}^{(k)}\right\|_2 = \left\|\boldsymbol{\alpha}_{0,\boldsymbol{z}}^{(k)} + \Delta\boldsymbol{\alpha}_{\boldsymbol{z}\boldsymbol{z}}^{(k)}\right\|_2 \le \left\|\boldsymbol{\alpha}_0^{(k)}\right\|_2 + \left\|\Delta\boldsymbol{\alpha}_{\boldsymbol{z}\boldsymbol{z}}^{(k)}\right\|_2 = O(1) + O(\tau/\sqrt{m}) = O(1).$$

For $i)$, we can get

$$\boldsymbol{g}'_{k+1,\boldsymbol{z}\boldsymbol{z}} = \sqrt{\frac{2}{m}}\left(\boldsymbol{W}_t^{(k+1)}\boldsymbol{\alpha}_{t,\boldsymbol{z}}^{(k)} - \boldsymbol{W}_0^{(k+1)}\boldsymbol{\alpha}_{0,\boldsymbol{z}}^{(k)}\right) = \sqrt{\frac{2}{m}}\left(\Delta\boldsymbol{W}^{(k+1)}\boldsymbol{\alpha}_{t,\boldsymbol{z}}^{(k)} + \boldsymbol{W}_0^{(k+1)}\Delta\boldsymbol{\alpha}_{\boldsymbol{z}\boldsymbol{z}}^{(k)}\right),$$

which can lead to

$$\left\|\boldsymbol{g}'_{k+1,\boldsymbol{z}\boldsymbol{z}}\right\|_2 \le \sqrt{\frac{2}{m}} \left(\left\|\Delta\boldsymbol{W}^{(k+1)}\right\|_2 \left\|\boldsymbol{\alpha}^{(k)}_{t,\boldsymbol{z}}\right\|_2 + \left\|\boldsymbol{W}^{(k+1)}_0\right\|_2 \left\|\Delta\boldsymbol{\alpha}^{(k)}_{\boldsymbol{z}\boldsymbol{z}}\right\|_2\right)$$

$$\le \sqrt{\frac{2}{m}} \left(\tau \cdot O(1) + O(\sqrt{m})O(\tau/\sqrt{m})\right) \le O\left(\frac{\tau}{\sqrt{m}}\right).$$

Then by Lemma B.13 and taking $\boldsymbol{W}_0 = \boldsymbol{W}^{(k+1)}_0$, $\boldsymbol{h} = \boldsymbol{\alpha}^{(k)}_{0,\boldsymbol{z}}$, $\boldsymbol{g}' = \boldsymbol{g}'_{k+1}(\boldsymbol{z})$ and $\delta = \Theta(\tau/\sqrt{m}) \le O\left((\log m)^{-3}\right)$, we can get $ii)$ holds for $l = k+1$ with probability at least $1 - \exp\left(-\Omega(m^{2/3}\tau^{2/3})\right)$ since we have shown that $\|\boldsymbol{h}\|_2 = \left\|\boldsymbol{\alpha}^{(k)}_{0,\boldsymbol{z}}\right\|_2 = \Theta(1)$ in Lemma B.10 and Lemma B.11.

As for $iii)$, it is easy to check that

$$\Delta\boldsymbol{\alpha}^{(k+1)}_{\boldsymbol{z}\boldsymbol{z}} = \Delta\boldsymbol{\alpha}^{(k)}_{\boldsymbol{z}\boldsymbol{z}} + \frac{\sqrt{2}a}{m}\left[\boldsymbol{V}^{(k+1)}_t \boldsymbol{D}^{(k+1)\prime}_{\boldsymbol{z}\boldsymbol{z}} \boldsymbol{W}^{(k+1)}_t \boldsymbol{\alpha}^{(k)}_{t,\boldsymbol{z}} + \Delta\boldsymbol{V}^{(k+1)} \boldsymbol{D}^{(k+1)}_{0,\boldsymbol{z}} \boldsymbol{W}^{(k+1)}_t \boldsymbol{\alpha}^{(k)}_{t,\boldsymbol{z}}\right.$$
$$\left. + \boldsymbol{V}^{(k+1)}_0 \boldsymbol{D}^{(k+1)}_{0,\boldsymbol{z}}\left(\boldsymbol{W}^{(k+1)}_t \boldsymbol{\alpha}^{(k)}_{t,\boldsymbol{z}} - \boldsymbol{W}^{(k+1)}_0 \boldsymbol{\alpha}^{(k)}_{0,\boldsymbol{z}}\right)\right].$$

We have shown that, with probability at least $1 - \exp\left(-\Omega(m^{2/3}\tau^{2/3})\right)$, $i)$ $ii)$ hold for $l = k+1$, i.e.

$$\left\|\boldsymbol{D}^{(k+1)\prime}_{\boldsymbol{z}\boldsymbol{z}} \boldsymbol{W}^{(k+1)}_t \boldsymbol{\alpha}^{(k)}_{t,\boldsymbol{z}}\right\|_2 = O(\tau);$$

$$\left\|\boldsymbol{W}^{(k+1)}_t \boldsymbol{\alpha}^{(k)}_{t,\boldsymbol{z}} - \boldsymbol{W}^{(k+1)}_0 \boldsymbol{\alpha}^{(k)}_{0,\boldsymbol{z}}\right\|_2 = \sqrt{\frac{m}{2}} \left\|\boldsymbol{g}'_{k+1}\right\|_2 = O(\tau),$$

which can lead to $\left\|\Delta\boldsymbol{\alpha}^{(k+1)}_{\boldsymbol{z}\boldsymbol{z}}\right\|_2 = O(\tau/\sqrt{m})$.

Thus, we finish the proof.

$\square$

**Lemma B.16.** *Let* $\tau = O\left(\sqrt{m}/(\log m)^3\right)$ *and* $T \subseteq [0, \infty)$. *Suppose that* $\left\|\boldsymbol{W}^{(l)}_t - \boldsymbol{W}^{(l)}_0\right\|_F \le \tau$ *and* $\left\|\boldsymbol{V}^{(l)}_t - \boldsymbol{V}^{(l)}_0\right\|_F \le \tau$ *hold for all* $t \in T$ *and* $l \in [L]$. *Then there exists a positive absolute constant* $C$, *such that for any fixed* $\boldsymbol{z} \in \mathcal{D}$, *with probability at least* $1 - \exp\left(-\Omega(m^{2/3}\tau^{2/3})\right)$, *for all* $t \in T$ *and* $l \in [L]$, *we have*

$$\left\|\Delta\boldsymbol{\delta}^{(l)}_{\boldsymbol{z}\boldsymbol{z}}\right\|_2 = O\left(m^{1/3}\tau^{1/3}\sqrt{\log m}\right),$$

*when* $m$ *is greater than the positive constant* $C$.

*Proof of Lemma B.16.* We inductively prove this lemma.

Base case: $\left\|\Delta\boldsymbol{\delta}^{(L)}_{\boldsymbol{z}\boldsymbol{z}}\right\|_2 = 0$ since $\boldsymbol{v}$ is fixed during the training process.

Assume that this lemma holds for $l + 1$, then with probability at least $1 - \exp(-\Omega(m))$, we have

$$\left\|\boldsymbol{\delta}^{(l+1)}_{t,\boldsymbol{z}}\right\|_2 = \left\|\Delta\boldsymbol{\delta}^{(l+1)}_{\boldsymbol{z}\boldsymbol{z}} + \boldsymbol{\delta}^{(l+1)}_{0,\boldsymbol{z}}\right\|_2 \le \left\|\Delta\boldsymbol{\delta}^{(l+1)}_{\boldsymbol{z}\boldsymbol{z}}\right\|_2 + \left\|\boldsymbol{\delta}^{(l+1)}_{0,\boldsymbol{z}}\right\|_2 \le O(\sqrt{m})$$

because of Lemma B.10. Moreover, it is easy to check that

$$\Delta\boldsymbol{\delta}^{(l)}_{\boldsymbol{z}\boldsymbol{z}} = \Delta\boldsymbol{\delta}^{(l+1)}_{\boldsymbol{z}\boldsymbol{z}} + \frac{\sqrt{2}a}{m}\left[\boldsymbol{W}^{(l+1),T}_0 \boldsymbol{D}^{(l+1)\prime}_{\boldsymbol{z}\boldsymbol{z}} \boldsymbol{V}^{(l+1),T}_0 \boldsymbol{\delta}^{(l+1)}_{0,\boldsymbol{z}} + \Delta\boldsymbol{W}^{(l+1),T} \boldsymbol{D}^{(l+1)}_{t,\boldsymbol{z}} \boldsymbol{V}^{(l+1),T}_0 \boldsymbol{\delta}^{(l+1)}_{0,\boldsymbol{z}}\right.$$
$$\left. + \boldsymbol{W}^{(l+1),T}_t \boldsymbol{D}^{(l+1)}_{t,\boldsymbol{z}} \Delta\boldsymbol{V}^{(l+1),T} \boldsymbol{\delta}^{(l+1)}_{0,\boldsymbol{z}} + \boldsymbol{W}^{(l+1),T}_t \boldsymbol{D}^{(l+1)}_{t,\boldsymbol{z}} \boldsymbol{V}^{(l+1),T}_t \Delta\boldsymbol{\delta}^{(l+1)}_{\boldsymbol{z}\boldsymbol{z}}\right].$$

Let us denote the four terms within the square brackets, excluding the factor '$\sqrt{2}a/m$' outside the brackets, as $\boldsymbol{u}_1$ to $\boldsymbol{u}_4$ respectively. First of all, it is easy to check that, with probability at least $1 - \exp(-\Omega(m))$, we have

$$\|\boldsymbol{u}_2\|_2 \leq \tau \cdot 1 \cdot O(\sqrt{m}) \cdot O(\sqrt{m}) = O(\tau \cdot m) \leq O\left(m \cdot m^{1/3}\tau^{1/3}\sqrt{\log m}\right);$$

$$\|\boldsymbol{u}_3\|_2 \leq O(\sqrt{m}) \cdot 1 \cdot \tau \cdot O(\sqrt{m}) = O(\tau \cdot m) \leq O\left(m \cdot m^{1/3}\tau^{1/3}\sqrt{\log m}\right);$$

$$\|\boldsymbol{u}_4\|_2 \leq O(\sqrt{m}) \cdot 1 \cdot O(\sqrt{m}) \cdot O\left(m^{1/3}\tau^{1/3}\sqrt{\log m}\right) = O\left(m \cdot m^{1/3}\tau^{1/3}\sqrt{\log m}\right).$$

As for $\boldsymbol{u}_1$, if $\boldsymbol{\delta}_{0,\boldsymbol{z}}^{(l+1)} = \boldsymbol{0}$ or $\left\|\boldsymbol{D}_{\boldsymbol{zz}}^{(l+1)\prime}\right\|_0 = 0$, we have $\|\boldsymbol{u}_1\|_2 = 0$. Therefore, we consider the case where $\boldsymbol{\delta}_{0,\boldsymbol{z}}^{(l+1)} \neq \boldsymbol{0}$ and $\left\|\boldsymbol{D}_{\boldsymbol{zz}}^{(l+1)\prime}\right\|_0 \geq 1$. Denote $\tilde{\boldsymbol{\delta}} = \boldsymbol{\delta}_{0,\boldsymbol{z}}^{(l+1)} / \left\|\boldsymbol{\delta}_{0,\boldsymbol{z}}^{(l+1)}\right\|_2$ for $\boldsymbol{\delta}_{0,\boldsymbol{z}}^{(l+1)} \in \mathbb{R}^m \backslash \{\boldsymbol{0}\}$, we can get

$$\|\boldsymbol{u}_1\|_2 \leq \left\|\boldsymbol{W}_0^{(l+1)}\right\|_2 \left\|\boldsymbol{D}_{\boldsymbol{zz}}^{(l+1)\prime}\boldsymbol{V}_0^{(l+1)}\tilde{\boldsymbol{\delta}}\right\|_2 \left\|\boldsymbol{\delta}_{0,\boldsymbol{z}}^{(l+1)}\right\|_2 \leq O(m) \left\|\boldsymbol{D}_{\boldsymbol{zz}}^{(l+1)\prime}\boldsymbol{V}_0^{(l+1)}\tilde{\boldsymbol{\delta}}\right\|_2.$$

Using the randomness of $\boldsymbol{V}_0^{(l+1)}$, for any fixed $\tilde{\boldsymbol{\delta}}$, we have $\boldsymbol{V}_0^{(l+1)}\tilde{\boldsymbol{\delta}} \sim \mathcal{N}(\boldsymbol{0}, \boldsymbol{I}_m)$. Thus, by Lemma B.14 and taking $s = \Theta(m^{2/3}\tau^{2/3})$, with probability at least $1 - \exp\left(-\Omega(m^{2/3}\tau^{2/3})\right)$, we can get

$$\left\|\boldsymbol{D}_{\boldsymbol{zz}}^{(l+1)\prime}\boldsymbol{V}_0^{(l+1)}\tilde{\boldsymbol{\delta}}\right\|_2 = \sup_{\boldsymbol{u} \in \mathbb{S}^{m-1}} \left\|\boldsymbol{u}^\top \boldsymbol{D}_{\boldsymbol{zz}}^{(l+1)\prime}\boldsymbol{V}_0^{(l+1)}\tilde{\boldsymbol{\delta}}\right\|_2 = \sup_{\boldsymbol{u} \in \mathbb{S}^{m-1}} \left\|\left(\boldsymbol{D}_{\boldsymbol{zz}}^{(l+1)\prime}\boldsymbol{u}\right)^\top \boldsymbol{V}_0^{(l+1)}\tilde{\boldsymbol{\delta}} \cdot 1\right\|_2$$
$$\leq O\left(\sqrt{m^{2/3}\tau^{2/3}\log m}\right) = O\left(m^{1/3}\tau^{1/3}\sqrt{\log m}\right),$$

because of $\left\|\boldsymbol{D}_{\boldsymbol{zz}}^{(l+1)\prime}\boldsymbol{u}\right\|_0 \leq \left\|\boldsymbol{D}_{\boldsymbol{zz}}^{(l+1)\prime}\right\|_0 \leq \Theta(m^{2/3}\tau^{2/3})$ and $\|1\|_0 = 1 \leq \left\|\boldsymbol{D}_{\boldsymbol{zz}}^{(l+1)\prime}\right\|_0$.

Combining the above discussions, we can conclude that $\left\|\Delta\boldsymbol{\delta}_{\boldsymbol{zz}}^{(l)}\right\|_2 \leq O\left(m^{1/3}\tau^{1/3}\sqrt{\log m}\right)$.

$\square$

**Lemma B.17.** *Fix $l \in [L]$ and let $\tau = O\left(\sqrt{m}/(\log m)^3\right)$, $T \subseteq [0, \infty)$. Suppose that $\left\|\boldsymbol{W}_t^{(l)} - \boldsymbol{W}_0^{(l)}\right\|_F \leq \tau$ and $\left\|\boldsymbol{V}_t^{(l)} - \boldsymbol{V}_0^{(l)}\right\|_F \leq \tau$ hold for all $t \in T$, then there exists a positive absolute constant $C$, such that for any fixed $\boldsymbol{z} \in \mathcal{D}$, with probability at least $1 - \exp\left(-\Omega(m^{2/3}\tau^{2/3})\right)$ over the randomness of $\boldsymbol{W}_0^{(l)}$ and $\boldsymbol{V}_0^{(l)}$, for all $t \in T$, we have*

$$\left\|\Delta\boldsymbol{\gamma}_{\boldsymbol{zz}}^{(l)}\right\|_2 = O\left(m^{-1/6}\tau^{1/3}\sqrt{\log m}\right), \quad \left\|\Delta\boldsymbol{\eta}_{\boldsymbol{zz}}^{(l)}\right\|_2 = O\left(\frac{\tau}{m}\right);$$
$$\left\|\boldsymbol{\gamma}_{t,\boldsymbol{z}}^{(l)}\right\|_2 = O(1), \quad \left\|\boldsymbol{\eta}_{t,\boldsymbol{z}}^{(l)}\right\|_2 = O(1/\sqrt{m})$$

*when $m$ is greater than the positive constant $C$.*

*Proof of Lemma B.17.* First of all, we have

$$\Delta\boldsymbol{\gamma}_{\boldsymbol{zz}}^{(l)} = \frac{\sqrt{2}}{m}\left(\boldsymbol{D}_{t,\boldsymbol{z}}^{(l)}\boldsymbol{V}_t^{(l),T}\boldsymbol{\delta}_{t,\boldsymbol{z}}^{(l)} - \boldsymbol{D}_{0,\boldsymbol{z}}^{(l)}\boldsymbol{V}_0^{(l),T}\boldsymbol{\delta}_{0,\boldsymbol{z}}^{(l)}\right)$$
$$= \frac{\sqrt{2}}{m}\left(\boldsymbol{D}_{\boldsymbol{zz}}^{(l)\prime}\boldsymbol{V}_0^{(l),T}\boldsymbol{\delta}_{0,\boldsymbol{z}}^{(l)} + \boldsymbol{D}_{t,\boldsymbol{z}}^{(l)}\Delta\boldsymbol{V}^{(l),T}\boldsymbol{\delta}_{0,\boldsymbol{z}}^{(l)} + \boldsymbol{D}_{t,\boldsymbol{z}}^{(l)}\boldsymbol{V}_t^{(l),T}\Delta\boldsymbol{\delta}_{\boldsymbol{zz}}^{(l)}\right).$$

Using the similar proof technique as the previous lemma, we can establish that with probability at least $1 - \exp\left(-\Omega(m^{2/3}\tau^{2/3})\right)$, we have

$$\left\|\boldsymbol{D}_{\boldsymbol{zz}}^{(l)\prime}\boldsymbol{V}_0^{(l),T}\boldsymbol{\delta}_{0,\boldsymbol{z}}^{(l)}\right\|_2 = O\left(m^{1/3}\tau^{1/3}\sqrt{\log m}\right).$$

According to Corollary B.9, Lemma B.10 and Lemma B.16 we can get

$$\left\|\boldsymbol{D}_{t,\boldsymbol{z}}^{(l)}\Delta\boldsymbol{V}^{(l),T}\boldsymbol{\delta}_{0,\boldsymbol{z}}^{(l)}\right\|_2 = O(\tau\sqrt{m}); \quad \left\|\boldsymbol{D}_{t,\boldsymbol{z}}^{(l)}\boldsymbol{V}_t^{(l),T}\Delta\boldsymbol{\delta}_{\boldsymbol{zz}}^{(l)}\right\|_2 = O\left(m^{5/6}\tau^{1/3}\sqrt{\log m}\right).$$

Thus, we can get $\left\|\Delta\gamma_{zz}^{(l)}\right\|_2 = O\big(m^{-1/6}\tau^{1/3}\sqrt{\log m}\big)$.

As for $\left\|\Delta\eta_{zz}^{(l)}\right\|_2$, we can similarly get

$$\left\|\Delta\eta_{zz}^{(l)}\right\|_2 = \frac{\sqrt{2}}{m}\left\|\boldsymbol{D}_{zz}^{(l)\prime}\boldsymbol{W}_t^{(l)}\boldsymbol{\alpha}_{t,z}^{(l-1)} + \boldsymbol{D}_{0,z}^{(l)}\Delta\boldsymbol{W}^{(l)}\boldsymbol{\alpha}_{t,z}^{(l-1)} + \boldsymbol{D}_{0,z}^{(l)}\boldsymbol{W}_0^{(l)}\Delta\boldsymbol{\alpha}_{zz}^{(l-1)}\right\|_2$$
$$\leq \frac{\sqrt{2}}{m}\big(O(\tau) + O(\tau) + O(\tau)\big) = O\Big(\frac{\tau}{m}\Big)$$

according to Corollary B.9, Lemma B.10 and Lemma B.15.

With the above results, we can easily get

$$\left\|\gamma_{0,z}^{(l)}\right\|_2 = \frac{\sqrt{2}}{m}\left\|\boldsymbol{D}_{0,z}^{(l)}\boldsymbol{V}_0^{(l),T}\boldsymbol{\delta}_{0,z}^{(l)}\right\|_2 = O(1), \quad \left\|\eta_{0,z}^{(l)}\right\|_2 = \frac{\sqrt{2}}{m}\left\|\boldsymbol{D}_{0,z}^{(l)}\boldsymbol{W}_0^{(l)}\boldsymbol{\alpha}_{0,z}^{(l-1)}\right\|_2 = O\Big(\frac{1}{\sqrt{m}}\Big),$$
$$\left\|\gamma_{t,z}^{(l)}\right\|_2 \leq \left\|\gamma_{0,z}^{(l)}\right\|_2 + \left\|\Delta\gamma_{zz}^{(l)}\right\|_2 = O(1), \quad \left\|\eta_{t,z}^{(l)}\right\|_2 \leq \left\|\eta_0^{(l)}\right\|_2 + \left\|\Delta\eta_{zz}^{(l)}\right\|_2 = O\Big(\frac{1}{\sqrt{m}}\Big)$$

since $\tau = O\big(\sqrt{m}/(\log m)^3\big)$.

$\square$

**Lemma B.18.** *Let $\tau = O\big(\sqrt{m}/(\log m)^3\big)$ and $T \subseteq [0, \infty)$. Suppose that $\left\|\boldsymbol{W}_t^{(l)} - \boldsymbol{W}_0^{(l)}\right\|_F \leq \tau$ and $\left\|\boldsymbol{V}_t^{(l)} - \boldsymbol{V}_0^{(l)}\right\|_F \leq \tau$ hold for all $t \in T$ and $l \in [L]$. Then there exists a positive absolute constant $C$, such that for any fixed $\boldsymbol{z} \in \mathcal{D}$, with probability at least $1 - \exp\big(-\Omega(m^{2/3}\tau^{2/3})\big)$, for all $l \in [L]$, we have*

$$\sup_{t\in T}\left\|\nabla_{\boldsymbol{W}^{(l)}} f_t^{(p),m}(\boldsymbol{z}) - \nabla_{\boldsymbol{W}^{(l)}} f_0^{(p),m}(\boldsymbol{z})\right\|_F = O\Big(m^{-1/6}\tau^{1/3}\sqrt{\log m}\Big);$$
$$\sup_{t\in T}\left\|\nabla_{\boldsymbol{V}^{(l)}} f_t^{(p),m}(\boldsymbol{z}) - \nabla_{\boldsymbol{V}^{(l)}} f_0^{(p),m}(\boldsymbol{z})\right\|_F = O\Big(m^{-1/6}\tau^{1/3}\sqrt{\log m}\Big),$$

*when $m$ is greater than the positive constant $C$.*

*Proof of Lemma B.18.* According to Equation (8), we have

$$\left\|\nabla_{\boldsymbol{W}^{(l)}} f_t^{(p),m}(\boldsymbol{z}) - \nabla_{\boldsymbol{W}^{(l)}} f_0^{(p),m}(\boldsymbol{z})\right\|_F = \left\|a\gamma_{t,z}^{(l)}\boldsymbol{\alpha}_{t,z}^{(l-1),T} - a\gamma_{0,z}^{(l)}\boldsymbol{\alpha}_{0,z}^{(l-1),T}\right\|_F$$
$$= a\left\|\gamma_{t,z}^{(l)}\Delta\boldsymbol{\alpha}_{zz}^{(l-1),T} + \Delta\gamma_{zz}^{(l)}\boldsymbol{\alpha}_{0,z}^{(l-1),T}\right\|_F \leq a\left\|\gamma_{t,z}^{(l)}\Delta\boldsymbol{\alpha}_{zz}^{(l-1),T}\right\|_F + a\left\|\Delta\gamma_{zz}^{(l)}\boldsymbol{\alpha}_{0,z}^{(l-1),T}\right\|_F$$
$$= a\left\|\gamma_{t,z}^{(l)}\right\|_2\left\|\Delta\boldsymbol{\alpha}_{zz}^{(l-1)}\right\|_2 + a\left\|\Delta\gamma_{zz}^{(l)}\right\|_2\left\|\boldsymbol{\alpha}_{0,z}^{(l-1)}\right\|_2 \leq O\Big(m^{-1/6}\tau^{1/3}\sqrt{\log m}\Big)$$

according to Lemmas B.17, B.15 *iii*) and B.10. Similarly, we can also get

$$\left\|\nabla_{\boldsymbol{V}^{(l)}} f_t^{(p),m}(\boldsymbol{z}) - \nabla_{\boldsymbol{V}^{(l)}} f_0^{(p),m}(\boldsymbol{z})\right\|_F = \left\|a\boldsymbol{\delta}_{t,z}^{(l)}\eta_{t,z}^{(l),T} - a\boldsymbol{\delta}_{0,z}^{(l)}\eta_{0,z}^{(l),T}\right\|_F$$
$$\leq a\left\|\eta_{t,z}^{(l)}\right\|_2\left\|\Delta\boldsymbol{\delta}_{zz}^{(l)}\right\|_2 + a\left\|\Delta\eta_{zz}^{(l)}\right\|_2\left\|\boldsymbol{\delta}_{0,z}^{(l)}\right\|_2 \leq O\Big(m^{-1/6}\tau^{1/3}\sqrt{\log m}\Big)$$

according to Lemmas B.17, B.16 and B.10.

Thus, we finish the proof.

$\square$

**Proposition B.19.** *There exists a polynomial* $\mathrm{poly}(\cdot) : \mathbb{R}^4 \to \mathbb{R}$, *such that for any given training data* $\{(\boldsymbol{x}_i, y_i), i \in [n]\}$, *any $\delta \in (0, 1)$ and any fixed $\boldsymbol{z}, \boldsymbol{z}' \in \mathcal{D}$, when the width $m \geq \mathrm{poly}(n, \lambda_0^{-1}, \|\boldsymbol{y}\|_2, \log(1/\delta))$, with probability at least $1 - \delta$, we have*

$$\sup_{t\geq 0}|r_t^m(\boldsymbol{z}, \boldsymbol{z}') - r(\boldsymbol{z}, \boldsymbol{z}')| = O\Big(m^{-\frac{1}{12}}\sqrt{\log m}\Big).$$

*Proof.* This proposition can be deduced in conjunction with Corollary B.7, Lemma B.20, and the forthcoming Lemma B.23 to be proven in the next subsection. $\qquad\square$

**Lemma B.20.** *Fix* $z, z' \in \mathcal{D}$ *and let* $\delta \in (0,1)$, $T \subseteq [0, \infty)$. *Suppose that* $\left\| W_t^{(l)} - W_0^{(l)} \right\|_F = O(m^{1/4})$ *and* $\left\| V_t^{(l)} - V_0^{(l)} \right\|_F = O(m^{1/4})$ *hold for all* $t \in T$ *and* $l \in [L]$. *Then there exist some positive absolute constants* $C_1 > 0$ *and* $C_2 \geq 1$, *such that with probability at least* $1 - \delta$, *we have*

$$\sup_{t \in T} \left| r_t^{(p),m}(z, z') - r_0^{(p),m}(z, z') \right| = O\left( m^{-\frac{1}{12}} \sqrt{\log m} \right), \text{ when } m \geq C_1 \left( \log(C_2/\delta) \right)^{6/5}.$$

*Proof of Proposition B.19.* By Lemma B.18 (choose parameter $\tau = \Theta(m^{1/4})$), Lemma B.12 and

$$\left\| \nabla_{W^{(l)}} f_t^{(p),m}(z') \right\|_F \leq \left\| \nabla_{W^{(l)}} f_0^{(p),m}(z') \right\|_F + \left\| \nabla_{W^{(l)}} f_t^{(p),m}(z') - \nabla_{W^{(l)}} f_0^{(p),m}(z') \right\|_F;$$

$$\left\| \nabla_{V^{(l)}} f_t^{(p),m}(z') \right\|_F \leq \left\| \nabla_{V^{(l)}} f_0^{(p),m}(z') \right\|_F + \left\| \nabla_{V^{(l)}} f_t^{(p),m}(z') - \nabla_{V^{(l)}} f_0^{(p),m}(z') \right\|_F,$$

with probability at least $1 - \exp(-\Omega(m^{5/6}))$, we have

$$\left| \left\langle \nabla_{W^{(l)}} f_t^{(p),m}(z), \nabla_{W^{(l)}} f_t^{(p),m}(z') \right\rangle - \left\langle \nabla_{W^{(l)}} f_0^{(p),m}(z), \nabla_{W^{(l)}} f_0^{(p),m}(z') \right\rangle \right|$$

$$\leq \left\| \nabla_{W^{(l)}} f_0^{(p),m}(z) \right\|_F \left\| \nabla_{W^{(l)}} f_t^{(p),m}(z') - \nabla_{W^{(l)}} f_0^{(p),m}(z') \right\|_F$$

$$+ \left\| \nabla_{W^{(l)}} f_t^{(p),m}(z') \right\|_F \left\| \nabla_{W^{(l)}} f_t^{(p),m}(z) - \nabla_{W^{(l)}} f_0^{(p),m}(z) \right\|_F$$

$$\leq O(1) \cdot O\left( m^{-\frac{1}{12}} \sqrt{\log m} \right) + O(1) \cdot O\left( m^{-\frac{1}{12}} \sqrt{\log m} \right) \leq O\left( m^{-\frac{1}{12}} \sqrt{\log m} \right)$$

and similarly have

$$\left| \left\langle \nabla_{V^{(l)}} f_t^{(p),m}(z), \nabla_{V^{(l)}} f_t^{(p),m}(z') \right\rangle - \left\langle \nabla_{V^{(l)}} f_0^{(p),m}(z), \nabla_{V^{(l)}} f_0^{(p),m}(z') \right\rangle \right|$$

$$\leq O\left( m^{-\frac{1}{12}} \sqrt{\log m} \right)$$

for all $l \in [L]$ and $t \in T$ when $m$ is greater than some positive absolute constant $C$. Combine with Equation (9), with probability at least $1 - \exp(-\Omega(m^{5/6}))$, we can get

$$\sup_{t \in T} \left| r_t^{(p),m}(z, z') - r_0^{(p),m}(z, z') \right| = O\left( m^{-\frac{1}{12}} \sqrt{\log m} \right).$$

Also, it is easy to check that there exist some positive absolute constants $C_1 > 0$ and $C_2 \geq 1$ such that $C_1 \left( \log(C_2/\delta) \right)^{6/5} \geq C$ holds for $\delta \in (0,1)$ and when $m \geq C_1 \left( \log(C_2/\delta) \right)^{6/5}$, we have $1 - \exp\left( -\Omega\left( m^{5/6} \right) \right) \geq 1 - \delta$. $\qquad\square$

## B.5. Lazy Regime

**Lemma B.21.** *Let* $\delta \in (0,1)$ *and* $t \geq 0$. *Suppose that* $\left\| W_s^{(p,l)} - W_0^{(p,l)} \right\|_F = O(m^{1/4})$ *and* $\left\| V_s^{(p,l)} - V_0^{(p,l)} \right\|_F = O(m^{1/4})$ *hold for all* $s \in [0,t]$, $l \in [L]$ *and* $p \in [2]$. *Then there exists a polynomial* $\text{poly}(\cdot)$, *such that when* $m \geq \text{poly}\left( n, \lambda_0^{-1}, \log(1/\delta) \right)$, *with probability at least* $1 - \delta$, *for all* $s \in [0,t]$, *we have*

$$\| u(s) \|_2^2 \leq \exp\left( -\frac{\lambda_0}{n} s \right) \| u(0) \|_2^2 = \exp\left( -\frac{\lambda_0}{n} s \right) \| y \|_2^2,$$

*where* $u(t) := f_t^m(X) - y$.

*Proof.* Denote $\tilde{\lambda}_0(s) = \lambda_{\min}\big(r_s^m(\boldsymbol{X}, \boldsymbol{X})\big)$. By Weyl's inequality, we can get

$$
\begin{aligned}
\left|\tilde{\lambda}_0(s) - \lambda_0\right| &\leq \|r_s^m(\boldsymbol{X}, \boldsymbol{X}) - r(\boldsymbol{X}, \boldsymbol{X})\|_2 \leq \|r_s^m(\boldsymbol{X}, \boldsymbol{X}) - r(\boldsymbol{X}, \boldsymbol{X})\|_F \\
&\leq \|r_s^m(\boldsymbol{X}, \boldsymbol{X}) - r_0^m(\boldsymbol{X}, \boldsymbol{X})\|_F + \|r_0^m(\boldsymbol{X}, \boldsymbol{X}) - r(\boldsymbol{X}, \boldsymbol{X})\|_F \\
&\leq \frac{1}{2}\sum_{p=1}^{2}\left[\sum_{i,j=1}^{n}\left|r_s^{(p),m}(\boldsymbol{x}_i, \boldsymbol{x}_j) - r_0^{(p)m}(\boldsymbol{x}_i, \boldsymbol{x}_j)\right| + \sum_{i,j=1}^{n}\left|r_0^{(p),m}(\boldsymbol{x}_i, \boldsymbol{x}_j) - r(\boldsymbol{x}_i, \boldsymbol{x}_j)\right|\right].
\end{aligned}
$$

According to Proposition B.19 and Corollary B.7, for $\delta_0 = \delta/(2n^2)$, with probability at least $1 - 2n^2\delta_0 = 1 - \delta$, we can get

$$
\left|\tilde{\lambda}_0(s) - \lambda_0\right| \leq n^2 \cdot O\left(m^{-\frac{1}{12}}\sqrt{\log m}\right) + n^2 \cdot O(m^{-0.2}) \leq n^2 \cdot O\left(m^{-\frac{1}{15}}\right) \leq \frac{\lambda_0}{2} \text{ for all } s \in [0, t]
$$

when $m \geq C_1\left[\left(n^2\lambda_0^{-1}\right)^{15} + \left(\log\left(C_2 n^2/\delta\right)\right)^5\right]$ for some positive absolute constants $C_1 > 0$ and $C_2 \geq 1$. This implies that $\tilde{\lambda}_0(s) \geq \lambda_0/2$ holds for all $s \in [0, t]$. Then we have

$$
\frac{\mathrm{d}}{\mathrm{d}s}\|\boldsymbol{u}(s)\|_2^2 = -\frac{2}{n}\boldsymbol{u}(s)^\top K_s(\boldsymbol{X}, \boldsymbol{X})\boldsymbol{u}(s) \leq -\frac{\lambda_0}{n}\|\boldsymbol{u}(s)\|_2^2
$$

and thus

$$
\frac{\mathrm{d}}{\mathrm{d}s}\left(\exp\!\left(\tfrac{\lambda_0}{n}s\right)\|\boldsymbol{u}(s)\|_2^2\right) = \exp\!\left(\frac{\lambda_0}{n}s\right)\left(\frac{\lambda_0}{n}\|\boldsymbol{u}(s)\|_2^2 + \frac{\mathrm{d}\|\boldsymbol{u}(s)\|_2^2}{\mathrm{d}s}\right) \leq 0.
$$

Thus, with probability at least $1 - \delta$, we can get $\exp(\lambda_0 s/n)\|\boldsymbol{u}(s)\|_2^2 \leq \|\boldsymbol{u}(0)\|_2^2 = \|\boldsymbol{y}\|_2^2$ holds for all $s \in [0, t]$ when $m \geq C_1\left[\left(n^2\lambda_0^{-1}\right)^{15} + \left(\log\left(C_2 n^2/\delta\right)\right)^5\right]$. Finally, by choosing

$$
\mathrm{poly}\big(n, \lambda_0^{-1}, \log(1/\delta)\big) = C_1\left[\left(n^2\lambda_0^{-1}\right)^{15} + \left(2n + \log(1/\delta) + \log C_2\right)^5\right],
$$

we can complete the proof of this lemma.

$\square$

**Lemma B.22.** *Fix $l \in [L]$, $p \in [2]$ and let $\delta \in (0, 1)$, $t \geq 0$. Suppose that*

$$
\|f_s(\boldsymbol{X}) - \boldsymbol{y}\|_2 \leq \exp\!\left(-\tfrac{\lambda_0}{4n}s\right)\|\boldsymbol{y}\|_2 \qquad \text{holds for all } s \in [0, t],
$$

*then we have the following results:*

- *i) Suppose that $\left\|\boldsymbol{W}_s^{(p',l')} - \boldsymbol{W}_0^{(p',l')}\right\|_F \leq \frac{\sqrt{m}}{(\log m)^3}$ holds for all $(p', l') \neq (p, l)$ and $\left\|\boldsymbol{V}_s^{(p'',l'')} - \boldsymbol{V}_0^{(p'',l'')}\right\|_F \leq \frac{\sqrt{m}}{(\log m)^3}$ holds for all $l'' \in [L]$ and $p'' \in [2]$ when $s \in [0, t]$. Then there exists a polynomial $\mathrm{poly}(\cdot)$, such that when $m \geq \mathrm{poly}\big(n, \|\boldsymbol{y}\|_2, \lambda_0^{-1}, \log(1/\delta)\big)$, with probability at least $1 - \delta$, we have*

$$
\sup_{s \in [0,t]}\left\|\boldsymbol{W}_s^{(p,l)} - \boldsymbol{W}_0^{(p,l)}\right\|_F = O(n\|\boldsymbol{y}\|_2/\lambda_0);
$$

- *ii) Suppose that $\left\|\boldsymbol{V}_s^{(p',l')} - \boldsymbol{V}_0^{(p',l')}\right\|_F \leq \frac{\sqrt{m}}{(\log m)^3}$ holds for all $(p', l') \neq (p, l)$ and $\left\|\boldsymbol{W}_s^{(p'',l'')} - \boldsymbol{W}_0^{(p'',l'')}\right\|_F \leq \frac{\sqrt{m}}{(\log m)^3}$ holds for all $l'' \in [L]$ and $p'' \in [2]$ when $s \in [0, t]$. Then there exists a polynomial $\mathrm{poly}(\cdot)$, such that when $m \geq \mathrm{poly}\big(n, \|\boldsymbol{y}\|_2, \lambda_0^{-1}, \log(1/\delta)\big)$, with probability at least $1 - \delta$, we have*

$$
\sup_{s \in [0,t]}\left\|\boldsymbol{V}_t^{(p,l)} - \boldsymbol{V}_0^{(p,l)}\right\|_F = O(n\|\boldsymbol{y}\|_2/\lambda_0).
$$

*Proof.* First of all, we have

$$\left\| \boldsymbol{W}_{t_0}^{(p,l)} - \boldsymbol{W}_0^{(p,l)} \right\|_F = \left\| \int_0^{t_0} \mathrm{d}\boldsymbol{W}_s^{(p,l)} \right\|_F = \left\| \int_0^{t_0} \frac{1}{n} \sum_{i=1}^n (f_s^m(\boldsymbol{x}_i) - y_i) \nabla_{\boldsymbol{W}^{(p,l)}} f_s^m(\boldsymbol{x}_i) \, \mathrm{d}s \right\|_F$$

$$\leq \frac{1}{\sqrt{2}n} \sum_{i=1}^n \max_{0 \leq s \leq t_0} \left\| \nabla_{\boldsymbol{W}^{(p,l)}} f_s^{(p),m}(\boldsymbol{x}_i) \right\|_F \int_0^{t_0} \| f_s^m(\boldsymbol{X}) - \boldsymbol{y} \|_2 \, \mathrm{d}s$$

$$\leq O\left( \frac{\|\boldsymbol{y}\|_2}{\lambda_0} \right) \cdot \sum_{i=1}^n \max_{0 \leq s \leq t_0} \left\| \nabla_{\boldsymbol{W}^{(p,l)}} f_s^{(p),m}(\boldsymbol{x}_i) \right\|_F$$

for all $t_0 \in [0, t]$, and

$$\sup_{t_0 \in [0,t]} \left\| \boldsymbol{W}_{t_0}^{(p,l)} - \boldsymbol{W}_0^{(p,l)} \right\|_F \leq O\left( \frac{\|\boldsymbol{y}\|_2}{\lambda_0} \right) \cdot \sum_{i=1}^n \max_{0 \leq s \leq t} \left\| \nabla_{\boldsymbol{W}^{(p,l)}} f_s^{(p),m}(\boldsymbol{x}_i) \right\|_F. \tag{10}$$

Also, we can get

$$\left\| \nabla_{\boldsymbol{W}^{(p,l)}} f_s^{(p),m}(\boldsymbol{x}_i) \right\|_F \leq \left\| \nabla_{\boldsymbol{W}^{(p,l)}} f_0^{(p),m}(\boldsymbol{x}_i) \right\|_F + \left\| \nabla_{\boldsymbol{W}^{(p,l)}} f_s^{(p),m}(\boldsymbol{x}_i) - \nabla_{\boldsymbol{W}^{(p,l)}} f_0^{(p),m}(\boldsymbol{x}_i) \right\|_F$$

by the triangle inequality. For the first term, by Lemma B.12, we know that with probability at least $1 - \exp(-\Omega(m))$, we have $\left\| \nabla_{\boldsymbol{W}^{(p,l)}} f_0^{(p),m}(\boldsymbol{x}_i) \right\|_F = O(1)$ for any $i \in [n]$. So it suffices to bound the second term.

Denote $\mathcal{A} = \left\{ s \in [0, t] : \left\| \boldsymbol{W}_s^{(p,l)} - \boldsymbol{W}_0^{(p,l)} \right\|_F \geq \sqrt{m}/(\log m)^3 \right\}$. Assume that $\mathcal{A} \neq \varnothing$ and let $s_0 = \min \mathcal{A}$. Then for any $p', l'$, we have $\left\| \boldsymbol{W}_s^{(p',l')} - \boldsymbol{W}_0^{(p',l')} \right\|_F \leq \sqrt{m}/(\log m)^3$ and $\left\| \boldsymbol{V}_s^{(l')} - \boldsymbol{V}_0^{(l')} \right\|_F \leq \sqrt{m}/(\log m)^3$ when $s \in [0, s_0]$.

By Lemma B.18, we know for any $i \in [n]$, with probability at least $1 - \exp\left(-\Omega\left(m(\log m)^{-2}\right)\right) \geq 1 - \exp\left(-\Omega(m^{5/6})\right)$, we have

$$\max_{s \in [0,s_0]} \left\| \nabla_{\boldsymbol{W}^{(p,l)}} f_s^{(p),m}(\boldsymbol{x}_i) - \nabla_{\boldsymbol{W}^{(p,l)}} f_0^{(p),m}(\boldsymbol{x}_i) \right\|_F = O(1). \tag{11}$$

Combine with the definition of $s_0$, with probability at least $1 - n \exp\left(-\Omega(m^{5/6})\right)$, we have

$$\sqrt{m}/(\log m)^3 \leq \left\| \boldsymbol{W}_{s_0}^{(p,l)} - \boldsymbol{W}_0^{(p,l)} \right\|_F = O(n \|\boldsymbol{y}\|_2 / \lambda_0),$$

which will lead to contradiction when $m \geq \Omega\left(n \|\boldsymbol{y}\|_2 \lambda_0^{-1}\right)^5$. This means that $\mathcal{A} = \varnothing$ and Equation (11) holds for $s_0 = t$. Comibine with Equation (10), we can get the conclusion of $i$). Also, it is easy to check that there exists a positive absolute constant $C$ such that when $m \geq C \log(n/\delta)^{6/5}$, we have $1 - n \exp\left(-\Omega(m^{5/6})\right) \geq 1 - \delta$.

Finally, by choosing

$$\text{poly}\left(n, \lambda_0^{-1}, \log(1/\delta)\right) = C' \left[ \left(n \|\boldsymbol{y}\|_2 \lambda_0^{-1}\right)^5 + (n + \log(1/\delta))^2 + 1 \right]$$

for some positive absolute constant $C' > 0$, we can complete the proof of $i$). And we can prove $ii$) with the same above argument.

$\square$

**Lemma B.23.** *There exists a polynomial* $\text{poly}(\cdot)$*, such that for any* $\delta \in (0, 1)$*, when* $m \geq \text{poly}\left(n, \|\boldsymbol{y}\|_2, \lambda_0^{-1}, \log(1/\delta)\right)$*, then with probability at least* $1 - \delta$*, for all* $p \in [2]$ *and* $l \in [L]$*, we have*

$$\sup_{t \geq 0} \left\| \boldsymbol{W}_t^{(p,l)} - \boldsymbol{W}_0^{(p,l)} \right\|_F = O(m^{1/4}), \quad \sup_{t \geq 0} \left\| \boldsymbol{V}_t^{(p,l)} - \boldsymbol{V}_0^{(p,l)} \right\|_F = O(m^{1/4}).$$

*Proof of Lemma B.23.* Denote $t_0 = \min \left\{ t \geq 0 : \exists l, \, p \text{ such that } \left\| \mathbf{W}_t^{(p,l)} - \mathbf{W}_0^{(p,l)} \right\|_F \geq m^{1/4} \text{ or } \left\| \mathbf{V}_t^{(p,l)} - \mathbf{V}_0^{(p,l)} \right\|_F \geq m^{1/4} \text{ or } \|\mathbf{u}(t)\|_2 \geq \exp[-\lambda_0 t/(4n)] \|\mathbf{y}\|_2 \right\}$ and assume that $t_0$ is finite. Then for all $t \in [0, t_0]$, we can get

$$\left\| \mathbf{W}_t^{(p,l)} - \mathbf{W}_0^{(p,l)} \right\|_F \leq m^{1/4}, \quad \left\| \mathbf{V}_t^{(p,l)} - \mathbf{V}_0^{(p,l)} \right\|_F \leq m^{1/4} \text{ and } \|\mathbf{u}(t)\|_2 \leq \exp\left(-\frac{\lambda_0 t}{4n}\right) \|\mathbf{y}\|_2$$

hold for all $p$, $l$. According to Lemmas B.22 and B.21, there exists a polynomial $\mathrm{poly}(\cdot)$, such that when $m \geq \mathrm{poly}\left(n, \|\mathbf{y}\|_2, \lambda_0^{-1}, \log(1/\delta)\right)$, with probability at least $1 - \delta$, we have

$$\left\| \mathbf{W}_{t_0}^{(p,l)} - \mathbf{W}_0^{(p,l)} \right\|_F = O\left(n \|\mathbf{y}\|_2/\lambda_0\right), \qquad \left\| \mathbf{V}_{t_0}^{(p,l)} - \mathbf{V}_0^{(p,l)} \right\|_F = O\left(n \|\mathbf{y}\|_2/\lambda_0\right)$$

hold for all $p$, $l$ and

$$\|\mathbf{u}(t_0)\|_2 \leq \exp\left(-\frac{\lambda_0 t}{2n}\right) \|\mathbf{y}\|_2 \, .$$

Combine with the definition of $t_0$, we can get there exist $p$, $l$ such that

$$m^{1/4} \leq \left\| \mathbf{W}_{t_0}^{(p,l)} - \mathbf{W}_0^{(p,l)} \right\|_F = O\left(\frac{n\|\mathbf{y}\|_2}{\lambda_0}\right) \quad \text{or} \quad m^{1/4} \leq \left\| \mathbf{V}_{t_0}^{(p,l)} - \mathbf{V}_0^{(p,l)} \right\|_F = O\left(\frac{n\|\mathbf{y}\|_2}{\lambda_0}\right).$$

However, this will lead to contradiction when $m \geq C\left(n \|\mathbf{y}\|_2 \lambda_0^{-1}\right)^5$ for some positive absolute constant $C > 0$. $\qquad \square$

## B.6. Nearly Hölder Continuity of $r_{\theta(t)}^m$

**Lemma B.24.** *Let $\tau \in \left[\Omega(1/\sqrt{m}), O\left(\sqrt{m}/(\log m)^3\right)\right]$, $T \subseteq [0, \infty)$ and fix $\mathbf{z} \in \mathcal{D}$. Suppose that $\left\| \mathbf{W}_t^{(l)} - \mathbf{W}_0^{(l)} \right\|_F \leq \tau$ and $\left\| \mathbf{V}_t^{(l)} - \mathbf{V}_0^{(l)} \right\|_F \leq \tau$ hold for all $t \in T$ and $l \in [L]$. Then there exists a positive absolute constant $C$, such that with probability at least $1 - \exp\left(-\Omega(m^{2/3}\tau^{2/3})\right)$, for all $t \in T$, $l \in [L]$ and $\mathbf{x} \in \mathcal{D}$ such that $\|\mathbf{x} - \mathbf{z}\|_2 \leq O(1/m)$, we have*

- *i) $\left\| \mathbf{g}_{l,\mathbf{xz}}' \right\|_2 = O(\tau/\sqrt{m})$;*

- *ii) $\left\| \mathbf{D}_{\mathbf{xz}}^{(l)\prime} \right\|_0 = O\left(m^{2/3}\tau^{2/3}\right)$ and $\left\| \mathbf{D}_{\mathbf{xz}}^{(l)\prime} \mathbf{W}_t^{(l)} \boldsymbol{\alpha}_{t,\mathbf{x}}^{(l-1)} \right\|_2 = O(\tau)$;*

- *iii) $\left\| \Delta\boldsymbol{\alpha}_{\mathbf{xz}}^{(l)} \right\|_2 = O(\tau/\sqrt{m})$.*

*when $m$ is greater than the positive constant $C$.*

*Proof of Lemma B.24.* The proof of this lemma is similar to the proof of Lemma B.15. The only thing to note is that the conclusion of this lemma holds uniformly for $\mathbf{x} \in \mathcal{D}$ such that $\|\mathbf{x} - \mathbf{z}\|_2 \leq O(1/m)$ with high probability for any fixed $\mathbf{z}$. We inductively prove this lemma.

Since $\mathbf{A}$ does not change during the training, we can easily check that, with probability at least $1 - \exp(-\Omega(m))$, we have $\left\| \Delta\boldsymbol{\alpha}_{\mathbf{xz}}^{(0)} \right\|_2 = \|\mathbf{A}(\mathbf{x} - \mathbf{z})/\sqrt{m}\|_2 = O(1/m)$ for any $\mathbf{x} \in \mathcal{D}$ such that $\|\mathbf{x} - \mathbf{z}\|_2 \leq O(1/m)$, which means that $iii)$ holds for $l = 0$ since $\tau = \Omega(1/\sqrt{m})$. Then it only needs to be proven that

$$iii) \text{ holds for } l = k \quad \implies \quad \text{Lemma holds for } l = k + 1.$$

Now we assume that $iii)$ holds for $l = k \in \{0, 1, \cdots, L - 1\}$, then with probability at least $1 - \exp(-\Omega(m))$, for any $\mathbf{x} \in \mathcal{D}$ such that $\|\mathbf{x} - \mathbf{z}\|_2 = O(1/m)$, we have

$$\left\| \boldsymbol{\alpha}_{t,\mathbf{x}}^{(k)} \right\|_2 = \left\| \boldsymbol{\alpha}_{0,\mathbf{z}}^{(k)} + \Delta\boldsymbol{\alpha}_{\mathbf{xz}}^{(k)} \right\|_2 \leq \left\| \boldsymbol{\alpha}_{0,\mathbf{z}}^{(k)} \right\|_2 + \left\| \Delta\boldsymbol{\alpha}_{\mathbf{xz}}^{(k)} \right\|_2 = O(1) + O(\tau/\sqrt{m}) = O(1).$$

For $i$), similar to the proof of Lemma B.15, we can get

$$g'_{k+1,xz} = \sqrt{\frac{2}{m}}\left(\Delta W^{(k+1)}\alpha_{t,x}^{(k)} + W_0^{(k+1)}\Delta\alpha_{xz}^{(k)}\right).$$

Thus we can get $i$) holds for $l = k + 1$.

Considering that the conclusion of Lemma B.13 holds uniformly for $g'$ with high probability, taking $W_0 = W_0^{(k+1)}$, $h = \alpha_{0,z}^{(k)}$, $g' = g'_{k+1,xz}$ and $\delta = \Theta(\tau/\sqrt{m}) \le O\big((\log m)^{-3}\big)$, then we can get $ii$) holds for $l = k + 1$ with probability at least $1 - \exp\big(-\Omega(m^{2/3}\tau^{2/3})\big)$.

As for $iii$), it is easy to check that

$$\Delta\alpha_{xz}^{(k+1)} = \Delta\alpha_{xz}^{(k)} + \frac{\sqrt{2}a}{m}\Big[V_t^{(k+1)}D_{xz}^{(k+1)\prime}W_t^{(k+1)}\alpha_{t,x}^{(k)} + \Delta V^{(k+1)}D_{0,z}^{(k+1)}W_t^{(k+1)}\alpha_{t,x}^{(k)}$$
$$+ V_0^{(k+1)}D_{0,z}^{(k+1)}\left(W_t^{(k+1)}\alpha_{t,x}^{(k)} - W_0^{(k+1)}\alpha_{0,z}^{(k)}\right)\Big].$$

We have shown that, with probability at least $1 - \exp\big(-\Omega(m^{2/3}\tau^{2/3})\big)$, $i$) $ii$) hold for $l = k+1$. Combine with Lemma B.10, we can get $\left\|\Delta\alpha_{xz}^{(k+1)}\right\|_2 = O(\tau/\sqrt{m})$.

Thus, we finish the proof.

$\square$

**Lemma B.25.** *Let* $\tau \in \big[\Omega(1/\sqrt{m}), O\big(\sqrt{m}/(\log m)^3\big)\big]$, $T \subseteq [0,\infty)$ *and fix* $z \in \mathcal{D}$. *Suppose that* $\left\|W_t^{(l)} - W_0^{(l)}\right\|_F \le \tau$ *and* $\left\|V_t^{(l)} - V_0^{(l)}\right\|_F \le \tau$ *hold for all* $t \in T$ *and* $l \in [L]$. *Then there exists a positive absolute constant* $C$, *such that with probability at least* $1 - \exp\big(-\Omega(m^{2/3}\tau^{2/3})\big)$, *for all* $t \in T$, $l \in [L]$ *and* $x \in \mathcal{D}$ *such that* $\|x - z\|_2 \le O(1/m)$, *we have*

$$\left\|\Delta\delta_{xz}^{(l)}\right\|_2 = O\big(m^{1/3}\tau^{1/3}\sqrt{\log m}\big),$$

*when* $m$ *is greater than the positive constant* $C$.

*Proof of Lemma B.25.* The proof of this lemma is similar to the proof of Lemma B.16. The only thing to note is that the conclusion of this lemma holds uniformly for $x \in \mathcal{D}$ such that $\|x - z\|_2 \le O(1/m)$ with high probability for any fixed $z$. We inductively prove this lemma.

Base case: $\left\|\Delta\delta_{xz}^{(L)}\right\|_2 = 0$ since $v$ is fixed during the training process.

Assume that this lemma holds for $l + 1$, then with probability at least $1 - \exp(-\Omega(m))$, we have

$$\left\|\delta_{t,x}^{(l+1)}\right\|_2 = \left\|\Delta\delta_{xz}^{(l+1)} + \delta_{0,z}^{(l+1)}\right\|_2 \le \left\|\Delta\delta_{xz}^{(l+1)}\right\|_2 + \left\|\delta_{0,z}^{(l+1)}\right\|_2 \le O(\sqrt{m})$$

for all $x \in \mathcal{D}$. Moreover, it is easy to check that

$$\Delta\delta_{xz}^{(l)} = \Delta\delta_{xz}^{(l+1)} + \frac{\sqrt{2}a}{m}\Big[W_0^{(l+1),T}D_{xz}^{(l+1)\prime}V_0^{(l+1),T}\delta_{0,z}^{(l+1)} + \Delta W^{(l+1),T}D_{t,x}^{(l+1)}V_0^{(l+1),T}\delta_{0,x}^{(l+1)}$$
$$+ W_t^{(l+1),T}D_{t,x}^{(l+1)}\Delta V^{(l+1),T}\delta_{0,x}^{(l+1)} + W_t^{(l+1),T}D_{t,x}^{(l+1)}V_t^{(l+1),T}\Delta\delta_{xz}^{(l+1)}\Big].$$

Let us denote the four terms within the square brackets, excluding the factor '$\sqrt{2}a/m$' outside the brackets, as $u_1$ to $u_4$ respectively. We can control $\|u_2\|_2$, $\|u_3\|_2$, and $\|u_4\|_2$ using the same method as in the proof of Lemma B.16, since $\left\|D_{t,x}^{(l+1)}\right\|_2 \le 1$ and the conclusion of Lemma B.10 holds uniformly for $x$.

As for $u_1$, if $\delta_{0,z}^{(l+1)} = 0$ or $\left\|D_{xz}^{(l+1)\prime}\right\|_0 = 0$, we have $\|u_1\|_2 = 0$. Therefore, we consider the case where $\delta_{0,z}^{(l+1)} \ne 0$ and $\left\|D_{xz}^{(l+1)\prime}\right\|_0 \ge 1$. Denote $\tilde{\delta}_z = \delta_{0,z}^{(l+1)}/\left\|\delta_{0,z}^{(l+1)}\right\|_2$ for $\delta_{0,z}^{(l+1)} \in \mathbb{R}^m\backslash\{0\}$, we can get

$$\|u_1\|_2 \le \left\|W_0^{(l+1)}\right\|_2\left\|D_{xz}^{(l+1)\prime}V_0^{(l+1)}\tilde{\delta}_z\right\|_2\left\|\delta_{0,z}^{(l+1)}\right\|_2 \le O(m)\left\|D_{xz}^{(l+1)\prime}V_0^{(l+1)}\tilde{\delta}_z\right\|_2.$$

Using the randomness of $\boldsymbol{V}_0^{(l+1)}$, for any fixed $\tilde{\boldsymbol{\delta}}_z$, we have $\boldsymbol{V}_0^{(l+1)}\tilde{\boldsymbol{\delta}}_z \sim \mathcal{N}(\boldsymbol{0}, \boldsymbol{I}_m)$. Thus, by Lemma B.14 and taking $s \geq \left\|\boldsymbol{D}_{xz}^{(l+1)\prime}\right\|_0$, with probability at least $1 - \exp(-\Omega(s \log m))$, we can get

$$\forall \boldsymbol{u} \in \mathbb{R}^m \text{ s.t. } \|\boldsymbol{u}\|_0 \leq s, \quad \text{we have} \quad \left|\boldsymbol{u}^\top \boldsymbol{V}_0^{(l+1)}\tilde{\boldsymbol{\delta}}_z\right| \leq 9\sqrt{s \log m}\,\|\boldsymbol{u}\|_2.$$

According to Lemma B.24, with probability at least $1 - \exp\left(-\Omega(m^{2/3}\tau^{2/3})\right)$, for any $\boldsymbol{x} \in \mathcal{D}$ such that $\|\boldsymbol{x} - \boldsymbol{z}\|_2 \leq O(1/m)$, we have $\left\|\boldsymbol{D}_{xz}^{(l)\prime}\right\|_0 = O\left(m^{2/3}\tau^{2/3}\right)$. By taking $s = \Theta(m^{2/3}\tau^{2/3})$, we can get

$$\left\|\boldsymbol{D}_{xz}^{(l+1)\prime}\boldsymbol{V}_0^{(l+1)}\tilde{\boldsymbol{\delta}}_z\right\|_2 = \sup_{\boldsymbol{u} \in \mathbb{S}^{m-1}} \left|\boldsymbol{u}^\top \boldsymbol{D}_{xz}^{(l+1)\prime}\boldsymbol{V}_0^{(l+1)}\tilde{\boldsymbol{\delta}}_z\right| \leq 9\sqrt{s \log m}$$

holds uniformly for $\boldsymbol{x}$.

Combining the above discussions, we can conclude that $\left\|\Delta\boldsymbol{\delta}_{xz}^{(l)}\right\|_2 \leq O\left(m^{1/3}\tau^{1/3}\sqrt{\log m}\right)$.

$\square$

**Lemma B.26.** *Let* $\tau \in \left[\Omega(1/\sqrt{m}), O\left(\sqrt{m}/(\log m)^3\right)\right]$, $T \subseteq [0, \infty)$ *and fix* $l \in [L]$, $\boldsymbol{z} \in \mathcal{D}$. *Suppose that* $\left\|\boldsymbol{W}_t^{(l)} - \boldsymbol{W}_0^{(l)}\right\|_F \leq \tau$ *and* $\left\|\boldsymbol{V}_t^{(l)} - \boldsymbol{V}_0^{(l)}\right\|_F \leq \tau$ *hold for all* $t \in T$, *then there exists a positive absolute constant* $C$, *such that with probability at least* $1 - \exp\left(-\Omega(m^{2/3}\tau^{2/3})\right)$ *over the randomness of* $\boldsymbol{W}_0^{(l)}$ *and* $\boldsymbol{V}_0^{(l)}$, *for all* $t \in T$, $l \in [L]$ *and* $\boldsymbol{x} \in \mathcal{D}$ *such that* $\|\boldsymbol{x} - \boldsymbol{z}\|_2 \leq O(1/m)$, *we have*

$$\left\|\Delta\boldsymbol{\gamma}_{xz}^{(l)}\right\|_2 = O\left(m^{-1/6}\tau^{1/3}\sqrt{\log m}\right), \quad \left\|\Delta\boldsymbol{\eta}_{xz}^{(l)}\right\|_2 = O\left(\frac{\tau}{m}\right);$$
$$\left\|\boldsymbol{\gamma}_{t,x}^{(l)}\right\|_2 = O(1), \qquad \left\|\boldsymbol{\eta}_{t,x}^{(l)}\right\|_2 = O(1/\sqrt{m})$$

*when* $m$ *is greater than the positive constant* $C$.

*Proof of Lemma B.26.* First of all, we have

$$\boldsymbol{\gamma}_{t,x}^{(l)} - \boldsymbol{\gamma}_{0,z}^{(l)} = \frac{\sqrt{2}}{m}\left(\boldsymbol{D}_{t,x}^{(l)}\boldsymbol{V}_t^{(l),T}\boldsymbol{\delta}_{t,x}^{(l)} - \boldsymbol{D}_{0,z}^{(l)}\boldsymbol{V}_0^{(l),T}\boldsymbol{\delta}_{0,z}^{(l)}\right)$$
$$= \frac{\sqrt{2}}{m}\left(\boldsymbol{D}_{xz}^{(l)\prime}\boldsymbol{V}_0^{(l),T}\boldsymbol{\delta}_{0,z}^{(l)} + \boldsymbol{D}_{t,x}^{(l)}\Delta\boldsymbol{V}^{(l),T}\boldsymbol{\delta}_{t,x}^{(l)} + \boldsymbol{D}_t^{(l)}\boldsymbol{V}_0^{(l),T}\Delta\boldsymbol{\delta}_{xz}^{(l)}\right).$$

Using the similar proof technique as the previous lemma, we can establish that with probability at least $1 - \exp\left(-\Omega(m^{2/3}\tau^{2/3})\right)$, we have

$$\left\|\boldsymbol{D}_{xz}^{(l)\prime}\boldsymbol{V}_0^{(l),T}\boldsymbol{\delta}_{0,z}^{(l)}\right\|_2 = O\left(m^{5/6}\tau^{1/3}\sqrt{\log m}\right).$$

According to Corollary B.9, Lemma B.10 and Lemma B.25 we can get

$$\left\|\boldsymbol{D}_{t,x}^{(l)}\Delta\boldsymbol{V}^{(l),T}\boldsymbol{\delta}_{t,x}^{(l)}\right\|_2 = O\left(\tau\sqrt{m}\right); \qquad \left\|\boldsymbol{D}_t^{(l)}\boldsymbol{V}_0^{(l),T}\Delta\boldsymbol{\delta}_{xz}^{(l)}\right\|_2 = O\left(m^{5/6}\tau^{1/3}\log m\right).$$

Thus, we can get $\left\|\Delta\boldsymbol{\gamma}_{xz}^{(l)}\right\|_2 = O\left(m^{-1/6}\tau^{1/3}\log m\right)$.

As for $\left\|\Delta\boldsymbol{\eta}_{xz}^{(l)}\right\|_2$, we can similarly get

$$\left\|\Delta\boldsymbol{\eta}_{xz}^{(l)}\right\|_2 = \frac{\sqrt{2}}{m}\left\|\boldsymbol{D}_{xz}^{(l)\prime}\boldsymbol{W}_t^{(l)}\boldsymbol{\alpha}_{t,x}^{(l-1)} + \boldsymbol{D}_{0,z}^{(l)}\Delta\boldsymbol{W}^{(l)}\boldsymbol{\alpha}_{t,x}^{(l-1)} + \boldsymbol{D}_{0,z}^{(l)}\boldsymbol{W}_0^{(l)}\Delta\boldsymbol{\alpha}_{xz}^{(l-1)}\right\|_2$$
$$\leq \frac{\sqrt{2}}{m}\left(O(\tau) + O(\tau) + O(\tau)\right) = O\left(\frac{\tau}{m}\right)$$

according to Corollary B.9, Lemma B.10 and Lemma B.24. With the above results, we can easily get

$$\left\|\boldsymbol{\gamma}_{t,x}^{(l)}\right\|_2 \leq \left\|\boldsymbol{\gamma}_{0,z}^{(l)}\right\|_2 + \left\|\Delta\boldsymbol{\gamma}_{xz}^{(l)}\right\|_2 = O(1), \quad \left\|\boldsymbol{\eta}_{t,x}^{(l)}\right\|_2 \leq \left\|\boldsymbol{\eta}_0^{(l)}\right\|_2 + \left\|\Delta\boldsymbol{\eta}_{xz}^{(l)}\right\|_2 = O\left(\frac{1}{\sqrt{m}}\right)$$

since $\tau = O\left(\sqrt{m}/(\log m)^3\right)$.

$\square$

**Lemma B.27.** *Let $\tau \in \left[\Omega(1/\sqrt{m}), O\left(\sqrt{m}/(\log m)^3\right)\right]$, $T \subseteq [0, \infty)$ and fix $z \in \mathcal{D}$. Suppose that $\left\|\boldsymbol{W}_t^{(l)} - \boldsymbol{W}_0^{(l)}\right\|_F \leq \tau$ and $\left\|\boldsymbol{V}_t^{(l)} - \boldsymbol{V}_0^{(l)}\right\|_F \leq \tau$ hold for all $t \in T$ and $l \in [L]$. Then there exists a positive absolute constant $C$, such that with probability at least $1 - \exp\left(-\Omega(m^{2/3}\tau^{2/3})\right)$, for all $l \in [L]$ and $\boldsymbol{x} \in \mathcal{D}$ such that $\|\boldsymbol{x} - \boldsymbol{z}\|_2 \leq O(1/m)$, we have*

$$\sup_{t \in T} \left\|\nabla_{\boldsymbol{W}^{(l)}} f_t^{(p),m}(\boldsymbol{x}) - \nabla_{\boldsymbol{W}^{(l)}} f_0^{(p),m}(\boldsymbol{z})\right\|_F = O\left(m^{-1/6}\tau^{1/3}\sqrt{\log m}\right);$$

$$\sup_{t \in T} \left\|\nabla_{\boldsymbol{V}^{(l)}} f_t^{(p),m}(\boldsymbol{x}) - \nabla_{\boldsymbol{V}^{(l)}} f_0^{(p),m}(\boldsymbol{z})\right\|_F = O\left(m^{-1/6}\tau^{1/3}\sqrt{\log m}\right),$$

*when $m$ is greater than the positive constant $C$.*

*Proof of Lemma B.27.* According to Equation (8), we have

$$\left\|\nabla_{\boldsymbol{W}^{(l)}} f_t^{(p),m}(\boldsymbol{x}) - \nabla_{\boldsymbol{W}^{(l)}} f_0^{(p),m}(\boldsymbol{z})\right\|_F = \left\|a\boldsymbol{\gamma}_{t,\boldsymbol{x}}^{(l)}\boldsymbol{\alpha}_{t,\boldsymbol{x}}^{(l-1),T} - a\boldsymbol{\gamma}_{0,\boldsymbol{z}}^{(l)}\boldsymbol{\alpha}_{0,\boldsymbol{z}}^{(l-1),T}\right\|_F$$

$$= a\left\|\boldsymbol{\gamma}_{t,\boldsymbol{x}}^{(l)}\Delta\boldsymbol{\alpha}_{\boldsymbol{xz}}^{(l-1),T} + \Delta\boldsymbol{\gamma}_{\boldsymbol{xz}}^{(l)}\boldsymbol{\alpha}_{0,\boldsymbol{z}}^{(l-1),T}\right\|_F \leq a\left\|\boldsymbol{\gamma}_{t,\boldsymbol{x}}^{(l)}\Delta\boldsymbol{\alpha}_{\boldsymbol{xz}}^{(l-1),T}\right\|_F + a\left\|\Delta\boldsymbol{\gamma}_{\boldsymbol{xz}}^{(l)}\boldsymbol{\alpha}_{0,\boldsymbol{z}}^{(l-1),T}\right\|_F$$

$$= a\left\|\boldsymbol{\gamma}_{t,\boldsymbol{x}}^{(l)}\right\|_2\left\|\Delta\boldsymbol{\alpha}_{\boldsymbol{xz}}^{(l-1)}\right\|_2 + a\left\|\Delta\boldsymbol{\gamma}_{\boldsymbol{xz}}^{(l)}\right\|_2\left\|\boldsymbol{\alpha}_{0,\boldsymbol{z}}^{(l-1)}\right\|_2 \leq O\left(m^{-1/6}\tau^{1/3}\sqrt{\log m}\right)$$

according to Lemmas B.26 and B.24 *iii*). Similarly, we can also get

$$\left\|\nabla_{\boldsymbol{V}^{(l)}} f_t^{(p),m}(\boldsymbol{z}) - \nabla_{\boldsymbol{V}^{(l)}} f_0^{(p),m}(\boldsymbol{z})\right\|_F = \left\|a\boldsymbol{\delta}_{t,\boldsymbol{x}}^{(l)}\boldsymbol{\eta}_{t,\boldsymbol{x}}^{(l),T} - a\boldsymbol{\delta}_{0,\boldsymbol{z}}^{(l)}\boldsymbol{\eta}_{0,\boldsymbol{z}}^{(l),T}\right\|_F$$

$$\leq a\left\|\boldsymbol{\eta}_{t,\boldsymbol{x}}^{(l)}\right\|_2\left\|\Delta\boldsymbol{\delta}_{\boldsymbol{xz}}^{(l)}\right\|_2 + a\left\|\Delta\boldsymbol{\eta}_{\boldsymbol{xz}}^{(l)}\right\|_2\left\|\boldsymbol{\delta}_{0,\boldsymbol{z}}^{(l)}\right\|_2 \leq O\left(m^{-1/6}\tau^{1/3}\sqrt{\log m}\right)$$

according to Lemmas B.26 and B.25.

Thus, we finish the proof.

$\square$

**Proposition B.28.** *Fix $\boldsymbol{z}, \boldsymbol{z}' \in \mathcal{D}$ and let $\delta \in (0, 1)$, $T \subseteq [0, \infty)$. Suppose that $\left\|\boldsymbol{W}_t^{(l)} - \boldsymbol{W}_0^{(l)}\right\|_F = O(m^{1/4})$ and $\left\|\boldsymbol{V}_t^{(l)} - \boldsymbol{V}_0^{(l)}\right\|_F = O(m^{1/4})$ hold for all $l \in [L]$ and $t \in T$. Then there exist some positive absolute constants $C_1 > 0$ and $C_2 \geq 1$, such that with probability at least $1 - \delta$, for any $\boldsymbol{x}, \boldsymbol{x}' \in \mathcal{D}$ such that $\|\boldsymbol{x} - \boldsymbol{z}\|_2, \|\boldsymbol{x}' - \boldsymbol{z}'\|_2 \leq O(1/m)$, we have*

$$\sup_{t \in T} \left|r_t^{(p),m}(\boldsymbol{x}, \boldsymbol{x}') - r_0^{(p),m}(\boldsymbol{z}, \boldsymbol{z}')\right| = O\left(m^{-\frac{1}{12}}\sqrt{\log m}\right), \text{ when } m \geq C_1\left(\log(C_2/\delta)\right)^{6/5}.$$

*Proof of Proposition B.28.* By Lemma B.27 (choose parameter $\tau = \Theta(m^{1/4})$), Lemma B.12 and

$$\left\|\nabla_{\boldsymbol{W}^{(l)}} f_t^{(p),m}(\boldsymbol{x})\right\|_F \leq \left\|\nabla_{\boldsymbol{W}^{(l)}} f_0^{(p),m}(\boldsymbol{z})\right\|_F + \left\|\nabla_{\boldsymbol{W}^{(l)}} f_t^{(p),m}(\boldsymbol{x}) - \nabla_{\boldsymbol{W}^{(l)}} f_0^{(p),m}(\boldsymbol{z})\right\|_F;$$

$$\left\|\nabla_{\boldsymbol{V}^{(l)}} f_t^{(p),m}(\boldsymbol{x}')\right\|_F \leq \left\|\nabla_{\boldsymbol{V}^{(l)}} f_0^{(p),m}(\boldsymbol{z}')\right\|_F + \left\|\nabla_{\boldsymbol{V}^{(l)}} f_t^{(p),m}(\boldsymbol{x}') - \nabla_{\boldsymbol{V}^{(l)}} f_0^{(p),m}(\boldsymbol{z}')\right\|_F,$$

with probability at least $1 - \exp(-\Omega(m^{5/6}))$, we have

$$\left|\left\langle\nabla_{\boldsymbol{W}^{(l)}} f_t^{(p),m}(\boldsymbol{x}), \nabla_{\boldsymbol{W}^{(l)}} f_t^{(p),m}(\boldsymbol{x}')\right\rangle - \left\langle\nabla_{\boldsymbol{W}^{(l)}} f_0^{(p),m}(\boldsymbol{z}), \nabla_{\boldsymbol{W}^{(l)}} f_0^{(p),m}(\boldsymbol{z}')\right\rangle\right|$$

$$\leq \left\|\nabla_{\boldsymbol{W}^{(l)}} f_0^{(p),m}(\boldsymbol{z})\right\|_F\left\|\nabla_{\boldsymbol{W}^{(l)}} f_t^{(p),m}(\boldsymbol{x}') - \nabla_{\boldsymbol{W}^{(l)}} f_0^{(p),m}(\boldsymbol{z}')\right\|_F$$

$$+ \left\|\nabla_{\boldsymbol{W}^{(l)}} f_t^{(p),m}(\boldsymbol{x}')\right\|_F\left\|\nabla_{\boldsymbol{W}^{(l)}} f_t^{(p),m}(\boldsymbol{x}) - \nabla_{\boldsymbol{W}^{(l)}} f_0^{(p),m}(\boldsymbol{z})\right\|_F$$

$$\leq O(1) \cdot O\left(m^{-\frac{1}{12}}\sqrt{\log m}\right) + O(1) \cdot O\left(m^{-\frac{1}{12}}\sqrt{\log m}\right) \leq O\left(m^{-\frac{1}{12}}\sqrt{\log m}\right)$$

and similarly have

$$\left|\left\langle \nabla_{\boldsymbol{V}^{(l)}} f_t^{(p),m}(\boldsymbol{x}), \nabla_{\boldsymbol{V}^{(l)}} f_t^{(p),m}(\boldsymbol{x}') \right\rangle - \left\langle \nabla_{\boldsymbol{V}^{(l)}} f_0^{(p),m}(\boldsymbol{z}), \nabla_{\boldsymbol{V}^{(l)}} f_0^{(p),m}(\boldsymbol{z}') \right\rangle\right| \le O\left(m^{-\frac{1}{12}}\sqrt{\log m}\right)$$

for all $l \in [L]$, $t \in T$ and $\boldsymbol{x}, \boldsymbol{x}' \in \mathcal{D}$ such that $\|\boldsymbol{x} - \boldsymbol{z}\|_2$, $\|\boldsymbol{x}' - \boldsymbol{z}'\|_2 \le O(1/m)$ when $m$ is greater than some positive absolute constant $C$. Combine with Equation (9), with probability at least $1 - \exp(-\Omega(m^{5/6}))$, we can get

$$\sup_{t \in T}\left|r_t^{(p),m}(\boldsymbol{x}, \boldsymbol{x}') - r_0^{(p),m}(\boldsymbol{z}, \boldsymbol{z}')\right| = O\left(m^{-\frac{1}{12}}\sqrt{\log m}\right).$$

Also, it is easy to check that there exist some positive absolute constants $C_1 > 0$ and $C_2 \ge 1$ such that $C_1\left(\log(C_2/\delta)\right)^{6/5} \ge C$ holds for $\delta \in (0,1)$ and when $m \ge C_1\left(\log(C_2/\delta)\right)^{6/5}$, we have $1 - \exp\left(-\Omega\left(m^{5/6}\right)\right) \ge 1 - \delta$.

□

### B.7. Hölder Continuity of $r$

For our convenience let us first introduce the following definition of Hölder spaces. For a compact set $\Omega$ and $\alpha \in [0,1]$, let us define a semi-norm for $f : \Omega \to \mathbb{R}$ by

$$|f|_{0,\alpha} = \sup_{x,y\in\Omega,\ x\neq y} \frac{|f(x) - f(y)|}{\|x - y\|^\alpha}$$

and define the Hölder space by

$$C^{0,\alpha}(\Omega) = \left\{ f \in C(\Omega) : |f|_{0,\alpha} < \infty \right\}, \tag{12}$$

which is equipped with norm $\|f\|_{C^{0,\alpha}(\Omega)} = \sup_{x\in\Omega}|f(x)| + |f|_\alpha$. Then it is easy to show that

- i) $C^{0,\alpha}(\Omega) \subseteq C^{0,\beta}(\Omega)$ if $\beta \le \alpha$;
- ii) if $f, g \in C^{0,\alpha}(\Omega)$, then $f + g$, $fg \in C^\alpha(\Omega)$;
- iii) if $f \in C^{0,\alpha}(\Omega_1)$ and $g \in C^{0,\beta}(\Omega_2)$ with Ran $g \subseteq \Omega_1$, then $f \circ g \in C^{0,\alpha\beta}(\Omega_2)$.

**Proposition B.29.** *We have $r \in C^{0,s}(\Omega)$ with $s = 2^{-L}$ and $\Omega = \mathcal{D}^2$, that is, there is some constant $C > 0$ that*

$$|r(\boldsymbol{x}, \boldsymbol{x}') - r(\boldsymbol{z}, \boldsymbol{z}')| \le C\|(\boldsymbol{x}, \boldsymbol{x}') - (\boldsymbol{z}, \boldsymbol{z}')\|_2^s.$$

*Proof of Proposition B.29.* Recall that $r$ is given by

$$r(\boldsymbol{x}, \boldsymbol{x}') = a^2\|\boldsymbol{x}\|\|\boldsymbol{x}'\| \sum_{l=1}^{L} B_{l+1}(\tilde{\boldsymbol{x}}, \tilde{\boldsymbol{x}}')\left[(1+a^2)^{l-1}\kappa_1\left(\frac{K_{l-1}(\tilde{\boldsymbol{x}},\tilde{\boldsymbol{x}}')}{(1+a^2)^{l-1}}\right) + K_{l-1}(\tilde{\boldsymbol{x}}, \tilde{\boldsymbol{x}}')\cdot\kappa_0\left(\frac{K_{l-1}(\tilde{\boldsymbol{x}},\tilde{\boldsymbol{x}}')}{(1+a^2)^{l-1}}\right)\right],$$

where $\tilde{\boldsymbol{x}} = \boldsymbol{x}/\|\boldsymbol{x}\|$, $\tilde{\boldsymbol{x}}' = \boldsymbol{x}'/\|\boldsymbol{x}'\|$, $K_0(\tilde{\boldsymbol{x}}, \tilde{\boldsymbol{x}}') = \tilde{\boldsymbol{x}}^\top\tilde{\boldsymbol{x}}'$, $B_{L+1}(\tilde{\boldsymbol{x}}, \tilde{\boldsymbol{x}}') = 1$ and

$$\kappa_0(u) = \frac{1}{\pi}(\pi - \arccos u), \qquad \kappa_1(u) = \frac{1}{\pi}\left(u(\pi - \arccos u) + \sqrt{1 - u^2}\right)$$

$$K_l(\tilde{\boldsymbol{x}}, \tilde{\boldsymbol{x}}') = K_{l-1}(\tilde{\boldsymbol{x}}, \tilde{\boldsymbol{x}}') + a^2(1-a^2)^{l-1}\kappa_1\left(\frac{K_{l-1}(\tilde{\boldsymbol{x}}, \tilde{\boldsymbol{x}}')}{(1+a^2)^{l-1}}\right),$$

$$B_l(\tilde{\boldsymbol{x}}, \tilde{\boldsymbol{x}}') = B_{l+1}(\tilde{\boldsymbol{x}}, \tilde{\boldsymbol{x}}')\left[1 + a^2\kappa_0\left(\frac{K_{l-1}(\tilde{\boldsymbol{x}}, \tilde{\boldsymbol{x}}')}{(1+a^2)^{l-1}}\right)\right].$$

Since $r$ is symmetric, by triangle inequality it suffices to prove that $r(\boldsymbol{x}_0, \cdot) \in C^{0,s}(\mathcal{D})$ with $|r(\boldsymbol{x}_0, \cdot)|_{0,s}$ bounded by a constant independent of $\boldsymbol{x}_0$. It is easy to check that $\boldsymbol{x} \mapsto \tilde{\boldsymbol{x}}^\top\tilde{\boldsymbol{x}}_0 \in C^{0,1}(\mathcal{D})$ and

$$|\arccos\mu - \arccos\nu| = O(\sqrt{|\mu - \nu|}) \text{ and } |\sqrt{1 - \mu^2} - \sqrt{1 - \nu^2}| = O(\sqrt{|\mu - \nu|}),$$

meaning that $\kappa_0, \kappa_1 \in C^{0,1/2}([-1, 1])$. Thus, $r \in C^{0,s}(\Omega)$ with $s = (1/2)^L$.

□

## B.8. Proof of the kernel uniform convergence

*Proof of Theorem B.1.* By Lemma B.23, there exists a polynomial $\mathrm{poly}_1(\cdot)$, such that for any $\delta \in (0,1)$, when $m \geq \mathrm{poly}_1\big(n, \|\boldsymbol{y}\|_2, \lambda_0^{-1}, \log(1/\delta)\big)$, then with probability at least $1 - \delta/2$, for all $p \in [2]$ and $l \in [L]$, we have

$$\sup_{t \geq 0} \left\| \boldsymbol{W}_t^{(p,l)} - \boldsymbol{W}_0^{(p,l)} \right\|_F = O(m^{1/4}), \quad \sup_{t \geq 0} \left\| \boldsymbol{V}_t^{(p,l)} - \boldsymbol{V}_0^{(p,l)} \right\|_F = O(m^{1/4}).$$

Since $\|\boldsymbol{x}\|_2 \leq C_{\mathcal{D}}$, we have an $\varepsilon$-net $\mathcal{N}_\varepsilon$ of $\mathcal{D}$ such that the cardinality $|\mathcal{N}_\varepsilon| = O(\varepsilon^{-d})$. We choose $\varepsilon = 1/m^{2^L}$ and thus $\log |\mathcal{N}_\varepsilon| = O(\log m)$. Denote $B_{\boldsymbol{z}}(\varepsilon) = \{\boldsymbol{x} \in \mathcal{D} : \|\boldsymbol{x} - \boldsymbol{z}\|_2 \leq \varepsilon\}$. Then, fixing $\boldsymbol{z}, \boldsymbol{z}' \in \mathcal{N}_\varepsilon$, for any $\boldsymbol{x} \in B_{\boldsymbol{z}}(\varepsilon)$ and $\boldsymbol{x}' \in B_{\boldsymbol{z}'}(\varepsilon)$, we have

$$|r_t^m(\boldsymbol{x}, \boldsymbol{x}') - r(\boldsymbol{x}, \boldsymbol{x}')| \leq |r_0^m(\boldsymbol{z}, \boldsymbol{z}') - r(\boldsymbol{z}, \boldsymbol{z}')| + |r(\boldsymbol{z}, \boldsymbol{z}') - r(\boldsymbol{x}, \boldsymbol{x}')|$$
$$+ |r_t^m(\boldsymbol{x}, \boldsymbol{x}') - r_0^m(\boldsymbol{z}, \boldsymbol{z}')|$$

Then, noticing that $r_t^m = \left( r_t^{(1),m} + r_t^{(2),m} \right)/2$, we control the three terms on the right hand side by Propositions B.7, B.29 and B.28 respectively. We have shown that

$$|r_0^m(\boldsymbol{z}, \boldsymbol{z}') - r(\boldsymbol{z}, \boldsymbol{z}')| \leq O(m^{-0.2}), \qquad |r(\boldsymbol{x}, \boldsymbol{x}') - r(\boldsymbol{z}, \boldsymbol{z}')| \leq O(1/m),$$
$$\sup_{t \geq 0} \sup_{\boldsymbol{x} \in B_{\boldsymbol{z}}(\varepsilon)} \sup_{\boldsymbol{x}' \in B_{\boldsymbol{z}'}(\varepsilon)} |r_t^m(\boldsymbol{x}, \boldsymbol{x}') - r_0^m(\boldsymbol{z}, \boldsymbol{z}')| \leq O\big(m^{-1/12} \sqrt{\log m}\big),$$

with probability at least $1 - \delta/\big(2|\mathcal{N}_\varepsilon|^2\big)$ if $m \geq C_1 \log\big(C_2 |\mathcal{N}_\varepsilon|^2/\delta\big)^5$ for some positive absolute constants $C_1 > 0$ and $C_2 \geq 1$. There exists a polynomial $\mathrm{poly}_2(\cdot)$, such that when $m \geq \mathrm{poly}_2(\log(1/\delta))$, we have $m \geq C_1 \log\big(C_2 |\mathcal{N}_\varepsilon|^2/\delta\big)^5$, since $\log |\mathcal{N}_\varepsilon| = O(\log m)$. By applying the union bound for any pair $\boldsymbol{z}, \boldsymbol{z}' \in \mathcal{N}_\varepsilon$, we have with probability at least $1 - \delta$,

$$\sup_{t \geq 0} \sup_{\boldsymbol{x}, \boldsymbol{x}' \in \mathcal{D}} |r_t^m(\boldsymbol{x}, \boldsymbol{x}') - r(\boldsymbol{x}, \boldsymbol{x}')| \leq O\big(m^{-1/12} \sqrt{\log m}\big)$$

if $m \geq \mathrm{poly}_1\big(n, \|\boldsymbol{y}\|_2, \lambda_0^{-1}, \log(1/\delta)\big) + \mathrm{poly}_2(\log(1/\delta))$.

$\square$

# C. Proof of Theorem 3.1

## C.1. Useful simplification when the data is on $\mathbb{S}^{d-1}$

To facilitate the proof, we first perform some preprocessing on the expression of the RNTK. These steps are also applicable to the proof of Theorem 3.4.

Using the explicit expression of the RNTK (5) and the definition of the normalized RNTK (6), we derive the following expression:

$$\bar{r}^{(L)}(\boldsymbol{x}, \boldsymbol{x}') = \frac{r^{(L)}(\boldsymbol{x}, \boldsymbol{x}')}{4L\alpha^2(1+\alpha^2)^{L-1}} = \frac{\left\| \binom{\boldsymbol{x}}{1} \right\| \left\| \binom{\boldsymbol{x}'}{1} \right\| \cdot r_0(\tilde{\boldsymbol{x}}, \tilde{\boldsymbol{x}}')}{4L(1+\alpha^2)^{L-1}}.$$

Furthermore, when $\boldsymbol{x}, \boldsymbol{x}' \in \mathbb{S}^{d-1}$ (this is exactly the condition of Theorem 3.1, and all subsequent discussions in this section are based on this condition), the expression simplifies significantly to $\bar{r}^{(L)}(\boldsymbol{x}, \boldsymbol{x}') = r_0(\tilde{\boldsymbol{x}}, \tilde{\boldsymbol{x}}')/\big[2L(1+\alpha^2)^{L-1}\big]$, where $\tilde{\boldsymbol{x}} = \binom{\boldsymbol{x}}{1}/\left\| \binom{\boldsymbol{x}}{1} \right\| \in \mathbb{S}^d$ and $\tilde{\boldsymbol{x}}' = \binom{\boldsymbol{x}'}{1}/\left\| \binom{\boldsymbol{x}'}{1} \right\| \in \mathbb{S}^d$. By combining this result with the expression of $r_0(\tilde{\boldsymbol{x}}, \tilde{\boldsymbol{x}}')$ (see Equation (5)), we obtain

$$\bar{r}^{(L)}(\boldsymbol{x}, \boldsymbol{x}') = \frac{1}{2L} \sum_{\ell=1}^{L} \frac{B_{\ell+1}}{(1+\alpha^2)^{L-1}} \left[ (1+\alpha^2)^{\ell-1} \kappa_1\left(\frac{K_{\ell-1}}{(1+\alpha^2)^{\ell-1}}\right) + K_{\ell-1} \cdot \kappa_0\left(\frac{K_{\ell-1}}{(1+\alpha^2)^{\ell-1}}\right) \right]$$

$$= \frac{1}{2L} \sum_{\ell=1}^{L} \frac{B_{\ell+1}}{(1+\alpha^2)^{L-\ell}} \left[ \kappa_1\left(\frac{K_{\ell-1}}{(1+\alpha^2)^{\ell-1}}\right) + \frac{K_{\ell-1}}{(1+\alpha^2)^{\ell-1}} \cdot \kappa_0\left(\frac{K_{\ell-1}}{(1+\alpha^2)^{\ell-1}}\right) \right], \tag{13}$$

where $K_0 = \tilde{\boldsymbol{x}}^\top \tilde{\boldsymbol{x}}'$, $B_{L+1} = 1$, and the recurrence relations are given by

$$K_\ell = K_{\ell-1} + \alpha^2 (1+\alpha^2)^{\ell-1} \kappa_1\left(\frac{K_{\ell-1}}{(1+\alpha^2)^{\ell-1}}\right); \qquad B_\ell = B_{\ell+1}\left[1 + \alpha^2 \kappa_0\left(\frac{K_{\ell-1}}{(1+\alpha^2)^{\ell-1}}\right)\right].$$

From the form of Equation (13), we can see that by defining $P_{\ell+1} = B_{\ell+1}(1+\alpha^2)^{\ell-L}$ and $u_\ell = K_\ell(1+\alpha^2)^{-\ell}$, the normalized RNTK can be compactly expressed as

$$\bar{r}^{(L)}(\boldsymbol{x}, \boldsymbol{x}') = \frac{1}{2L} \sum_{\ell=1}^{L} P_{\ell+1}\big(\kappa_1(u_{\ell-1}) + u_{\ell-1} \cdot \kappa_0(u_{\ell-1})\big). \tag{14}$$

Based on the recurrence relation of $B_\ell$, we derive the following expression:

$$P_{\ell+1} = B_{\ell+1}(1+\alpha^2)^{\ell-L} = B_{\ell+2}(1+\alpha^2)^{\ell-L}\left[1 + \alpha^2 \kappa_0(u_\ell)\right] = B_{\ell+2}(1+\alpha^2)^{\ell+1-L}\frac{1+\alpha^2\kappa_0(u_\ell)}{1+\alpha^2}$$

$$= B_{\ell+3}(1+\alpha^2)^{\ell+2-L}\prod_{i=\ell}^{\ell+1}\frac{1+\alpha^2\kappa_0(u_i)}{1+\alpha^2} = \cdots = B_{L+1}\prod_{i=\ell}^{L-1}\frac{1+\alpha^2\kappa_0(u_i)}{1+\alpha^2} = \prod_{i=\ell}^{L-1}\frac{1+\alpha^2\kappa_0(u_i)}{1+\alpha^2}.$$

Finally, based on the recurrence relation for $u_\ell$, we get:

$$\frac{K_\ell}{(1+\alpha^2)^\ell} = \frac{K_{\ell-1}}{(1+\alpha^2)^\ell} + \frac{\alpha^2}{1+\alpha^2}\kappa_1\left(\frac{K_{\ell-1}}{(1+\alpha^2)^{\ell-1}}\right) \implies u_\ell = \frac{u_{\ell-1} + \alpha^2\kappa_1(u_{\ell-1})}{1+\alpha^2} = \varphi_1(u_{\ell-1}),$$

where $\varphi_1(\rho) = \left[\rho + \alpha^2\kappa_1(\rho)\right]/(1+\alpha^2)$.

### C.2. The limit of $u_\ell$ as $\ell \to \infty$

For $\boldsymbol{x}, \boldsymbol{x}' \in \mathbb{S}^{d-1}$, if $\boldsymbol{x} = \boldsymbol{x}'$, it is easy to verify that $u_0 = K_0 = \tilde{\boldsymbol{x}}^\top\tilde{\boldsymbol{x}} = 1$. Furthermore, using the recurrence relation's function $\varphi_1$, which satisfies $\varphi_1(1) = 1$, we conclude that $u_\ell = 1$ for all $\ell$. Based on this, we can further deduce from Equation (14) that

$$\bar{r}^{(L)}(\boldsymbol{x}, \boldsymbol{x}) = \frac{1}{2L}\sum_{\ell=1}^{L}\left(\prod_{i=\ell}^{L-1}\frac{1+\alpha^2\kappa_0(1)}{1+\alpha^2}\right)[\kappa_0(1) + \kappa_1(1)] = \frac{1}{L}\sum_{\ell=1}^{L-1}\prod_{i=\ell}^{L-1}1 = 1.$$

Hence, we only need to study the case when $\boldsymbol{x} \neq \boldsymbol{x}'$. To solve this, we introduce the following property of $\varphi_1$:

**Lemma C.1.** $\varphi_1 : [-1, 1] \to [-1/(1+\alpha^2), 1]$ is a monotonic increasing and convex function satisfying

$$0 \leq \frac{\sqrt{2}}{3\pi\beta}(1-\rho)^{\frac{3}{2}} \leq \varphi_1(\rho) - \rho \leq \frac{\sqrt{2}}{8\beta}(1-\rho)^{\frac{3}{2}}, \qquad \text{where } \beta = \beta(\alpha) = \frac{1+\alpha^2}{2\alpha^2} > \frac{1}{2} \tag{15}$$

and that equality holds if and only if $\rho = 1$.

*Proof.* By direct calculation, we have

$$\frac{\mathrm{d}\varphi_1(\rho)}{\mathrm{d}\rho} = 1 - \frac{\arccos\rho}{2\pi\beta} > \frac{1}{1+\alpha^2} > 0; \qquad \frac{\mathrm{d}^2\varphi_1(\rho)}{\mathrm{d}\rho^2} = \frac{1}{2\pi\beta\sqrt{1-\rho^2}} > 0.$$

Therefore, $\varphi_1$ is a monotonic increasing and convex function.

For *Equation* (15), it is easy to check that the equality holds for $\rho = 1$. If $\rho \neq 1$, let $f(\rho) = [\varphi_1(\rho) - \rho]/(1-\rho)^{3/2}$, then we can get

$$f(\rho) = \frac{\varphi_1(\rho) - \rho}{(1-\rho)^{\frac{3}{2}}} = \frac{\sqrt{1-\rho^2} - \rho\arccos\rho}{\pi\beta(1-\rho)^{\frac{3}{2}}}; \qquad f'(\rho) = \frac{3\sqrt{1-\rho^2} - (2+\rho)\arccos\rho}{2\beta(1-\rho)^{\frac{5}{2}}}.$$

Define $g(\rho) = 3\sqrt{1-\rho^2}/(2+\rho) - \arccos\rho$, we have $g'(\rho) = (\rho-1)^2/\left[(\rho+2)^2\sqrt{1-\rho^2}\right] > 0$, so $g(\rho) < g(1) = 0$ and $f'(\rho) < 0$. Finally, we can get

$$\frac{\sqrt{2}}{8\beta} = \lim_{\rho\to-1} f(\rho) > f(\rho) > \lim_{\rho\to1} f(\rho) = \frac{\sqrt{2}}{3\pi\beta}, \qquad \forall\rho \in [-1, 1).$$

$\square$

Because of $u_\ell = \varphi_1(u_{\ell-1}) \geq u_{\ell-1}$, we can get $\{u_\ell\}$ is an increasing sequence. Considering that $|u_\ell| \leq 1$, we have $u_\ell$ converges as $\ell \to \infty$. Taking the limit of both sides of $u_\ell = \varphi_1(u_{\ell-1})$, we have $u_\ell \to 1$ as $\ell \to \infty$.

Let $e_\ell = 1 - u_\ell \in [0, 2]$. Since $e_{\ell-1} - e_\ell = u_\ell - u_{\ell-1} = \varphi_1(u_{\ell-1}) - u_{\ell-1}$, we can get

$$e_{\ell-1} - \frac{\sqrt{2}}{8\beta} e_{\ell-1}^{\frac{3}{2}} \leq e_\ell \leq e_{\ell-1} - \frac{\sqrt{2}}{3\pi\beta} e_{\ell-1}^{\frac{3}{2}}$$

according to *Equation* (15). Hence as $e_\ell \to 0$, we have $e_\ell / e_{\ell-1} \to 1$, which implies $\{u_\ell\}$ converges sublinearly. More precisely, we have the following results:

**Lemma C.2.** *For each $u_0 < 1$, there exists $n_0 = n_0(u_0) > 0$, such that*

$$1 - \frac{18\pi^2\beta^2}{(n+3\pi\beta)^2} \leq u_n \leq 1 - \frac{18\pi^2\beta^2}{(n+n_0)^{2+\frac{\log(n+n_0)}{n+n_0}}}, \quad \forall n \in \mathbb{Z}_{\geq 0}.$$

*Proof.* For the left hand side, first we can easily check that

$$1 - \frac{18\pi^2\beta^2}{(n+3\pi\beta)^2} \in [-1, 1) \qquad \text{and} \qquad 1 - \frac{18\pi^2\beta^2}{(0+3\pi\beta)^2} = -1 \leq u_0.$$

Assuming that the left hand side holds for $n$. According to (*Equation* (15)) we have

$$\left(1 - \frac{18\pi^2\beta^2}{(n+3\pi\beta+1)^2}\right) - \varphi_1\left(1 - \frac{18\pi^2\beta^2}{(n+3\pi\beta)^2}\right)$$

$$\leq \left(1 - \frac{18\pi^2\beta^2}{(n+3\pi\beta+1)^2}\right) - \left(1 - \frac{18\pi^2\beta^2}{(n+3\pi\beta)^2}\right) - \frac{\sqrt{2}}{3\pi\beta}\left(\frac{18\pi^2\beta^2}{(n+3\pi\beta)^2}\right)^{\frac{3}{2}}$$

$$= \frac{-18\pi^2\beta^2(3n+9\pi\beta+2)}{(n+3\pi\beta)^3(n+3\pi\beta+1)^2} \leq 0.$$

Thus, we can get

$$u_{n+1} = \varphi_1(u_n) \geq \varphi_1\left(1 - \frac{18\pi^2\beta^2}{(n+3\pi\beta)^2}\right) \geq 1 - \frac{18\pi^2\beta^2}{(n+3\pi\beta+1)^2}.$$

Hence we have the left hand side.

For the right hand side, we have, by series expansion,

$$\left(1 - \frac{18\pi^2\beta^2}{(n+1)^{2+\frac{\log(n+1)}{n+1}}}\right) - \varphi_1\left(1 - \frac{18\pi^2\beta^2}{n^{2+\frac{\log n}{n}}}\right) \sim 36\pi^2\beta^2 \cdot \frac{\log n}{n^4},$$

which means that there exists $N$ such that when $n_0 > N$ we can get

$$\left(1 - \frac{18\pi^2\beta^2}{(n+1+n_0)^{2+\frac{\log(n+1+n_0)}{n+1+n_0}}}\right) - \varphi_1\left(1 - \frac{18\pi^2\beta^2}{(n+n_0)^{2+\frac{\log(n+n_0)}{n+n_0}}}\right) \geq 0. \tag{16}$$

Then, by choosing $n_0$ such that $n_0 > N$ and $n_0 \geq \sqrt{\frac{18\pi^2\beta^2}{1-u_0}}$, we have $u_0 \leq 1 - \frac{18\pi^2\beta^2}{n_0^{2+\frac{\log n_0}{n_0}}}$ and (16). Using the mathematical induction, we can get the conclusion. $\qquad\square$

In the following, let us denote by $N_\alpha$ a positive constant satisfying $\frac{1}{1-\left(\frac{2\beta-1}{2\beta}\right)^{1/3}} - 2 \leq N_\alpha \leq \frac{1}{1-\left(\frac{2\beta-1}{2\beta}\right)^{1/3}} - 1$.

Similar to the analysis in Huang et al. (2020), let $F(n) = \cos\left(2\pi\beta\left(1 - \left(\frac{n+N_\alpha}{n+N_\alpha+1}\right)^{3-\log^2 L/L}\right)\right)$ and let $N_0 = N_0(L)$ be the solution to the equation $F(n+1) = \varphi_1(F(n))$. Through similar analysis, we can obtain the following results:

$$\begin{cases} F(n+1) \geq \varphi_1(F(n)), & n \geq N_0; \\ F(n+1) \leq \varphi_1(F(n)), & n \leq N_0. \end{cases}$$

The following lemma provides the range of $N_0$:

**Lemma C.3.** *We have $N_0 \in \left[ \frac{9L}{2(\log L)^2} - \frac{\log L}{2}, \frac{9L}{2(\log L)^2} + \frac{1}{2}(\log L)^2 - 1 \right]$ when $L$ is large enough.*

*Proof.* By series expansion, we have

$$F\left( \frac{9L}{2(\log L)^2} - \frac{\log L}{2} + 1 \right) - \varphi_1\left( F\left( \frac{9L}{2(\log L)^2} - \frac{\log L}{2} \right) \right) \sim -\frac{32\pi^2\beta^2}{2187} \frac{(\log L)^{11}}{L^5}$$

and

$$F\left( \frac{9L}{2(\log L)^2} + \frac{1}{2}\log(L)^2 \right) - \varphi_1\left( F\left( \frac{9L}{2(\log L)^2} + \frac{1}{2}\log(L)^2 - 1 \right) \right) \sim \frac{32\pi^2\beta^2}{2187} \frac{(\log L)^{12}}{L^5}.$$

$\square$

Next we would like to find $n$ such that

$$u_n \leq \cos\left( 2\pi\beta \left( 1 - \left( \frac{\frac{9L}{2(\log L)^2} + N_\alpha - \frac{\log L}{2}}{\frac{9L}{2(\log L)^2} + N_\alpha + 1 - \frac{\log L}{2}} \right)^{3 - \frac{(\log L)^2}{L}} \right) \right).$$

By series expansion, we know

$$\cos\left( 2\pi\beta \left( 1 - \left( \frac{\frac{9L}{2(\log L)^2} + N_\alpha - \frac{\log L}{2}}{\frac{9L}{2(\log L)^2} + N_\alpha + 1 - \frac{\log L}{2}} \right)^{3 - \frac{(\log L)^2}{L}} \right) \right) \succeq 1 - \frac{18\pi^2\beta^2}{\left( \frac{9L}{2(\log L)^2} - \frac{\log L}{2} \right)^2}.$$

Then it suffices to solve

$$1 - \frac{18\pi^2\beta^2}{\left( \frac{9L}{2(\log L)^2} - \frac{\log L}{2} \right)^2} \succeq 1 - \frac{18\pi^2\beta^2}{(n+n_0)^{2 + \frac{\log(n+n_0)}{n+n_0}}} \geq u_n,$$

or equivalently, to solve

$$(n+n_0)^{2 + \frac{\log(n+n_0)}{n+n_0}} \preceq \left( \frac{9L}{2(\log L)^2} - \frac{\log L}{2} \right)^2. \tag{17}$$

**Lemma C.4.** *When $L$ is large enough, $n \leq \frac{9L}{2(\log L)^2} - \frac{1}{2}(\log L)^2$ satisfies (17).*

*Proof.* It is a straightforward computation to check that

$$(n+n_0)^{2 + \frac{\log(n+n_0)}{n+n_0}} - \left( \frac{9L}{2(\log L)^2} - \frac{\log L}{2} \right)^2$$

$$\leq \left( \frac{9L}{2(\log L)^2} - \frac{1}{2}(\log L)^2 + n_0 \right)^{2 + \frac{\log\left( \frac{9L}{2(\log L)^2} - \frac{1}{2}(\log L)^2 + n_0 \right)}{\frac{9L}{2(\log L)^2} - \frac{1}{2}(\log L)^2 + n_0}} - \left( \frac{9L}{2(\log L)^2} - \frac{\log L}{2} \right)^2$$

$$\sim -\frac{18L \log\log L}{\log L}.$$

$\square$

**Lemma C.5.** *For each $u_0 < 1$, we have*

$$\cos\left( 2\pi\beta \left( 1 - \left( \frac{n + N_\alpha}{n + N_\alpha + 1} \right)^3 \right) \right) \leq u_n \leq \cos\left( 2\pi\beta \left( 1 - \left( \frac{n + \log^2 L + N_\alpha}{n + \log^2 L + N_\alpha + 1} \right)^{3 - \frac{(\log L)^2}{L}} \right) \right), \quad \forall n \in [L].$$

*when $L$ is large enough.*

*Proof.* For the left hand side, we can easily check that

$$\cos\left(2\pi\beta\left(1-\left(\frac{n+N_\alpha}{n+N_\alpha+1}\right)^3\right)\right) \le 1 - \frac{18\pi^2\beta^2}{(n+3\pi\beta)^2} \le u_n.$$

For the right hand side, let $G(n) = \cos\left(2\pi\beta\left(1-\left(\frac{n+\log^2 L+N_\alpha}{n+\log^2 L+N_\alpha+1}\right)^{3-\frac{(\log L)^2}{L}}\right)\right) = F\left(n+(\log L)^2\right)$. We want to proof $u_n \le G(n)$.

Let $N_1 = N_0 - (\log L)^2 \in \left[\frac{9L}{2(\log L)^2} - \frac{1}{2}\log L - (\log L)^2, \frac{9L}{2(\log L)^2} - \frac{1}{2}(\log L)^2 - 1\right]$. When $n \ge N_1$, we have $n + (\log L)^2 \ge N_0$, which means that

$$\begin{cases} G(n+1) \ge \varphi_1\big(G(n)\big), & n \ge N_1; \\ G(n+1) \le \varphi_1\big(G(n)\big), & n \le N_1. \end{cases}$$

Let $N_2 = \lceil N_1 \rceil$ be the least integer greater than or equal to $N_1$, it is easy to see that

$$\frac{9L}{2(\log L)^2} - \frac{1}{2}(\log L) - (\log L)^2 \le N_1 \le N_2 \le N_1 + 1 \le \frac{9L}{2(\log L)^2} - \frac{1}{2}(\log L)^2.$$

Because of the monotonicity of $G(n)$ and *Lemma C.4*, we can get

$$G(N_2) \ge G\left(\frac{9L}{2(\log L)^2} - \frac{1}{2}(\log L) - (\log L)^2\right) = \cos\left(2\pi\beta\left[1-\left(\frac{\frac{9L}{2(\log L)^2}+N_\alpha-\frac{\log L}{2}}{\frac{9L}{2(\log L)^2}+N_\alpha+1-\frac{\log L}{2}}\right)^{3-\frac{(\log L)^2}{L}}\right]\right) \ge u_{N_2}.$$

Assuming that $u_n \le G(n)$ holds for $n = k$. If $k \ge N_2$, we have $k \ge N_1$ and

$$u_{k+1} = \varphi_1(u_k) \le \varphi_1(G_k) \le G_{k+1}.$$

Also, if $n = k \le N_2$, we can get $k \le N_1 + 1$ and

$$\varphi_1(u_{k-1}) = u_k \le G(k) \le \varphi_1\big(G(k-1)\big) \implies u_{k-1} \le G(k-1).$$

Therefore, we have the right hand side. □

## C.3. The limit of $r^{(L)}$ as $L \to \infty$

Denote $N_L = (\log L)^2 + N_\alpha$. Because $\kappa_0$ is a monotonic increasing function, we have

$$\kappa_0\left(\cos\left(2\pi\beta\left(1-\left(\frac{n+N_\alpha}{n+N_\alpha+1}\right)^3\right)\right)\right) \le \kappa_0(u_n) \le \kappa_0\left(\cos\left(2\pi\beta\left(1-\left(\frac{n+N_L}{n+N_L+1}\right)^{3-\frac{(\log L)^2}{L}}\right)\right)\right).$$

When $L$ is large enough, it is easy to see that

$$\beta\left(1-\left(\frac{n+N_\alpha}{n+N_\alpha+1}\right)^3\right) \in [0, 1/2] \text{ for } n \ge 0.$$

$$\beta\left(1-\left(\frac{n+N_L}{n+N_L+1}\right)^3\right) \in [0, 1/2] \text{ for } n \ge 0.$$

Thus

$$1 - 2\beta\left(1-\left(\frac{n+N_\alpha}{n+N_\alpha+1}\right)^3\right) \le \kappa_0(u_n) \le 1 - 2\beta\left(1-\left(\frac{n+N_L}{n+N_L+1}\right)^{3-\frac{(\log L)^2}{L}}\right).$$

i.e.

$$\left(\frac{n+N_\alpha}{n+N_\alpha+1}\right)^3 \leq \frac{1+\alpha^2\kappa_0(u_n)}{1+\alpha^2} \leq \left(\frac{n+N_L}{n+N_L+1}\right)^{3-\frac{(\log L)^2}{L}}.$$

Then

$$\left(\frac{\ell+N_\alpha}{L+N_\alpha+1}\right)^3 \leq \prod_{i=\ell}^{L}\frac{1+\alpha^2\kappa_0(u_{i-1})}{1+\alpha^2} \leq \left(\frac{\ell+N_L-1}{L+N_L}\right)^{3-\frac{(\log L)^2}{L}}.$$

For the right hand side, if we sum over $\ell$, we have

$$\frac{1}{L}\sum_{\ell=1}^{L}\left(\frac{\ell+N_L-1}{L+N_L}\right)^{3-\frac{(\log L)^2}{L}} \leq \frac{1}{L}\int_{1}^{L+1}\left(\frac{x+N_L-1}{L+N_L}\right)^{3-\frac{(\log L)^2}{L}} \mathrm{d}x$$

$$= \frac{(L+N_L)^{4-\frac{(\log L)^2}{L}} - N_L^{4-\frac{(\log L)^2}{L}}}{L(L+N_L)^{3-\frac{(\log L)^2}{L}}\left(4-\frac{(\log L)^2}{L}\right)}.$$

Similarly, we can get

$$\frac{1}{L}\sum_{i=1}^{L}\left(\frac{\ell+N_\alpha}{L+N_\alpha+1}\right)^3 \geq \frac{1}{L}\int_{1}^{L}\left(\frac{x+N_\alpha}{L+N_\alpha+1}\right)^3 \mathrm{d}x = \frac{(L+N_\alpha)^4-(N_\alpha+1)^4}{4L(L+N_\alpha+1)^3}.$$

Hence,

$$\frac{(L+N_\alpha)^4-(N_\alpha+1)^4}{4L(L+N_\alpha+1)^3} \leq \frac{1}{L}\sum_{\ell=1}^{L}\prod_{i=\ell}^{L}\frac{1+\alpha^2\kappa_0(u_{i-1})}{1+\alpha^2} \leq \frac{(L+N_L)^{4-\frac{(\log L)^2}{L}} - N_L^{4-\frac{(\log L)^2}{L}}}{L(L+N_L)^{3-\frac{(\log L)^2}{L}}\left(4-\frac{(\log L)^2}{L}\right)}.$$

Taking the limit of both sides, we have

$$\lim_{L\to\infty}\frac{(L+N_\alpha)^4-(N_\alpha+1)^4}{4L(L+N_\alpha+1)^3} = \lim_{L\to\infty}\frac{(L+N_L)^{4-\frac{(\log L)^2}{L}} - N_L^{4-\frac{(\log L)^2}{L}}}{L(L+N_L)^{3-\frac{(\log L)^2}{L}}\left(4-\frac{(\log L)^2}{L}\right)} = \frac{1}{4}.$$

Hence,

$$\lim_{L\to\infty}\frac{1}{L}\sum_{\ell=1}^{L}\left(\frac{\ell+N-1}{L+N}\right)^{3-\frac{(\log L)^2}{L}} = \lim_{L\to\infty}\frac{1}{L}\sum_{\ell=1}^{L}\left(\frac{\ell+N_\alpha}{L+N_\alpha+1}\right)^3$$

$$= \lim_{L\to\infty}\frac{1}{L}\sum_{\ell=1}^{L}\prod_{i=\ell}^{L}\frac{1+\alpha^2\kappa_0(u_{i-1})}{1+\alpha^2} = \frac{1}{4}.$$

Let $v_\ell = u_\ell\kappa_0(u_\ell) + \kappa_1(u_\ell)$, then

$$r^{(L)} = \frac{1}{L}\sum_{\ell=1}^{L}\frac{v_{\ell-1}}{2}\prod_{i=\ell}^{L}\frac{1+\alpha^2\kappa_0(u_{i-1})}{1+\alpha^2}.$$

Define $\varphi_0(x) = x\kappa_0(x) + \kappa_1(x)$, we can get

$$0 \leq 1 - \frac{v_\ell}{2} = \frac{1}{2}\big(\varphi_0(1) - \varphi_0(u_\ell)\big) = \frac{\sqrt{2}}{2\pi}(1-u_\ell)^{\frac{1}{2}} + \mathcal{O}(1-u_\ell).$$

Recall from previous discussion, $u_\ell = 1 - \mathcal{O}(\ell^{-2})$. Therefore, we have $\frac{v_\ell}{2} = 1 - \mathcal{O}(\ell^{-1})$ and

$$
\begin{aligned}
\lim_{L\to\infty} r^{(L)} &= \lim_{L\to\infty} \frac{1}{L}\sum_{\ell=1}^{L} \frac{v_{\ell-1}}{2} \prod_{i=\ell}^{L} \frac{1+\alpha^2\kappa_0(u_{i-1})}{1+\alpha^2} \\
&= \lim_{L\to\infty} \frac{1}{L}\sum_{\ell=1}^{L}\prod_{i=\ell}^{L} \frac{1+\alpha^2\kappa_0(u_{i-1})}{1+\alpha^2} - \lim_{L\to\infty} \frac{1}{L}\sum_{\ell=1}^{L}\mathcal{O}(\ell^{-1})\prod_{i=\ell}^{L}\frac{1+\alpha^2\kappa_0(u_{i-1})}{1+\alpha^2} \\
&= \frac{1}{4} - \lim_{L\to\infty} \frac{1}{L}\sum_{\ell=1}^{L}\mathcal{O}(\ell^{-1})\prod_{i=\ell}^{L}\frac{1+\alpha^2\kappa_0(u_{i-1})}{1+\alpha^2}.
\end{aligned}
$$

Because

$$
\begin{aligned}
\left| \frac{1}{L}\sum_{\ell=1}^{L}\mathcal{O}(\ell^{-1})\prod_{i=\ell}^{L}\frac{1+\alpha^2\kappa_0(u_{i-1})}{1+\alpha^2} \right| &\leq \frac{C}{L}\sum_{\ell=1}^{L}\frac{1}{\ell}\prod_{i=\ell}^{L}\frac{1+\alpha^2\kappa_0(u_{i-1})}{1+\alpha^2} \\
&\leq \frac{C}{L}\sum_{\ell=1}^{L}\frac{1}{\ell}\left(\frac{\ell+N_L-1}{L+N_L}\right)^{3-\frac{(\log L)^2}{L}} \leq \frac{C}{L}\sum_{\ell=1}^{L}\frac{(\ell+N_L)^3}{\ell\cdot L^{3-\frac{(\log L)^2}{L}}} \\
&\leq \frac{C}{L^{4-\frac{(\log L)^2}{L}}}\int_{1}^{L+1}\frac{(x+N_L)^3}{x}\,\mathrm{d}x \leq \frac{\mathcal{O}(L^3)}{L^{4-\frac{(\log L)^2}{L}}} = \mathcal{O}(L^{-1}) \to 0,
\end{aligned}
$$

we can finally get

$$
\lim_{L\to\infty} r^{(L)} = \frac{1}{4}.
$$

Also, when $L$ is large, we have

$$
\frac{(L+N_\alpha)^4 - (N_\alpha+1)^4}{4L(L+N_\alpha+1)^3} < \frac{1}{4} < \frac{(L+N_L)^{4-\frac{(\log L)^2}{L}} - N_L^{4-\frac{(\log L)^2}{L}}}{L(L+N_L)^{3-\frac{(\log L)^2}{L}}\left(4-\frac{(\log L)^2}{L}\right)}.
$$

Then

$$
\begin{aligned}
\left| \frac{1}{L}\sum_{\ell=1}^{L}\prod_{i=\ell}^{L}\frac{1+\alpha^2\kappa_0(u_{i-1})}{1+\alpha^2} - \frac{1}{4} \right| &\leq \left| \frac{(L+N_L)^{4-\frac{(\log L)^2}{L}} - N_L^{4-\frac{(\log L)^2}{L}}}{L(L+N_L)^{3-\frac{(\log L)^2}{L}}\left(4-\frac{(\log L)^2}{L}\right)} - \frac{(L+N_\alpha)^4 - (N_\alpha+1)^4}{4L(L+N_\alpha+1)^3} \right| \\
&\leq \left( \frac{(L+N_L)^{4-\frac{(\log L)^2}{L}} - N_L^{4-\frac{(\log L)^2}{L}}}{L(L+N_L)^{3-\frac{(\log L)^2}{L}}\left(4-\frac{(\log L)^2}{L}\right)} - \frac{1}{4} \right) + \left( \frac{1}{4} - \frac{(L+N_\alpha)^4 - (N_\alpha+1)^4}{4L(L+N_\alpha+1)^3} \right) \\
&\lesssim \frac{4N_L + (\log L)^2 + 4}{16L}.
\end{aligned}
$$

Finally we can estimate the convergence rate of the kernel

$$
\begin{aligned}
\left| \frac{1}{L}\sum_{\ell=1}^{L}\frac{v_{\ell-1}}{2}\prod_{i=\ell}^{L}\frac{1+\alpha^2\kappa_0(u_{i-1})}{1+\alpha^2} - \frac{1}{4} \right| &= \left| \frac{1}{L}\sum_{\ell=1}^{L}\left(1-\mathcal{O}(\ell^{-1})\right)\prod_{i=\ell}^{L}\frac{1+\alpha^2\kappa_0(u_{i-1})}{1+\alpha^2} - \frac{1}{4} \right| \\
&= \left| \frac{1}{L}\sum_{\ell=1}^{L}\prod_{i=\ell}^{L}\frac{1+\alpha^2\kappa_0(u_{i-1})}{1+\alpha^2} - \frac{1}{4} \right| + \left| \frac{1}{L}\sum_{\ell=1}^{L}\mathcal{O}(\ell^{-1})\prod_{i=\ell}^{L}\frac{1+\alpha^2\kappa_0(u_{i-1})}{1+\alpha^2} \right| \\
&\lesssim \frac{4N_L + (\log L)^2 + 4}{16L} + \mathcal{O}(L^{-1}) = \mathcal{O}\left(\frac{\mathrm{poly}\log(L)}{L}\right).
\end{aligned}
$$

# D. Proof of Theorem 3.4

In the following, let us denote $N_\alpha = 3L^{2\gamma}$ on $\alpha = L^{-\frac{1}{4}}$ satisfying

$$\frac{1}{1 - \left(\frac{2\beta-1}{2\beta}\right)^{1/3}} - 2 \le N_\alpha \le \frac{1}{1 - \left(\frac{2\beta-1}{2\beta}\right)^{1/3}} - 1$$

when $L$ is large enough.

Similar to the analysis in Huang et al. (2020), let $F(n) = \cos\left(2\pi\beta\left(1 - \left(\frac{n+N_\alpha}{n+N_\alpha+1}\right)^{3-\log^2 L/L}\right)\right)$ and let $N_0 = N_0(L)$ be the solution to the equation $F(n+1) = \varphi_1\big(F(n)\big)$. Through similar analysis, we can obtain the following results:

$$\begin{cases} F(n+1) \ge \varphi_1\big(F(n)\big), & n \ge N_0; \\ F(n+1) \le \varphi_1\big(F(n)\big), & n \le N_0. \end{cases}$$

The following lemma provides the range of $N_0$:

**Lemma D.1.** *We have* $N_0 \in \left[\frac{3\sqrt{5}\pi L}{5\log L}, \frac{3\sqrt{5}\pi L}{5\log L} + \frac{3\sqrt{5}\pi L}{4\log^2 L} - 1\right]$ *when $L$ is large enough.*

*Proof.* By series expansion, we have

$$F\left(\frac{3\sqrt{5}\pi L}{5\log L} + 1\right) - \varphi_1\left(F\left(\frac{3\sqrt{5}\pi L}{5\log L}\right)\right) \sim -\frac{25}{6\pi^2}\frac{(\log L)^4}{L^3}$$

and

$$F\left(\frac{3\sqrt{5}\pi L}{5\log L} + \frac{3\sqrt{5}\pi L}{4\log^2 L}\right) - \varphi_1\left(F\left(\frac{3\sqrt{5}\pi L}{5\log L} + \frac{3\sqrt{5}\pi L}{4\log^2 L} - 1\right)\right) \asymp \frac{51200\log^{10}(L)}{3\pi(4\log(L)+5)^6 L^3}\left(\frac{\sqrt{5}}{3} - \frac{1}{\pi}\right).$$

$\square$

Next we would like to find $n$ such that

$$u_n \le \cos\left(2\pi\beta\left(1 - \left(\frac{\frac{3\sqrt{5}\pi L}{5\log L} + N_\alpha}{\frac{3\sqrt{5}\pi L}{5\log L} + N_\alpha + 1}\right)^{3 - \frac{(\log L)^2}{L}}\right)\right).$$

By series expansion, we know

$$\cos\left(2\pi\beta\left(1 - \left(\frac{\frac{3\sqrt{5}\pi L}{5\log L} + N_\alpha}{\frac{3\sqrt{5}\pi L}{5\log L} + N_\alpha + 1}\right)^{3 - \frac{(\log L)^2}{L}}\right)\right) \succeq 1 - \frac{18\pi^2\beta^2}{\left(\frac{3\sqrt{5}\pi L}{5\log L} + N_\alpha\right)^2}.$$

Then it suffices to solve

$$1 - \frac{18\pi^2\beta^2}{\left(\frac{3\sqrt{5}\pi L}{5\log L} + N_\alpha\right)^2} \succeq 1 - \frac{18\pi^2\beta^2}{(n+n_0)^{2+\frac{\log(n+n_0)}{n+n_0}}} \ge u_n,$$

or equivalently, to solve

$$(n+n_0)^{2+\frac{\log(n+n_0)}{n+n_0}} \preceq \left(\frac{3\sqrt{5}\pi L}{5\log L} + N_\alpha\right)^2. \tag{18}$$

**Lemma D.2.** *When $L$ is large enough, $n \leq \frac{3\sqrt{5}\pi L}{5 \log L}$ satisfies* (18).

*Proof.* It is a straightforward computation to check that

$$(n + n_0)^{2 + \frac{\log(n + n_0)}{n + n_0}} - \left( \frac{3\sqrt{5}\pi L}{5 \log L} + N_\alpha \right)^2$$

$$\leq \left( \frac{3\sqrt{5}\pi L}{5 \log L} \right)^{2 + \frac{\log\left( \frac{3\sqrt{5}\pi L}{5 \log L} + n_0 \right)}{\frac{3\sqrt{5}\pi L}{5 \log L} + n_0}} - \left( \frac{3\sqrt{5}\pi L}{5 \log L} + N_\alpha \right)^2$$

$$\sim -\frac{\sqrt{5}\pi L^{3/2}}{\log L}.$$

$\square$

**Lemma D.3.** *For each $u_0 < 1$, we have*

$$\cos\left( 2\pi\beta \left( 1 - \left( \frac{n + N_\alpha}{n + N_\alpha + 1} \right)^3 \right) \right) \leq u_n \leq \cos\left( 2\pi\beta \left( 1 - \left( \frac{n + \frac{3\sqrt{5}\pi L}{4 \log^2 L} + N_\alpha}{n + \frac{3\sqrt{5}\pi L}{4 \log^2 L} + N_\alpha + 1} \right)^{3 - \frac{(\log L)^2}{L}} \right) \right), \quad \forall n \in [L].$$

*when $L$ is large enough.*

*Proof.* For the left hand side, we can easily check that

$$\cos\left( 2\pi\beta \left( 1 - \left( \frac{n + N_\alpha}{n + N_\alpha + 1} \right)^3 \right) \right) \leq 1 - \frac{18\pi^2\beta^2}{(n + 3\pi\beta)^2} \leq u_n$$

For the right hand side, let $G(n) = \cos\left( 2\pi\beta \left( 1 - \left( \frac{n + \frac{3\sqrt{5}\pi L}{4 \log^2 L} + N_\alpha}{n + \frac{3\sqrt{5}\pi L}{4 \log^2 L} + N_\alpha + 1} \right)^{3 - \frac{(\log L)^2}{L}} \right) \right) = F\left( n + \frac{3\sqrt{5}\pi L}{4 \log^2 L} \right)$. We want to

proof $u_n \leq G(n)$.

Let $N_1 = N_0 - \frac{3\sqrt{5}\pi L}{4 \log^2 L} \in \left[ \frac{3\sqrt{5}\pi L}{5 \log L} - \frac{3\sqrt{5}\pi L}{4 \log^2 L}, \frac{3\sqrt{5}\pi L}{5 \log L} - 1 \right]$. When $n \geq N_1$, we have $n + \frac{3\sqrt{5}\pi L}{4 \log^2 L} \geq N_0$, which means that

$$\begin{cases} G(n + 1) \geq \varphi_1\big(G(n)\big), & n \geq N_1; \\ G(n + 1) \leq \varphi_1\big(G(n)\big), & n \leq N_1. \end{cases}$$

Let $N_2 = \lceil N_1 \rceil$ be the least integer greater than or equal to $N_1$, it is easy to see that

$$\frac{3\sqrt{5}\pi L}{5 \log L} - \frac{3\sqrt{5}\pi L}{4 \log^2 L} \leq N_1 \leq N_2 \leq N_1 + 1 \leq \frac{3\sqrt{5}\pi L}{5 \log L}.$$

Because of the monotonicity of $G(n)$ and *Lemma D.2*, we can get

$$G(N_2) \geq G\left( \frac{3\sqrt{5}\pi L}{5 \log L} - \frac{3\sqrt{5}\pi L}{4 \log^2 L} \right) = \cos\left( 2\pi\beta \left[ 1 - \left( \frac{\frac{3\sqrt{5}\pi L}{5 \log L} + N_\alpha}{\frac{3\sqrt{5}\pi L}{5 \log L} + N_\alpha + 1} \right)^{3 - \frac{(\log L)^2}{L}} \right] \right) \geq u_{N_2}.$$

Assuming that $u_n \leq G(n)$ holds for $n = k$. If $k \geq N_2$, we have $k \geq N_1$ and

$$u_{k+1} = \varphi_1(u_k) \leq \varphi_1(G_k) \leq G_{k+1}.$$

Also, if $n = k \leq N_2$, we can get $k \leq N_1 + 1$ and

$$\varphi_1(u_{k-1}) = u_k \leq G(k) \leq \varphi_1\big(G(k-1)\big) \implies u_{k-1} \leq G(k-1).$$

Therefore, we have the right hand side. $\square$

Then as the same reasoning of Section C.3, we can complete the proof by letting $N_L = \frac{3\sqrt{5}\pi L}{4\log^2 L} + N_\alpha$.

# E. Proofs of Other Propositions in Section 3

## E.1. Proof of Theorem 3.3

We first present the following result:

**Proposition E.1** (modified by Theorem 3 in Belfer et al. (2024)). *RNTK $\overline{r}^{(L)}$ for ResNets with the hyperparameter $\alpha = L^{-\gamma}, \gamma \in (1/2, 1]$, approaches the 1-hidden-layer RNTK uniformly in the interval $\boldsymbol{x}^\top \boldsymbol{x}' \in [-1, 1]$, where $\boldsymbol{x}, \boldsymbol{x}' \in \mathbb{S}^{d-1}$; that is, let $\epsilon > 0$, for any $L > c(\epsilon, \gamma)$,*

$$|\overline{r}^{(L)}(\boldsymbol{x}, \boldsymbol{x}') - \overline{r}^{(1)}(\boldsymbol{x}, \boldsymbol{x}')| \leq \epsilon,$$

*where $c$ is a constant depending on $\epsilon$ and $\gamma$.*

*Remark* E.2. Theorem 3 in Belfer et al. (2024) addresses the case without bias terms. It can be straightforwardly extended to the setting considered in this paper, where the first layer incorporates bias terms.

This Proposition shows that when $\gamma \in (1/2, 1]$, RNTK tends to one-hidden-layer RNTK (see e.g., Huang et al. (2020); Belfer et al. (2024)) as $L \to \infty$. Thus, the limit of RNTK has adaptability to real distributions and performs better than infinite-depth RNTK when $\alpha$ is an arbitrary constant or decay with increasing $L$ at a slow decay rate.

The generalization capability of kernel regression is given by the following proposition:

**Proposition E.3** (Theorem 1 in Zhang et al. (2024)). *Suppose Assumption 3.2 holds. For any given $\delta \in (0, 1)$, if the training process is stopped at $t_* \propto n^{\beta/(s\beta+1)}$ for the kernel regression with the kernel $K$ and the eigen-decay rate $\beta > 1$, then for sufficiently large $n$, there exists a constant $C$ independent of $\delta$ and $n$, such that*

$$\mathcal{E}(f_{t_*}^K) \leq Cn^{-\frac{s\beta}{s\beta+1}}(6/\delta)^2$$

*holds with probability at least $1 - \delta$.*

According to the aforementioned proposition, it suffices to verify that the eigenvalue decay rate on $\mathbb{S}^{d-1}$ is $d/(d-1)$ in order to complete the proof of Theorem 3.3. From Equation (13), for $L = 1$, the expression of the NTK can be derived as follows:

$$\overline{r}^{(1)}(\boldsymbol{x}, \boldsymbol{x}') = \frac{1}{2}\left[\kappa_1\left(\frac{\boldsymbol{x}^\top \boldsymbol{x}'+1}{2}\right) + \frac{\boldsymbol{x}^\top \boldsymbol{x}'+1}{2} \cdot \kappa_0\left(\frac{\boldsymbol{x}^\top \boldsymbol{x}'+1}{2}\right)\right] = \text{NTK}^{(1)}(\boldsymbol{x}^\top \boldsymbol{x}'),$$

where

$$\text{NTK}^{(1)}(u) = \frac{1}{2}\left[\kappa_1\left(\frac{u+1}{2}\right) + \frac{u+1}{2} \cdot \kappa_0\left(\frac{u+1}{2}\right)\right].$$

If a kernel function $K$ can be expressed as a function $\kappa$ of the dot product of the inputs $\boldsymbol{x}^\top \boldsymbol{x}'$ (such as $\overline{r}^{(1)}(\boldsymbol{x}, \boldsymbol{x}')$), then it is called a dot-product kernel. For dot-product kernels, we have the following decomposition:

$$K(\boldsymbol{x}, \boldsymbol{x}') = \sum_{k=0}^{\infty} \mu_k \sum_{h=1}^{N(d,k)} Y_{k,h}(\boldsymbol{x})Y_{k,h}(\boldsymbol{x}'), \text{ where } N(d, k) = \frac{\Gamma(k+d-2)}{\Gamma(d-1)\Gamma(k)}, \tag{19}$$

and $Y_{k,j}$ is the $k$-th spherical harmonic polynomial of degree $k$. In addition, $\mu_k$ can also be computed using the following result:

**Lemma E.4** (Theorem 1 in Bietti & Bach (2021)). *Let $\kappa : [-1, 1] \to \mathbb{R}$ be a function that is $C^\infty$ on $(-1, 1)$ and has the following asymptotic expansions around $\pm 1$:*

$$\kappa(1 - t) = p_1(t) + c_1 t^\nu + o(t^\nu),$$
$$\kappa(-1 + t) = p_{-1}(t) + c_{-1} t^\nu + o(t^\nu),$$

*for $t \geq 0$, where $p_1, p_{-1}$ are polynomials and $\nu > 0$ is not an integer. Also, assume that the derivatives of $\kappa$ admit similar expansions obtained by differentiating the above ones. Then, there is an absolute constant $C(d, \nu)$ depending on $d$ and $\nu$ such that:*

- *For $k$ even, if $c_1 \neq -c_{-1}$: $\mu_k \sim (c_1 + c_{-1})C(d, \nu)k^{-d-2\nu+1}$;*

- *For $k$ odd, if $c_1 \neq c_{-1}$: $\mu_k \sim (c_1 - c_{-1})C(d, \nu)k^{-d-2\nu+1}$.*

*In the case $|c_1| = |c_{-1}|$, then we have $\mu_k = o(k^{-d-2\nu+1})$ for one of the two parities (or both if $c_1 = c_{-1} = 0$). If $\kappa$ is infinitely differentiable on $[-1, 1]$ so that no such $\nu$ exists, then $\mu_k$ decays faster than any polynomial.*

By comparing (1) and (19), we can establish the relationship between the $j$-th eigenvalue of the kernel function $k$ and $\mu_l$ as follows:

$$\lambda_j = \mu_l, \quad \sum_{i=1}^{l-1} N(d, 2i) \leq j \leq \sum_{i=1}^{l} N(d, 2i),$$

By Stirling approximation, we have $\Gamma(x) = Cx^{x-1/2}\exp(-x)(1 + \mathcal{O}(1/x))$. Therefore, $N(d, k)$ is of order $k^{d-2}$ for large $k$. Therefore, the only remaining task in proving Theorem 3.3 is to show that $\mu_k \sim k^{-d}$ for $\text{NTK}^{(1)}$.

Combine Lemma E.4 and the following expansions

$$\text{NTK}^{(1)}(1 - t) = 1 - \frac{t^{1/2}}{2\pi} + O(t^{1/2});$$
$$\text{NTK}^{(1)}(-1 + t) = \frac{1}{2\pi} + 0 \cdot t^{1/2} + O(t^{1/2}),$$

we can conclude that $\mu_k \sim k^{-d}$ for $\text{NTK}^{(1)}$, which completes the proof.

