# OpenReview forum: "Branch Scaling Manifests as Implicit Architectural Regularization for Improving Generalization in Overparameterized ResNets"
_ICML.cc/2026/Conference — ICML 2026 regular_

### Official Review · Reviewer_K4mM · 2026-03-10

**Soundness:** 3
**Presentation:** 3
**Significance:** 3
**Originality:** 3
**Overall Recommendation:** 5
**Confidence:** 3

**Summary:**

Branch scaling, i.e., the use of different multiplicative factors in the branches of residual connections, could be a more efficient alternative to the use of normalization (such as batch- or layernorm). In this work, the authors compare the generalization capabilities between residual networks with and without constant branch scaling factors. To this end, they resort to the infinite width limit and use the (residual) neural tangent kernel framework to obtain generalization bounds. They find that only properly scaled residual branches (i.e., branches with scaling factors rapidly decreasing with depth)  allow the model to preserve its generalization capacity as depth increases.

**Compliance With Llm Reviewing Policy:**

Affirmed.

**Key Questions For Authors:**

/

**Limitations:**

yes

**Strengths And Weaknesses:**

**Soundness**

The mathematical analysis presented seems sound, but I could not check all the details.

As pointed out by the authors in the limitations section, the RNTK regime has quite strong assumptions, and ‘real’, finite width neural networks do not adhere to them. But the authors tried to show the validity of their theoretical results with a set of experiments using more practical datasets and a finite-width resnet. But this remains quite limited. (But one could also argue this is a theory-focused paper, and more extensive experiments are out of scope).

On important flaw in the experiments I noted is that the experiments were not repeated over random seeds. The authors should report/visualize confidence intervals, as they do in fig 1.

**Presentation**

The paper is clearly written and well-structured. It contains an introduction to the important used concepts, such as RNTK, but these do remain quite inaccessible for readers with a more general background. I would suggest adding a few sentences that explain the concepts at a higher, less math-heavy level.


**Significance/originality**

The paper advances our understanding of branch scaling, which (according to the paper), possibly forms an efficient alternative to normalization. It could both serve as a starting point to develop the best branch scaling strategies in practical settings, as wel as to develop more advanced theoretical frameworks to analyse both normalisation and branch scaling.

However, it would be beneficial to better position the paper with respect to the earlier work:
Huang, Kaixuan, et al. "Why Do Deep Residual Networks Generalize Better than Deep Feedforward Networks?---A Neural Tangent Kernel Perspective." Advances in neural information processing systems 33 (2020): 2698-2709.
The authors cite this paper, but there seems to be more overlap than discussed. This would allow to better asess the novelty and contribution of this work.

---

> ### Author Rebuttal · Authors · 2026-03-30
>
> We sincerely thank the reviewer for the positive assessment and for the constructive suggestions.
>
> > **W1: On the Limitations of Theory and Experiment.**
>
> - As you pointed out, and as we also noted in the limitations section, the RNTK framework relies on relatively strong assumptions and cannot fully capture all the complexities of real finite-width networks. We do not seek to downplay this limitation. Our main reason for adopting this framework is that a rigorous theoretical analysis of deep networks remains extremely challenging in realistic settings, while NTK/RNTK provides a tractable approximation that allows us to isolate the core mechanism of interest in this paper, namely residual branch scaling, and analyze its effect on generalization as rigorously as possible.
>
> - Regarding the relatively limited scale of the experiments, we would like to clarify that the main purpose of our empirical section is to validate the qualitative trends revealed by the theory, rather than to provide a large-scale empirical benchmark. Nevertheless, we have already observed consistent qualitative behavior across synthetic RNTK regression, MNIST, finite-width ConvResNet experiments, as well as the attention-based and Tiny-ImageNet experiments included in the appendix. We also appreciate the reviewer’s understanding of the theory-oriented nature of this work, and fully agree that sufficient empirical validation is important even for theoretical research. We are also committed to strengthening the empirical support as much as possible.
>
> - At the same time, we agree that reporting variability across random seeds would strengthen the empirical evidence. In the current version, Figure 1 already reports the standard error across multiple runs, whereas Figures 2–4 do not yet include the corresponding uncertainty quantification. In the revision, we will repeat the relevant experiments with multiple random seeds and report the corresponding confidence intervals or error bars.
>
> > **W2: On accessibility of the RNTK discussion.**
>
> Thank you for this helpful suggestion. We agree that the presentation can be made more accessible to readers without a strong NTK background. In the revision, we will add a short high-level explanation early in Section 2 to clarify the intuition behind RNTK, including why it provides a useful approximation for wide residual networks and how this connects to our generalization analysis.
>
> > **W3: On the positioning relative to Huang et al. (2020).**
>
> We sincerely thank the reviewer for this suggestion. Huang et al. (2020) is indeed a foundational work in this line of research and has provided important insights and a useful framework for understanding the advantages of ResNets. In the current version, we cited it mainly for the basic ResNet/RNTK setup and related technical details. In the revision, we will more clearly explain both the connections and the distinctions between our work and Huang et al. (2020), and we will add a dedicated comparison discussion.
>
> - **At the level of the main research question, the two works focus on different issues:** Huang et al. (2020) mainly studies why deep residual networks exhibit better properties than deep feedforward networks, which has been a central question since the introduction of ResNets. In contrast, our paper studies how residual branch scaling affects generalization as depth grows. In this sense, we focus on a class of structural/parametrization mechanisms closely related to deep-network stability, in the same broader research line as work on normalization and normalization-free parametrization schemes such as Fixup.
>
> Although our analysis builds on Huang et al. (2020), our paper goes beyond it in two technical aspects:
>
> - **From initialization to the full training process:**
> Huang et al. (2020) highlights the role of the RNTK structure at initialization in understanding deep residual networks. Building on this, we further prove that the empirical RNTK converges uniformly to its limiting kernel throughout training. Thus, our analysis is not limited to static kernel properties at initialization, but extends to unified control over the full training dynamics.
>
> - **From kernel analysis to generalization-error characterization:**
> Huang et al. (2020) primarily analyzes kernel properties, whereas our work further extends the analysis to a rigorous characterization at the level of generalization error. Specifically, we not only characterize the relationship between the empirical and limiting kernels along training, but also extend this control to the analysis of generalization error.
>
> These two advances are precisely what allow us to rigorously distinguish how different residual branch scaling regimes affect generalization as depth grows, showing that this effect is not merely an optimization phenomenon, but admits a rigorous characterization at the level of generalization rather than only an intuition based on NTK behavior.

---

> > ### Author Rebuttal · Reviewer_K4mM · 2026-04-02
> >
> > The issues I raised were minor either way. I thank the authors for their rebuttal and would like to congratulate them with their work. I keep my score at accept.

---

> > > ### Author Response · Authors · 2026-04-07
> > >
> > > We sincerely thank the reviewer for the congratulations and the support for our work. We are glad that our response has fully addressed your concerns. We will faithfully incorporate your suggestions into the final version to further improve the manuscript. Thank you again for your valuable guidance.

---

### Official Review · Reviewer_T3wC · 2026-03-12

**Soundness:** 3
**Presentation:** 3
**Significance:** 2
**Originality:** 2
**Overall Recommendation:** 4
**Confidence:** 3

**Summary:**

This paper studies how residual-branch scaling affects the generalization behavior of overparameterized ResNets through the NTK/RNTK lens. The main claim is that wide ResNets can be approximated by kernel regression with the Residual Neural Tangent Kernel (RNTK), and that constant residual scaling becomes unfavorable as depth grows, whereas sufficiently decaying depth-dependent scaling, together with early stopping, yields improved generalization and minimax-optimal rates under a source-condition setting.

The topic is relevant and potentially interesting. Residual scaling is an important design component in modern deep architectures, and a theory-oriented treatment of its role in generalization is worthwhile. However, my overall assessment is limited mainly by concerns about **originality** and, to a lesser extent, **practical significance beyond the NTK regime**. In particular, several closely related prior works already analyze deep ResNets from the NTK perspective, study scaling-dependent behavior of residual kernels, and in some cases make claims very similar to the headline message of this paper.

**Compliance With Llm Reviewing Policy:**

Affirmed.

**Final Justification:**

I raise my evaluation.

**Key Questions For Authors:**

1. **What is the precise theorem-level novelty over [3] and [4]?**
   Please provide a direct comparison table explaining, for each main theorem, exactly what is new relative to prior work. At present, the manuscript does not make this distinction sufficiently precise.

2. **How much of the claimed benefit is truly a generalization effect rather than an optimization effect?**
   More careful evidence would be needed to isolate the implicit-regularization interpretation. For example, depth sweeps, \(\gamma\)-sweeps in finite-width models, matched-training-loss comparisons, and stronger baseline comparisons would help.

3. **How robust are the conclusions beyond the specific NTK setup?**
   Please clarify more concretely which conclusions are expected to survive in standard finite-width training and which are best understood as kernel-regime phenomena only.

[3] J. Lai, Z. Yu, S. Tian, and Q. Lin. *Generalization Ability of Wide Residual Networks.* arXiv:2305.18506, 2023.

[4] S. Tian and Z. Yu. *Improve Generalization Ability of Deep Wide Residual Network with A Suitable Scaling Factor.* arXiv:2403.04545, 2024.

**Limitations:**

The main limitation is that the theory is developed under standard NTK assumptions and therefore does not fully capture realistic finite-width training dynamics. The analysis is also tied to a regression-style framework with early stopping and source-condition assumptions, so the broader practical conclusions should be interpreted with caution.

A second limitation is the literature positioning. Even if the technical results are correct, the paper does not convincingly establish that the contribution goes substantially beyond closely related existing papers and preprints.

**Strengths And Weaknesses:**

## Strengths

1. **The paper addresses a meaningful question.**
   Understanding residual scaling as more than an optimization trick is a worthwhile direction. Interpreting branch scaling as a form of implicit architectural regularization is an interesting angle.

2. **Theoretical development appears substantial.**
   The paper goes beyond heuristic discussion and attempts to connect wide-ResNet training dynamics to RNTK kernel regression uniformly over a compact input domain, then derive implications for depth-dependent scaling and generalization. If correct, this is mathematically nontrivial.

3. **The experiments are directionally aligned with the theory.**
   While not exhaustive, the empirical section does attempt to test the central qualitative claim: constant scaling degrades performance with increasing depth, while sufficiently decaying scaling improves behavior. The synthetic, MNIST, and CIFAR-100 results are at least consistent with the paper’s narrative.

## Weaknesses

1. **Originality is the main problem.**
   The paper’s core message is not sufficiently differentiated from closely related prior work:
   - Huang et al. [1] already studied deep ResNets from the NTK perspective and explained why they can generalize better than deep feedforward networks.
   - Belfer et al. [2] analyzed deep residual NTKs spectrally and explicitly discussed scaling-dependent stable versus spike-like behavior.
   - Lai et al. [3] already presented uniform convergence of wide residual-network kernels to the RNTK and argued that wide-ResNet generalization converges to that of RNTK kernel regression, including early-stopping-based rates.
   - Tian and Yu [4] stated an even closer claim: constant residual scaling becomes unfavorable as depth grows, while sufficiently decaying scaling plus early stopping improves generalization, with supporting theory and experiments.

   Because of this, the present submission reads more like a refinement or consolidation of existing preprint-level results than a clearly new conceptual advance. The manuscript does not make the exact theorem-level novelty sufficiently explicit.

2. **The positioning overstates novelty.**
   The manuscript presents the contribution as if the generalization story for residual scaling is largely missing from prior theory, but the literature above already covers much of that ground. In particular, the combination of
   - wide ResNet \(\approx\) RNTK regression,
   - constant scaling is bad at large depth, and
   - sufficiently decaying scaling plus early stopping is good,
   appears too close to [3,4] to support a strong originality score.

3. **The practical significance is limited by the theory-to-practice gap.**
   The theory is developed in a lazy-training / infinite-width / kernel-regime framework, while the practical takeaway is phrased broadly enough to suggest a general design principle for modern deep architectures. That jump is not fully justified by the current empirical evidence.

4. **The empirical comparison set is too narrow.**
   Since the paper argues that residual scaling is a meaningful architectural mechanism, I expected comparisons against stronger and more directly relevant baselines such as Fixup [5], ReZero [6], or DeepNorm / DeepNet-style residual stabilization methods [7]. Without such baselines, it is difficult to judge whether the paper provides practically new guidance or mainly reaffirms existing intuition.

5. **Presentation quality is meaningfully hurt by review-related artifacts in the manuscript.**
   The PDF contains text that looks like reviewer- or process-directed material embedded in the submission. Regardless of intent, this is inappropriate for a conference paper and negatively affects presentation quality.

[1] K. Huang, Y. Wang, M. Tao, and T. Zhao. *Why Do Deep Residual Networks Generalize Better than Deep Feedforward Networks? A Neural Tangent Kernel Perspective.* NeurIPS 2020.

[2] Y. Belfer, A. Brutzkus, and A. Globerson. *Spectral Analysis of the Neural Tangent Kernel for Deep Residual Networks.* JMLR 2024.

[3] J. Lai, Z. Yu, S. Tian, and Q. Lin. *Generalization Ability of Wide Residual Networks.* arXiv:2305.18506, 2023.

[4] S. Tian and Z. Yu. *Improve Generalization Ability of Deep Wide Residual Network with A Suitable Scaling Factor.* arXiv:2403.04545, 2024.

[5] H. Zhang, Y. N. Dauphin, and T. Ma. *Fixup Initialization: Residual Learning Without Normalization.* ICLR 2019.

[6] T. Bachlechner, B. P. Majumder, H. H. Mao, G. W. Cottrell, and J. McAuley. *ReZero is All You Need: Fast Convergence at Large Depth.* UAI 2021.

[7] H. Wang, S. Ma, L. Dong, S. Huang, D. Zhang, and F. Wei. *DeepNet: Scaling Transformers to 1,000 Layers.* 2022.

---

> ### Author Rebuttal · Authors · 2026-03-30
>
> Thank you for your valuable comments. We first briefly clarify two non-technical issues that may have affected the evaluation.
>
> > **(W1\&W2\&Q1\&L2)\&W5: [3,4] and the review-related content issue**
>
> Due to the **double-blind review policy**, we are unable to explain the background related to [3,4]; we have sought clarification through the AC. In addition, **we did not intentionally include any text directed to reviewers**. This issue is more likely related to the **system mechanism mentioned by the conference**. We therefore do not elaborate on these points here. However, they may have influenced the reviewer’s assessment, particularly the originality criticism.
>
> **Other technical questions:**
>
> > **W1: Relationship to [1,2]**
>
> A comparison with [1,2] is necessary, since both are important related works in the RNTK literature. However, [2] mainly studies properties of the RNTK itself; these results provide an important foundation. In contrast, our paper focuses more on using these properties to analyze the behavior of residual networks, which is not the main focus of [2]. For the parts of our paper that rely on these technical details, we have already cited them in the main text. As for [1], due to space limitations, please see our response to Reviewer K4mM under W3. A substantial part of that discussion, especially the explanation of technical differences and improvements, is also helpful for understanding how our work differs from [2].
>
> > **W3\&Q3\&L1: Limitations and Robustness**
>
> We do not seek to avoid the limitations of the RNTK framework, and we have already acknowledged them in the paper. NTK theory itself has intrinsic limitations; for example, the standard NTK perspective often downplays the effect of network depth differences [arXiv:2009.14397], which is one general limitation of NTK-based theory when explaining realistic deep learning phenomena.
>
> Real network learning dynamics are shaped jointly by kernel effects, feature learning, and other finite-width phenomena. Therefore, most theoretical works inevitably require abstraction and simplification. Early stopping and related settings are introduced for the same reason; for example, early stopping can be viewed as a simplified model of cross-validation in practice. These assumptions are not intended to fully characterize real training dynamics, but rather to preserve and analyze, within a controlled framework, the core mechanism we focus on.
>
> At the same time, we still believe this type of analysis is meaningful, because it helps identify and understand the role of a specific mechanism within a controlled theoretical framework. More concretely, we believe the conclusions here can be separated into two layers. First, the precise generalization rates, the early-stopping-based guarantees, and the strict theorem statements mainly belong to the kernel regime. Second, as depth increases, inappropriate residual branch scaling becomes increasingly harmful, while appropriate residual scaling may lead to better generalization performance; this qualitative trend is more likely to be at least partially preserved in finite-width practice. We agree that the latter should be presented as a cautiously supported practical implication, rather than as a universally established law, and we will revise the wording accordingly.
>
> > **W4\&Q2: Experimental Gaps**
>
> We acknowledge that the current experiments are still incomplete. Although Figure 4 already compares test accuracy when training accuracies are similar, we also understand that training losses may still differ even when training accuracies are close. Therefore, to further distinguish optimization effects from generalization effects, we added a **matched training loss evaluation**, namely comparing test accuracy when different methods reach a similar optimization level (training loss ≈ 0.05).
>
> In addition, to more fully illustrate the relation between the mechanism analyzed in our paper and existing residual-branch scaling methods, we added comparisons with **Fixup, ReZero, and DeepNorm**. We would like to emphasize that our paper is not proposing a completely new training module, nor are we claiming that the simplified branch scaling method studied here is practically superior to these strong baselines. Rather, our goal is to provide a partial mechanistic explanation for existing residual scaling methods. As long as these methods exhibit qualitative behavior consistent with our theoretical analysis, that is already sufficient to support the main conclusion of the paper.
>
> The experimental results are as follows:
>
> |Method|Matched Acc|
> |-|-|
> |base|96.66%|
> |fixup|97.84%|
> |rezero|97.49%|
> |deepnorm|97.30%|
> |proposed|97.29%|
>
> Due to time constraints, these additional experiments were only conducted on MNIST. Under matched optimization settings, residual scaling methods consistently achieved higher test accuracy than the unscaled baseline, which suggests that the gain is not purely an optimization artifact.

---

> > ### Author Rebuttal · Reviewer_T3wC · 2026-04-02
> >
> > Thank you for the response. My main concerns have been addressed to a satisfactory extent, and I therefore raise my score accordingly.

---

> > > ### Author Response · Authors · 2026-04-07
> > >
> > > We thank the reviewer for raising the score and for the positive feedback. We are glad that our response addressed your primary concerns. We will ensure all promised revisions and additional experiments are incorporated into the final version. Please let us know if you have any further questions; we are happy to provide more details.

---

### Official Review · Reviewer_6LjL · 2026-03-13

**Soundness:** 3
**Presentation:** 3
**Significance:** 3
**Originality:** 3
**Overall Recommendation:** 5
**Confidence:** 3

**Summary:**

The authors analyze the generalization error of ResNets under various scaling of the residual branch strength with depth.  The first establish a technical corollary establishing uniform convergence of the function learned by a wide ResNet to that learned by its infinite width limit under neural tangent kernel (NTK) scaling with respect to width.  This enables them to control the generalization error through an upper bound. Their key result is theorem 3.3 which upper bounds the generalization error of a sufficiently wide resnet in terms of the amount of training data with high probability when the residual strength decays with depth.  They perform several experiments showing qualitatively that if the residual branch strength is scaled in ways that are beneficial in theory, they are also beneficial in practice.

**Compliance With Llm Reviewing Policy:**

Affirmed.

**Ethical Review Concerns:**

None.

**Key Questions For Authors:**

I wrote my questions above in the strengths and weaknesses.

One more question:  in your experiments in Fig. 2 left, the solid curves are interpolations through experimental data points?  If so this should be stated.  An even stronger result could be that they are they theory curves that go through experimental data points?  Is it possible to put your theory and experimental curves on the same plot and see a match, as has been done in some of the work I think should be cited?  If not, what is the obstacle?  basically can there be a more quantitative comparison between theory and experiment than has been done so far, and if not what are the obstacles in your theoretical framework preventing such a quantitative comparison, as has been done in other works.  A discussion of this would be useful.

**Limitations:**

A main limitation is the lack of *quantitative* comparison between theory and experiment (for example through asympotically exact analysis instead of potentially loose upper bounds).

**Strengths And Weaknesses:**

Strengths:

Proving uniform convergence of the wide ResNet error to its infinite width NTK limit is a nice result (corollary 2.2)
Bounding the generalization error (Theorem 3.3 is a nice result)

Weaknesses:

At the same time, in Theorem 3.3, how does the structure of the input data distribution and/or the function to be learned play a role in the generalization bound?  This could be explained better.  How loose is the bound depending on the structure of the data or function? In what sense might be the bound not be vacuous as other generalization bounds based on worst case analyses have been?  What can you say about the constant C and its magnitude?

There is also a large literature on wide limits of neural networks, and analyses of kernels that has not been cited and should be.  Examples include:

1) https://papers.nips.cc/paper_files/paper/2016/hash/148510031349642de5ca0c544f31b2ef-Abstract.html (analyzed kernels in deep MLPs, and launched many studies on this)
2) https://openreview.net/pdf?id=H1W1UN9gg (showed optimal generalization at the edge of chaos numerically)
3) https://arxiv.org/abs/1712.08969 (mean field theory of ResNets - closely related to this paper)
Please check also the papers that this last paper cited and those that cite it to do justice to this line of literature that is closely related to this work. For example - the following also connects kernels and signal propagation to test error (though numerically) in transformers:
4) https://journals.aps.org/pre/abstract/10.1103/gjsm-l642

---

> ### Author Rebuttal · Authors · 2026-03-30
>
> We thank the reviewer for the positive assessment and helpful comments.
>
> > **W1: On the role of the data distribution / target function in Theorem 3.3, and on the constant $C$.**
>
> We agree that this point needs clearer explanation.
>
> - In our framework, the target function enters mainly through the regularity assumption $f^* \in [H]^s$. Here, $s$ characterizes the smoothness, or “learnability,” of $f^*$ relative to the kernel-induced function space: all else being equal, a larger $s$ leads to a smaller upper bound on the generalization error.
>
> - As for the data distribution, in kernel regression theory it affects generalization through the spectral structure of the corresponding integral operator, and the eigenvalue decay rate is a key quantity governing the generalization rate. In our paper, we focus on the uniform distribution on the sphere, under which the spectral decay of the NTK can be characterized relatively clearly.
>
> Regarding tightness, our conclusion should be understood at two levels.
>
> - First, under our setting and the source condition in Assumption 3.2, Theorem 3.3 is **tight at the rate level**, in the sense that it achieves the minimax-optimal rate over the corresponding function class. Here, “tight” refers to the exponent of $n$, not to a sharp characterization of the leading constant $C$. The constant $C$ may depend on problem-specific quantities such as the distribution, the kernel, and the regularity condition, but our current analysis does not provide a sharper characterization of its magnitude.
>
> - Second, the bound may still be loose at the instance level. Although we assume $f^* \in [H]^s$, the true target may be smoother than required by that class, so the actual error may decay faster than the theoretical worst-case rate. This is the usual meaning of a minimax result: it characterizes the optimal worst-case rate over a function class, rather than the exact behavior of every instance.
>
> Compared with coarse and structure-agnostic generalization bounds, our result is not merely a finite but weakly informative worst-case upper bound. It explicitly incorporates the regularity of the target function and the data / kernel structure, and under this setting yields a minimax-optimal generalization rate. We will clarify this point in the revision.
>
> > **W2: On related work.**
>
> Thank you for pointing us to these valuable references, which have been very helpful in broadening our perspective. We agree that the literature on wide-network limits, kernel analysis, signal propagation, and mean-field theory for ResNets provides important theoretical background for our work, and that the current draft does not review this line of research sufficiently. In the revision, we will incorporate these references more systematically in the Introduction and Related Work sections, and clarify more clearly how our work is related to and differs from these prior studies.
>
> > **Q: On Fig. 2 and the possibility of a more quantitative theory–experiment comparison.**
>
> Thank you for this insightful question.
>
> First, the solid curves in Figure 2 (left) are **not interpolations through discrete experimental points**, nor are they theoretical prediction curves. They are empirically evaluated test-error trajectories during training. Since the evaluation points are dense, we only added sparse markers as a visual aid. We apologize for the lack of clarity and will revise the caption accordingly.
>
> We agree that a more direct quantitative comparison between theory and experiment would be valuable. However, under our current framework, there are obstacles to obtaining such an exact overlay.
>
> - Most importantly, our theory, especially Theorem 3.3, provides a **high-probability upper bound and a minimax-optimal rate**, rather than an asymptotically exact finite-sample learning curve. Its purpose is to characterize the scaling behavior of the excess risk and the distinction between residual-scaling regimes, rather than to generate a pointwise theoretical trajectory that can be directly superimposed on the empirical curves.
>
> - Moreover, even a more quantitative comparison at the rate level is nontrivial. Theorem 3.3 relies on the source condition $f^* \in [H]^s$, but for a concrete target function it is generally difficult to estimate the corresponding smoothness parameter $s$ in practice. Conversely, constructing target functions with a prescribed $s$ would also require a more refined understanding of the kernel’s spectral properties. Thus, although Theorem 3.3 is tight at the rate level, the current analysis does not yield a sharp finite-sample numerical prediction that could be directly plotted as a theory curve.
>
> Following the reviewer’s suggestion, we will add a brief discussion in the revision, especially in the experimental clarification and limitations sections, to make clearer the distinction between establishing minimax rates / upper bounds and deriving exact theoretical learning curves.

---

> > ### Author Rebuttal · Reviewer_6LjL · 2026-04-03
> >
> > Thank you for your response - it addresses my concerns and i retain my score of accept.

---

> > > ### Author Response · Authors · 2026-04-07
> > >
> > > We sincerely thank the reviewer for the positive final assessment and the support for our work. Your valuable comments have played a key role in further enhancing the quality of the manuscript.

---

### Decision · Program_Chairs · 2026-04-30

**Decision:**

Accept (regular)

**Comment:**

This paper argues that in wide overparameterized ResNets, constant residual-branch scaling becomes harmful as depth increases, making the model effectively unlearnable. By contrast, depth-decaying residual scaling, combined with early stopping, yields better generalisation and even minimax-optimal rates under certain assumptions. The authors frame branch scaling as a form of implicit architectural regularization, not merely an optimization trick. They support this with theory via the NTK/RNTK lens and experiments on synthetic data, MNIST, and CIFAR-100.

All reviewers agreed to accept this paper based on grounds of (1) it addresses an important question and contains substantial technical work, (2) it proves uniform convergence result connecting wide ResNets to their kernel limit, (3) the generalisation bound for suitably scaled residual networks is good, (4) good experiments that qualitatively align with the theory.